# Matrix viscoelasticity promotes liver cancer progression in the pre-cirrhotic liver

Weiguo Fan[1,2], Kolade Adebowale[3,4,17], Lóránd Váncza[1,2,17], Yuan Li[1,2], Md Foysal Rabbi[5], Koshi Kunimoto[1,2], Dongning Chen[1,2], Gergely Mozes[1,2], David Kung-Chun Chiu[6,7], Yisi Li[8], Junyan Tao[9], Yi Wei[1,2], Nia Adeniji[1,2], Ryan L. Brunsing[10], Renumathy Dhanasekaran[1,2], Aatur Singhi[9], David Geller[9], Su Hao Lo[11], Louis Hodgson[12], Edgar G. Engleman[6,7], Gregory W. Charville[6], Vivek Charu[6,13], Satdarshan P. Monga[9], Taeyoon Kim[5,14], Rebecca G. Wells[15], Ovijit Chaudhuri[4,16] & Natalie J. Török[1,2] ✉

Type 2 diabetes mellitus is a major risk factor for hepatocellular carcinoma (HCC). Changes in extracellular matrix (ECM) mechanics contribute to cancer development[1,2], and increased stiffness is known to promote HCC progression in cirrhotic conditions[3,4]. Type 2 diabetes mellitus is characterized by an accumulation of advanced glycation end-products (AGEs) in the ECM; however, how this affects HCC in non-cirrhotic conditions is unclear. Here we find that, in patients and animal models, AGEs promote changes in collagen architecture and enhance ECM viscoelasticity, with greater viscous dissipation and faster stress relaxation, but not changes in stiffness. High AGEs and viscoelasticity combined with oncogenic β-catenin signalling promote HCC induction, whereas inhibiting AGE production, reconstituting the AGE clearance receptor AGER1 or breaking AGE-mediated collagen cross-links reduces viscoelasticity and HCC growth. Matrix analysis and computational modelling demonstrate that lower interconnectivity of AGE-bundled collagen matrix, marked by shorter fibre length and greater heterogeneity, enhances viscoelasticity. Mechanistically, animal studies and 3D cell cultures show that enhanced viscoelasticity promotes HCC cell proliferation and invasion through an integrin-β1–tensin-1–YAP mechanotransductive pathway. These results reveal that AGE-mediated structural changes enhance ECM viscoelasticity, and that viscoelasticity can promote cancer progression in vivo, independent of stiffness.

Type 2 diabetes mellitus (T2DM) and obesity are important risks for HCC, and it is estimated that up to 30% of HCCs in non-alcoholic stea-tohepatitis (NASH) occur at a precirrhotic stage when matrix stiffness is still low, and these patients often have poor glycaemic control[5,6]. AGEs are produced by the non-enzymatic glycation of serum or tissue proteins during T2DM or can be ingested through the consumption of food prepared at high temperatures[7]. Over time, AGEs accumulate in the matrix due to decreased clearance and metabolism[8–10] and can biochemically modify collagen and ECM proteins[11]. Although AGEs in NASH do not appear to increase stiffness greatly, they could affect ECM viscoelasticity. Tissues and ECMs are generally viscoelastic, exhibiting viscous energy dissipation in response to mechanical perturbations and a time-dependent mechanical response, such as stress relaxation in response to deformation[12]. Recent research has shown that changes in ECM viscoelasticity, independent of stiffness, have impacted cell

behaviours, including proliferation and migration of breast cancer cells[13–16]. Viscoelasticity can modulate tissue growth dynamics, symmetry and the growth of cancer cells[17]. Here we investigated the role of AGE-mediated changes in ECM mechanical properties on NASH and HCC progression.

## AGEs enhance liver viscoelasticity

To evaluate the potential role of AGE-mediated changes on the ECM, we studied the mechanical properties of liver samples from patients with T2DM, and in individuals with NASH with or without T2DM (average disease activity score, 4; fibrosis stage, 0–1; Fig. 1a). Liver AGEs were significantly higher in patients with T2DM, or with NASH and T2DM compared with those without T2DM (Fig. 1b). Using atomic-force microscopy (AFM), we found that patients with T2DM

[1]Gastroenterology and Hepatology, Stanford University, Stanford, CA, USA. [2]VA, Palo Alto, CA, USA. [3]Department of Chemical Engineering, Stanford University, Stanford, CA, USA. [4]Chemistry, Engineering and Medicine for Human Health (ChEM-H), Stanford University, Stanford, CA, USA. [5]Weldon School of Biomedical Engineering, Purdue University, West Lafayette, IN, USA. [6]Department of Pathology, Stanford University, Stanford, CA, USA. [7]Division of Immunology, Stanford University, Stanford, CA, USA. [8]Department of Automation, Tsinghua University, Beijing, China. [9]Pittsburgh Liver Research Center, University of Pittsburgh and University of Pittsburgh Medical Center, Pittsburgh, PA, USA. [10]Department of Radiology, Stanford University, Stanford, CA, USA. [11]Department of Biochemistry and Molecular Medicine, University of California at Davis, Sacramento, CA, USA. [12]Department of Molecular Pharmacology, Albert Einstein College of Medicine, New York, NY, USA. [13]Quantitative Sciences Unit, Department of Medicine, Stanford University School of Medicine, Stanford, CA, USA. [14]Faculty of Science and Technology, Keio University, Yokohama, Japan. [15]Departments of Medicine and Bioengineering, University of Pennsylvania, Philadelphia, PA, USA. [16]Department of Mechanical Engineering, Stanford University, Stanford, CA, USA. [17]These authors contributed equally: Kolade Adebowale, Lóránd Váncza. ✉e-mail: ntorok@stanford.edu

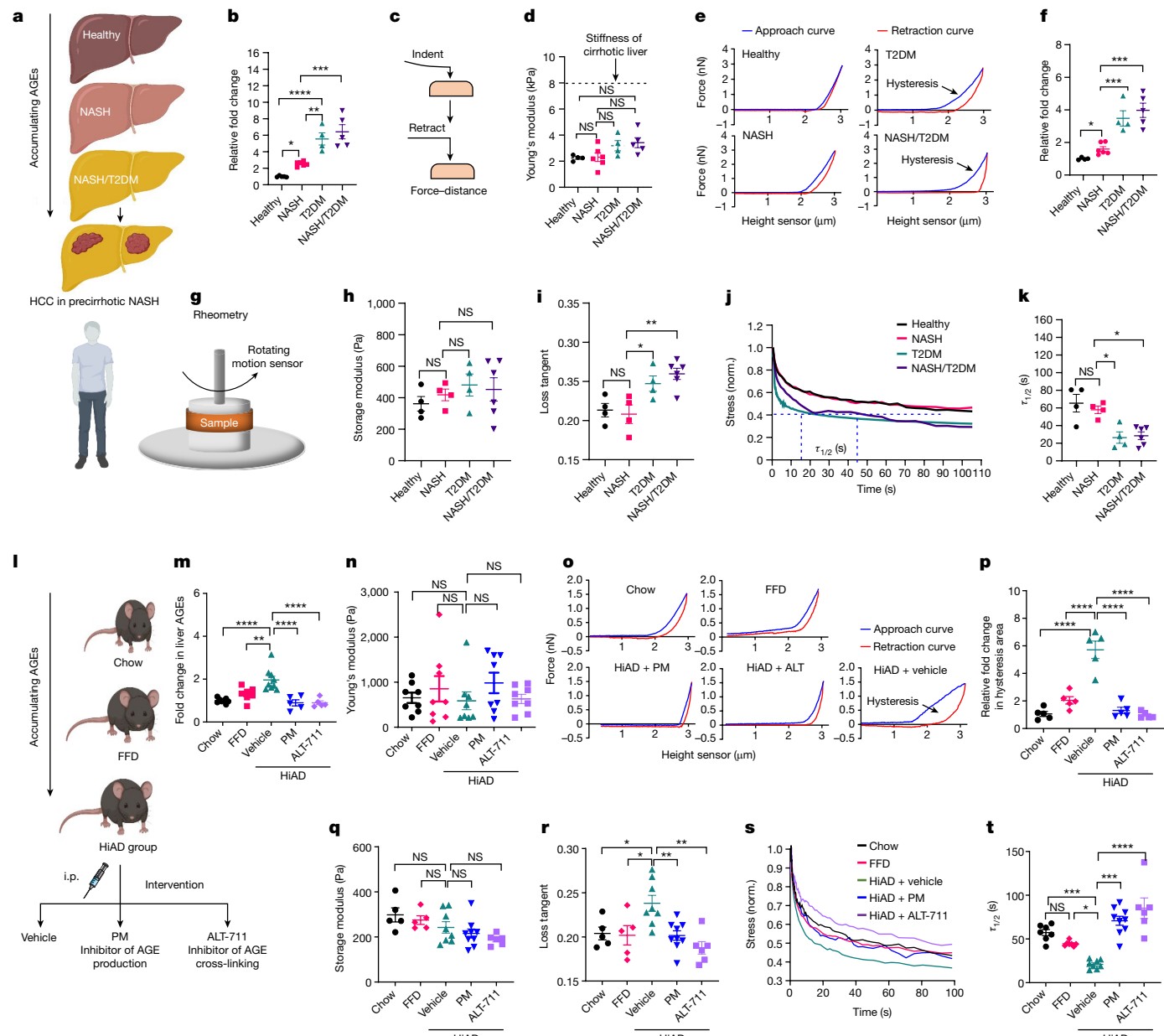

**Fig. 1 | Viscoelasticity is increased in the livers of individuals with NASH and T2DM and in mice on a HiAD. a**, Schematic of AGE increase in NASH/T2DM. **b–f. b**, Quantification of AGEs in the liver. **c**, Schematic of the AFM analysis. Indent-retract was used to generate force–distance curves. **d–f**, The stiffness (**d**; Young's modulus), representative force–distance curves (**e**) and quantification of hysteresis area (**f**, viscoelasticity) were assessed using AFM. The average stiffness of the human cirrhotic liver is indicated by a dashed line. $n = 4$ (healthy controls), $n = 6$ (patients with NASH), $n = 4$ (T2DM) and $n = 5$ (NASH + T2DM) individuals. **g–k**, Rheometry analysis of fresh liver tissues. **g**, Schematic of rheometry analysis of fresh liver tissues. **h**, Rheometry analysis of the storage modulus in precirrhotic livers. **i**, The loss tangent (viscoelasticity) in the livers of healthy participants, and individuals with NASH, T2DM or NASH/T2DM. **j**, Stress relaxation curves in liver samples from the different patient groups. Norm., normalized. **k**, Stress was normalized to the initial stress and depicted as $\tau_{1/2}$ (the timescale at which the stress is relaxed to half its original value). $n = 4$ (healthy controls), $n = 4$ (patients with NASH), $n = 4$ (T2DM) and $n = 6$ (NASH + T2DM) individuals. **l–p**, Mice were placed onto a chow or FFD diet, or a HiAD diet with daily vehicle, AGE inhibitor (PM) or AGE-cross-linking inhibitor (alagebrium, ALT-711) treatment. **l**, Schematic of the experiment. i.p., intraperitoneal. **m**, Quantification of liver AGEs (from left to right, $n = 8, 7, 9, 5$ and 5). **n–p**, Liver stiffness (**n**; from left to right, $n = 5, 5, 8, 9$ and 6 mice) and viscoelasticity (stress relaxation curves (**o**) and hysteresis area (**p**); $n = 5$ mice in each group) were assessed using AFM. **q–t**, Stiffness (**q**) and viscoelasticity (**r–t**) were assessed using rheometry. From left to right, $n = 5, 5, 8, 9$ and 6 (**q** and **r**) and $n = 7, 7, 8, 9$ and 6 (**t**) mice. The diagrams in **a** and **l** were created using BioRender. Data are mean ± s.e.m. $n$ values refer to individual mice. Statistical analysis was performed using one-way analysis of variance (ANOVA) followed by Tukey's multiple-comparison test (**b–q** and **t**) and two-sided Student's unpaired $t$-tests (**b**, **f** and **r**). NS, not significant; $*P < 0.05$, $**P < 0.01$, $***P < 0.001$, $****P < 0.0001$.

had similar stiffness (Fig. 1c,d) but higher hysteresis areas under loading–unloading cycles, corresponding to viscous energy dissipation or loss, indicating higher viscoelasticity (Fig. 1e,f). Furthermore, using rheometry, we confirmed that the livers of patients with NASH and T2DM without cirrhosis had similar storage moduli but a higher loss

tangent and faster stress relaxation under a constant deformation (Fig. 1g–k).

To investigate the link between AGEs and viscoelasticity, we studied a mouse model fed high-AGE diet (HiAD). These mice exhibit steatosis, hepatocyte ballooning, insulin resistance and higher liver AGEs

compared with those on regular (chow) or fast-food (FFD) diets[10] (Fig. 1l,m). On the basis of AFM (Fig. 1n–p) and rheometry (Fig. 1q–t) analyses, the stiffness was similar (Fig. 1n,q), but the viscoelasticity increased (Fig. 1o,p,r–t), in mice fed a HiAD compared with those fed chow or FFD. Notably, inhibiting AGE production with pyridoxamine (PM) or preventing the formation of AGE–collagen cross-links with alagebrium (ALT-711) reversed the changes in viscoelasticity (Fig. 1o,p,r–t). Together, these data indicate that AGEs in the precirrhotic ECM cause increased viscoelasticity.

## Viscoelasticity creates a tumorigenic niche

The above results prompted us to investigate the potential causal connections between ECM viscoelasticity and HCC progression. Studies on NASH-related HCC commonly used a high-fat diet combined with the carcinogen diethylnitrosamine, genetically modified models or a CDAA diet[18]. However, it is important to follow the typical pathogenesis characterized by cell injury and inflammation preceding HCC[19]. To experimentally imitate the conditions with increasing ECM stress relaxation before HCC onset, we used a more relevant model to modulate the metabolic/matrix milieu before tumour seeding. We hydrodynamically delivered human MET (hMET) with mutant β-catenin[20] or control vectors to chow-, FFD- or HiAD-fed mice. Earlier appearance and faster growth of transformed foci (glutamine synthetase (GS) and MYC-tag positive) were observed in HiAD-fed mice compared with in mice on the chow or FFD diet (Fig. 2a–c and Extended Data Fig. 1a–c). Importantly, tumour growth diminished, and the survival rate improved after AGE inhibition with PM or after breaking collagen–AGE cross-links with ALT (Fig. 2d–g and Extended Data Fig. 1d–f).

We next modulated AGE receptors using two additional approaches. We first addressed AGER1, the clearance receptor. As individuals with T2DM plus NASH and mice on a HiAD have significantly downregulated *AGER1* (also known as *DDOST*)[10], we sought to reverse this by AAV8-mediated AGER1 delivery before the hydrodynamic injection (HDI) (Fig. 2h). AGER1-reconstituted mice had higher levels of AGER1 (Fig. 2i) and lower levels of AGE receptors (Fig. 2j) owing to improved uptake, similar stiffness (Fig. 2k) but lower stress relaxation (Fig. 2l,m and Extended Data Fig. 2a), and reduced growth of transformed foci (Fig. 2n,o and Extended Data Fig. 2b), compared with mice that were injected with the control AAV8 construct.

Second, we studied mice with hepatocyte-specific deletion of *RAGE* (also known as *Ager*; proinflammatory AGE receptor; *RAGE^HepKO*) that were fed a HiAD, and hydrodynamically injected these mice with hMET with mutant β-catenin (Extended Data Fig. 2c). These mice had reduced growth of tumour foci (Extended Data Fig. 2d,e), decreased liver AGEs (Extended Data Fig. 2f), similar stiffness (Extended Data Fig. 2g) and reduced viscoelasticity (Extended Data Fig. 2h–j) compared with *RAGE^fl/fl* mice fed the HiAD.

Together, these studies with four animal models all converge on the conclusion that AGEs in the ECM create a more viscoelastic and tumorigenic environment that promotes HCC.

## Network connectivity and viscoelasticity

The data presented so far suggest that AGE accumulation is critical to increased ECM viscoelasticity and HCC progression. We next sought to elucidate how ECM viscoelasticity is modulated. Type-1 collagen networks are thought to be critical regulators of tissue mechanics[21]. AGEs can directly cross-link collagen and affect matrix mechanics by altering its helical structure[22,23]. Collagen networks were imaged using second harmonic generation (SHG) with two-photon microscopy in liver samples from humans (healthy control, NASH, NASH + T2DM; Extended Data Fig. 3a), as well as in mice (chow, HiAD, HiAD + PM, HiAD + ALT-711 and in AAV8-AGER1-injected livers; Extended Data Fig. 3b). The collagen network exhibited bundling in patients with T2DM + NASH and in mice

on the HiAD, whereas the network appeared to be more organized with thinner fibres in healthy humans, and in mice on the chow or HiAD diets after PM and ALT treatment or AGER1 reconstitution. Inhibiting AGE production or disrupting AGE–collagen cross-links decreased the amount of highly cross-linked insoluble collagen (Fig. 3a). We analysed decellularized native ECM from chow- or HiAD-fed mice using SHG to evaluate collagen fibre length, spatial orientation and interconnectivity. Compared with the well-organized collagen network in mice on the chow diet, in HiAD-fed mice, the network exhibited less connectivity and had collagen bundles with shorter fibres and lower fibre–fibre angles (Fig. 3b). Similar structures with shorter fibres and angles were seen in neutralized reconstituted collagen hydrogels with AGEs (Fig. 3c). The interactions between AGEs and collagen telopeptide lysine residues can interfere with fibre elongation and the formation of a structurally cohesive collagen network[24]. To evaluate the properties of AGE-cross-linked collagen hydrogels and the relationship between low-connectivity networks and matrix characteristics, we performed rheological measurements. AGE-cross-linked collagen hydrogels had similar stiffness (Fig. 3d) but faster stress relaxation compared with non-cross-linked collagen (Fig. 3e). We next measured the distribution of Young's moduli by AFM mapping of hydrogels. We saw that, while there was heterogeneity in local moduli in both gels, the ranges of elastic moduli were slightly higher in the collagen-only case (Extended Data Fig. 3c,d). The distribution of hysteresis areas (viscoelasticity) demonstrated higher frequencies in AGE-cross-linked collagen hydrogels (Extended Data Fig. 3e,f). To further study the collagen-rich matrix with stiffness comparable to that of the liver, interpenetrating polymer network (IPN) 3D hydrogels made from AGE-modified collagen and alginate were tested (Fig. 3f). Similar changes in architecture and connectivity were seen after AGE exposure (Fig. 3g). Breaking the AGE–collagen cross-links with ALT-711 decreased the amount of insoluble collagen (Fig. 3h) and increased the fibre length, fibre–fibre angles (Fig. 3i,j) and network connectivity. ALT-711 itself did not change the collagen matrix (Extended Data Fig. 3g) or its mechanical properties (Extended Data Fig. 3h–j). Indentation tests showed restored viscoelasticity after using ALT-711 (Fig. 3k). IPN hydrogels mixed with AGE-modified collagen exhibited similar stiffness (Fig. 3l), but faster stress relaxation compared with non-modified collagen (Fig. 3m–o) based on rheological measurements.

To further study how the matrix architecture impacts viscoelasticity, we simulated collagen matrices with a variation in two structural parameters to model the activity of AGEs: average collagen fibre bundle length and bundling angle, as a proxy for heterogeneity (Fig. 3p,q). The collagen matrices are connected by a combination of weak bonds, known to act within and between collagen fibres (Extended Data Fig. 4a–g) and could underlie stress relaxation and viscoelasticity in type-1 collagen matrices, and cross-links due to AGEs[25–27] (Extended Data Fig. 4h,j). Bundled networks are more heterogeneous and exhibit faster stress relaxation compared with unbundled networks (Fig. 3r). Reducing the fibre length (Fig. 3s and Extended Data Fig. 5a–e) or bundle angle (Fig. 3t and Extended Data Fig. 5f–i) increases the heterogeneity of the network leading to faster stress relaxation.

Together, these computational results show that the changes in architecture that we observed due to AGE activity in vivo and in vitro, namely enhanced heterogeneity and shorter fibre length, are predicted to lead to faster stress relaxation and enhanced viscoelasticity.

## Matrix viscoelasticity activates YAP through TNS1

We next focused on mechanotransductive pathways leading to tumour growth and invasion by faster stress relaxation. RNA-sequencing (RNA-seq) and bioinformatics analyses were performed on mouse liver samples (in the chow, FFD, HiAD, HiAD + PM and *RAGE^HepKO* groups), to identify potential pathways. Kyoto Encyclopedia of Genes and Genomes (KEGG) analyses were performed on differentially expressed genes

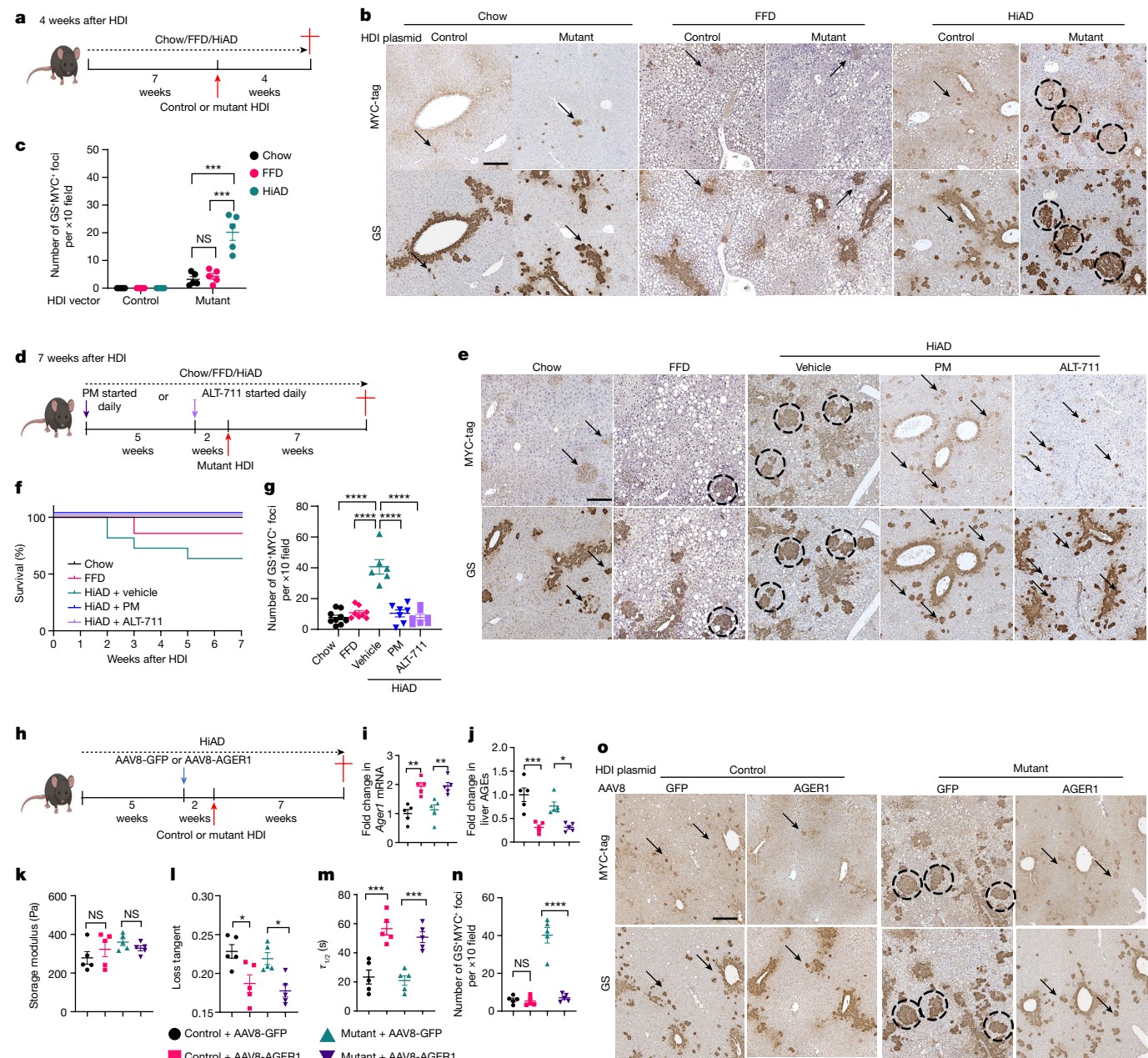

**Fig. 2 | Mice on a HiAD develop more transformed foci, and exhibit AGE-dependent higher viscoelasticity. a**, Schematic of the NASH-related HCC model (early timepoint). Mice were fed for 7 weeks with chow, FFD or HiAD. HDI was performed using vectors expressing human MET (pT3-EF5a-hMET) and sleeping beauty (SB) transposase combined with a vector expressing either wild-type (pT3-EF5a-β-catenin-MYC, control) or mutant (pT3-EF5a-S45Y-β-catenin-MYC) human β-catenin. Then, 4 weeks after HDI, the mice were euthanized. **b**, Immunohistochemistry on sequential slides for the co-localization of GS- and MYC-tag-positive foci (circles, more than 20 cells are considered to form a focus). Scattered positive cells denote transduced cells (arrows). GS at the baseline marks pericentral cells. **c**, Quantification of GS- and MYC-tag-positive foci 4 weeks after HDI. *n* = 5 each. **d**, Schematic of the NASH-related HCC model combined with AGE-lowering approaches; mice were euthanized 7 weeks after HDI. **e**, Immunohistochemistry on sequential slides showing the co-localization of GS- and MYC-tag-positive foci (circles). Scattered positive cells denote

transduced cells (arrows). **f**, The survival rates of mice after HDI. *n* = 7 (chow), *n* = 7 (FFD), *n* = 11 (HiAD + vehicle), *n* = 7 (HiAD + PM) and *n* = 7 (HiAD + ALT) mice. **g**, Quantification of transformed foci 7 weeks after HDI. Left to right, *n* = 8, 8, 6, 7 and 7, 20 areas per mouse. **h**, Schematic of the NASH-related HCC model combined with AAV8-mediated AGER1 induction before HDI. **i,j**, Liver *AGER1* (also known as *Ddost*) mRNA expression (**i**) and AGE levels (**j**) were assessed. *n* = 5 each. **k**–**m**, Rheometry data of studies on stiffness (**k**) and viscoelasticity (**l**, **m**) in AGER1-reconstituted mice. *n* = 5 each. **n**,**o**, Quantification of transformed foci (*n* = 5 each, 20 areas per mouse) (**n**) and GS and MYC immunohistochemistry analysis on sequential slides (**o**). Data are mean ± s.e.m. *n* values refer to individual mice. Statistical analysis was performed using one-way ANOVA followed by Tukey's multiple-comparison test. For **b**, **e** and **o**, scale bars, 300 μm. ALT, alagebrium. The diagrams in **a**, **d** and **h** were created using BioRender.

between the HiAD and FFD groups to categorize the enriched functional classification (Fig. 4a and Extended Data Fig. 6a). In the Hippo signalling pathway group, YAP/TAZ-regulated genes were enriched in HiAD-fed mice and decreased after PM treatment and in *RAGE^HepKO*

mice (Fig. 4a). YAP activation, as indicated by a nuclear signal of active non-phosphorylated YAP (Extended Data Fig. 6b,c,g) and the induction of targets CTGF (also known as CCN2) and CYR61 (CCN1), was observed in mice on the HiAD, but decreased after PM treatment or in

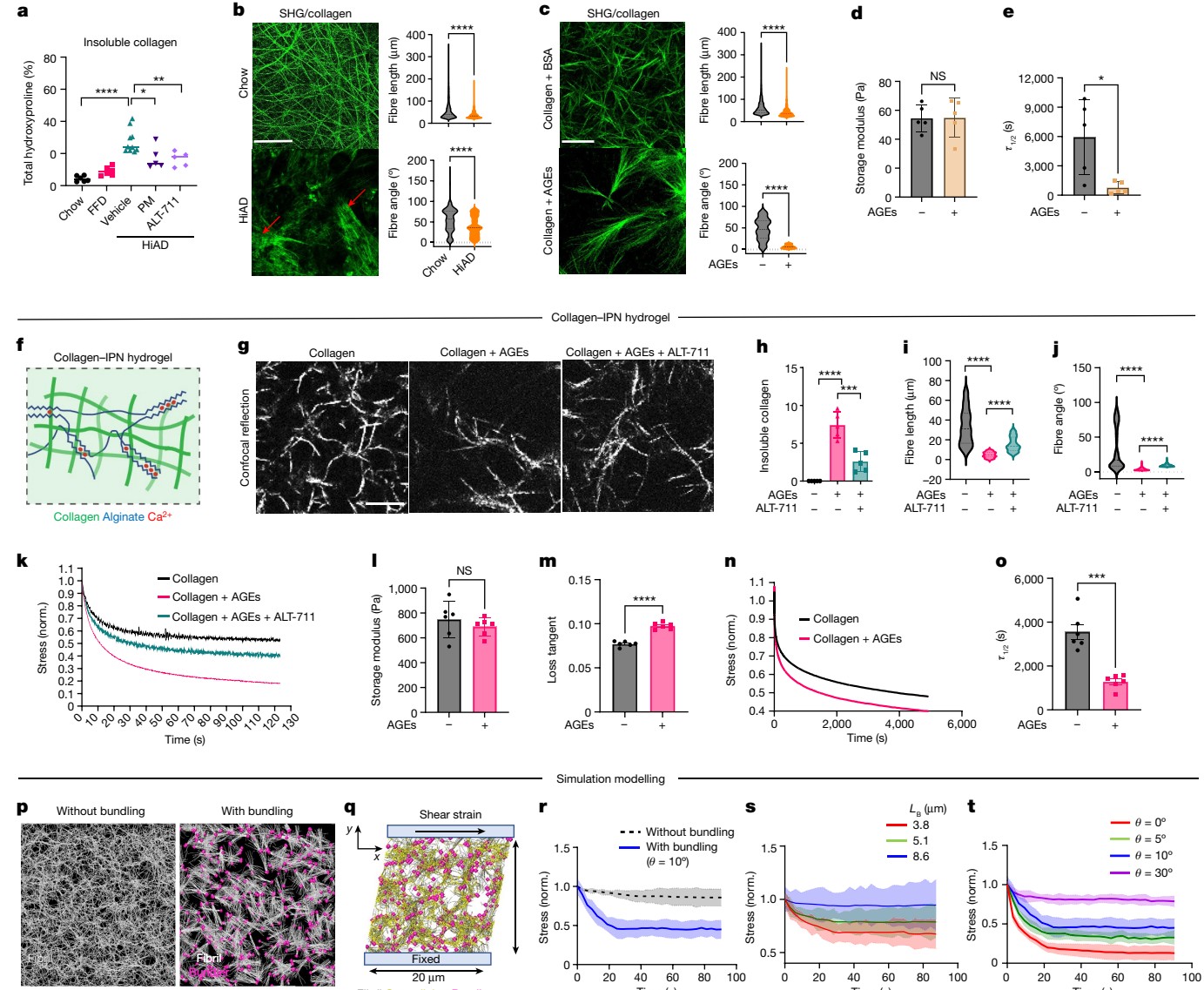

**Fig. 3 | AGEs modulate collagen architecture and network connectivity, leading to enhanced viscoelasticity in livers and 3D hydrogels.**
**a**, Quantification of insoluble collagen (from left to right, $n = 6, 6, 10, 5$ and $5$). **b**,**c**, SHG microscopy images of decellularized liver ECMs (**b**) (bundles, red arrows) and collagen hydrogels (**c**). Fibre lengths (CT-Fire) and fibre–fibre angles (ImageJ) were assessed (three mice or gels per group, five images per mouse or gel). **d**,**e**, Collagen hydrogels from **c** were assessed using rheometry for stiffness (**d**) and viscoelasticity (**e**), expressed as $\tau_{1/2}$ (the timescale at which stress is relaxed to half its original value). $n = 5$. **f**, Schematic of the collagen/IPN hydrogel. The diagram was adapted from ref. 16, under a Creative Commons licence CC BY 4.0. **g**, Confocal reflectance microscopy of collagen fibres after AGE ± ALT-711 exposure. **h**–**j**, Insoluble collagen (**h**; $n = 5$), fibre lengths (**i**; CT-Fire) and angles (**j**; ImageJ) in the IPN gels. Data are from three images per gel, three gels per group. **k**, Representative stress relaxation curves from IPN gels.

**l**–**o**. Rheometry of IPN gels testing stiffness (**l**; storage modulus) and viscoelasticity (**m**–**o**), showing the loss tangent (**m**) and stress relaxation (stress relaxation curves (**n**) and $\tau_{1/2}$ (**o**)) are shown. $n = 6$. **p**, Simulation modelling. A matrix structure consisting of individual fibrils (3 µm length) without (left) or with (right) bundlers connecting the ends of fibrils at $\theta = 10°$. **q**, After a matrix is assembled, it is deformed by 20% shear strain. **r**, Stress relaxation is measured using the two matrices shown in **p**. **s**,**t**, Stress relaxation was studied using matrices with different bundle length ($L_B$) with $\theta = 0°$ (**s**) or different bundling angle ($\theta$) with $L_B = 3$ µm (**t**). Data are mean ± s.e.m. (**a**–**o**). $n$ values refer to independent experiments. Statistical analysis was performed using Wilcoxon's rank-sum tests (**b** and **c**), two-tailed unpaired $t$-tests (**d**, **e** and **l**–**o**), Kruskal–Wallis tests with Dunn's test (**i** and **j**) and one-way ANOVA followed by Tukey's multiple-comparison test (**h**). For **r**–**t**, data are mean ± s.d. ($n = 4$ each). For **b**, **c** and **g**, scale bars, 100 µm.

the *RAGE^HepKO* mice on the HiAD (Extended Data Fig. 6d). YAP-positive hepatocytes were not observed in close proximity to collagen bundles in the HiAD group (Extended Data Fig. 6e). To substantiate that HCC in faster-stress-relaxing ECM was promoted by YAP and TAZ, we co-injected dominant-negative TEA domain transcription factor 2 (dn-TEAD2) or control plasmids by HDI with hMET and mutant β-catenin (Fig. 4b). The TEAD family of transcription factors are important binding partners for YAP and TAZ, and TEAD2 in particular was linked to HCC progression[28]. We found that the number of GS/MYC double-positive

foci significantly decreased after dn-TEAD2/HDI co-injection compared with in mice that were injected with the control plasmid (Fig. 4c,d and Extended Data Fig. 6f (haematoxylin and eosin images)), and these foci were negative for active, nuclear YAP (Fig. 4c (bottom, inset)). In mice treated with HDI + control vector co-injection, active YAP localized to the nuclei of GS-positive cells. Downstream targets CTGF and CYR61 were downregulated after injection with dn-TEAD2 (Fig. 4e). Together, these data indicate that HCC in faster-stress-relaxing ECM was promoted by YAP[28].

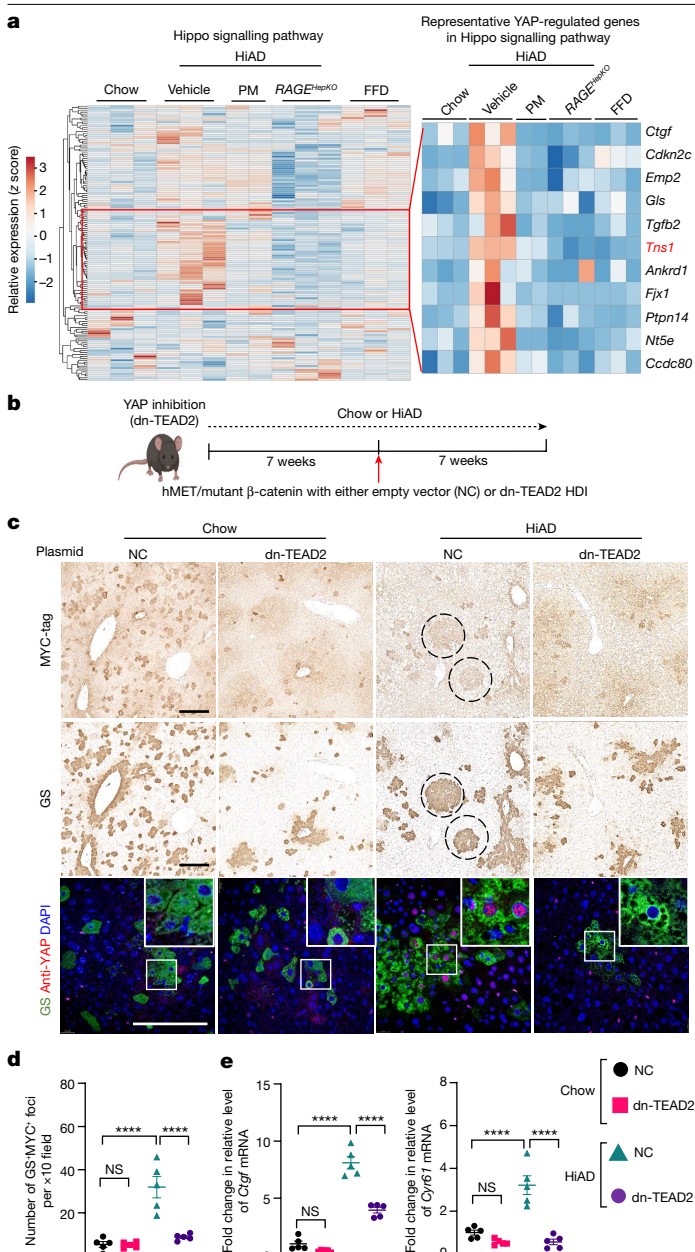

**a**, Analyses of bulk RNA-seq data from mice fed a chow or FFD diet or a HiAD diet with vehicle treatment ($n = 3$ each), mice fed a HiAD with PM treatment ($n = 2$) and $RAGE^{HepKO}$ mice fed a HiAD ($n = 3$). The heat map shows enrichment in several YAP/TAZ target genes in the HiAD group compared with in the other groups. Tensin1 (*Tns1*, red) target of interest. **b**, Schematic of the NASH-related HCC model combined with YAP inhibition. The diagram was created using BioRender. **c**, GS/MYC-tag immunohistochemistry on subsequent slides showing transformed foci (circles, top and middle). Active non-phosphorylated YAP (red) localized to the nuclei of GS positive cells (green) in control-vector-injected (for dn-TEAD2) mice fed a HiAD (bottom). Scale bars, 300 μm (top and middle) and 100 μm (bottom). NC, empty vector. **d**, Quantification of GS⁺MYC⁺ foci in **c**. $n = 5$. **e**, YAP targets *Ctgf* and *Cyr61* were downregulated in dn-TEAD2-treated mice. $n = 5$. Data are mean ± s.e.m. *n* values refer to individual mice. Statistical analysis was performed using one-way ANOVA followed by Tukey's multiple-comparison test.

Given these results, our next question was how the viscoelastic ECM conveys proliferative and invasive signals. To directly test the effects of viscoelasticity, we constructed IPNs of alginate and reconstituted basement membrane (rBM) matrix hydrogels for 3D culture studies (rBM-IPNs). HCC cells (Fig. 5 and Extended Data Fig. 7a–e) were encapsulated in rBM-IPN hydrogels[13,16] with the same stiffness (Young's modulus, 2 kPa) but different levels of viscoelasticity (Fig. 5a). In high-viscoelasticity gels, cells exhibited increased proliferation (Fig. 5b,e), YAP activation (Fig. 5c,f) and the formation of TKS5-positive (also known as SH3PXD2A) and MT1-MMP-positive invadopodia-like structures as well as a more elongated shape (Fig. 5d,g). When the Young's modulus was gradually increased to 5 kPa, we did not observe an increase in cell proliferation, YAP activation or induction of its target genes (Extended Data Fig. 7a–e). Invadopodia are thin linear protrusions that are linked to cancer cell invasion and migration. In high-viscoelasticity gels, we detected canonical markers such as TKS5 and MT1-MMP (Fig. 5d and Extended Data Fig. 7f) as well as active integrin β1, and phosphorylated myosin light chain (p-MLC) (Extended Data Fig. 7g,h, respectively). Antibody controls are shown in Extended Data Fig. 7i. As integrin-β1–matrix interactions modulating mechanotransduction were reported in breast cancer cells in high-viscoelasticity gels[13,16], we used integrin-β1-specific blocking antibodies (Fig. 5h–j). These reduced proliferation (Fig. 5h) and YAP target CTGF (Fig. 5i) and increased cell circularity (Fig. 5j).

Notably, on the basis of our RNA-seq data, *TNS1* was significantly induced in an AGE- and viscoelasticity-dependent manner (Figs. 4a and 5k). TNS1 was described as part of a molecular clutch that mediates the mechanical linkage between ECM-bound integrins and the actomyosin cytoskeleton to generate traction force[29–33]. To study the role in mechanosignalling, proximity ligation assays (PLA) were performed in HCC cells in low- versus high-viscoelasticity hydrogels to capture the potential interaction between TNS1 and integrin β1. High-viscoelasticity conditions promoted their interaction (Fig. 5l,m and Extended Data Fig. 7j (antibody controls)). We next studied the effects of *TNS1* or integrin β1 knockdown (CRISPR–Cas9; Fig. 5n–p). This reduced cell proliferation (Fig. 5n and Extended Data Fig. 8a) as well as the activation of YAP and target genes (Fig. 5o and Extended Data Fig. 8b,e) in high-viscoelasticity ECM. Invadopodia formation was reduced, and cell circularity increased after *TNS1* or integrin β1 knockdown (Fig. 5p and Extended Data Fig. 8c,d). Hep3B cells exhibited similar results (Extended Data Fig. 9). To further analyse the importance of the TNS1–integrin β1 interaction in invadopodia formation and proliferation, we transfected cells with a vector containing a mutant *TNS1* (with a deletion of the PTB domain preventing its binding to integrin β1, hereafter, dd-TNS1) or full-length *TNS1* (Extended Data Fig. 10a) and embedded them in low- or high-viscoelasticity hydrogels. We assessed whether dd-TNS1 is capable of binding to integrin β1 using a PLA. In dd-TNS1-transfected cells, the PLA signal was very low, whereas, in the full-TNS1–tdTomato-positive cells, the signals co-localized (Extended Data Fig. 10b,c). dd-TNS1-transfected cells in high-viscoelasticity hydro-gels had lower proliferation (Extended Data Fig. 10d,h), and reduced active YAP and its target genes (Extended Data Fig. 10e,i,k), while their circularity increased (Extended Data Fig. 10f,j) and they did not form protrusions (Extended Data Fig. 10g). As RhoA is involved in regulating cell shape and migration, and it was described to be modulated by TNS1[34], we next studied its activity. Cells in high-viscoelasticity hydrogels exhibited increased RhoA-GTPase activity in a TNS1- and integrin-β1-dependent manner (Extended Data Fig. 11a,b). LATS1[35] activity decreased and YAP was more active in cells in high-viscoelasticity gels, whereas the reverse was seen after *TNS1* or integrin β1 knockdown. Pharmacological inhibition of ROCK-GTPase and myosin 2 resulted in an increase in cell circularity, a lower proliferation rate and reduced expression of YAP targets (Extended Data Fig. 11c–e).

To study the in vivo role of TNS1, we used a CRISPR–Cas9-directed approach whereby TNS1 was targeted at the time of HDI. Mice on chow or HiAD diets were hydrodynamically injected as before, including either single-guide RNA for *Tns1* (sgTNS1) or control sgRNA plasmids, and were euthanized 7 weeks after the injections (Extended Data Fig. 12a). In the sgTNS1-treated mice, *Tns1* expression and the

**Fig. 4 | YAP is involved in HCC growth promoted by high viscoelasticity.**

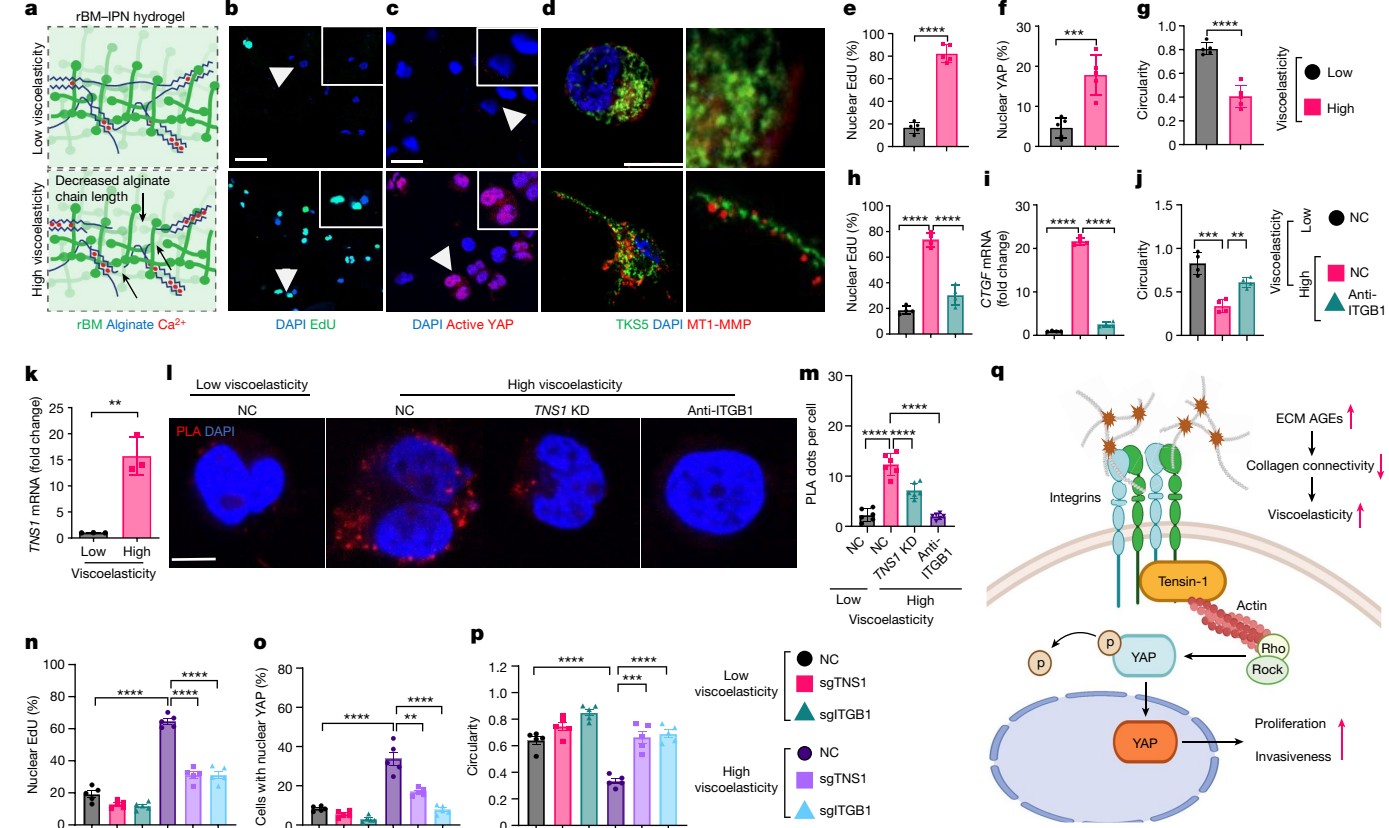

**Fig. 5 | The integrin-β1–TNS1–YAP axis mediates viscoelasticity-specific mechanocellular pathways for HCC cell invasion. a**, Schematics of the tuneable viscoelasticity IPNs of alginate (blue) and reconstituted basement membrane matrix (green) 3D hydrogels. Lowering the molecular mass of alginate cross-linked by calcium (red) decreases the network connectivity (arrows) to increase viscoelasticity. The diagram was adapted from ref. 16, under a Creative Commons licence CC BY 4.0. **b**–**g**, Cell proliferation in low- or high-viscoelasticity hydrogels was analysed using EdU assays (imaging (**b**) and quantification (**e**)). **c**,**f**, YAP activation was analysed using antibodies against active YAP (imaging (**c**) and quantification (**f**); the arrowheads denote enlarged areas). **d**, The formation of invadopodia-like structures after transfection (Clontech-N1, containing human TKS5-mNeonGreen), and immunofluorescence analysis of MT1-MMP (red) using Airyscan microscopy. **g**, Cell circularity analyses (ImageJ; *n* = 5 gels). Scale bars, 50 μm (**b**), 20 μm (**c**) and 10 μm (**d**). **h**–**j**. Huh7 cells in low- or high-viscoelasticity hydrogels were incubated with control

IgG or integrin β1 (ITGB1) blocking antibodies. Cell proliferation (**h**; EdU), YAP target *CTGF* mRNA (**i**) and cell circularity (**j**, *n* = 4) were analysed. **k**, *TNS1* mRNA expression in cells in low- or high-viscoelasticity hydrogels. *n* = 3. **l**,**m**, PLAs depict direct binding between TNS1 and integrin β1 (ITGB1) in high-viscoelasticity hydrogels (**l**; scale bar, 10 μm). **m**, The PLA signal was analysed (30 cells in 5 gels per group, *n* = 5, each). **n**–**p**, Huh7-Cas9 cells were transfected with plasmids containing CRISPR guide RNA for *TNS1* (sgTNS1), integrin β1 (sgITGB1) or control sgRNA (NC) and embedded in low- or high-viscoelasticity hydrogels. The proliferation (**n**), YAP activation (**o**) and cell circularity (**p**) were analysed. *n* = 5 each. **q**, Schematic of TNS1, which functions as a key component of the ECM mechanosensor complex by binding to integrin β1 in high-viscoelasticity ECM. The diagram was created using BioRender. Data are mean ± s.e.m. *n* values refer to independent experiments. Statistical analysis was performed using two-tailed unpaired *t*-tests (**e**–**g** and **k**) and one-way ANOVA followed by Tukey's multiple-comparison test (**h**–**j** and **n**–**p**).

number of transformed MYC⁺GS⁺ foci decreased compared with in the control-sgRNA-injected mice (Extended Data Fig. 12b–d, respectively), and YAP targets *Ctgf* (also known as *Ccn2*) and *Cyr61* (*Ccn1*) had lower expression (Extended Data Fig. 12e). Together, these data suggest that TNS1 is part of a molecular clutch that mediates the mechanical linkage between ECM-bound integrins and the actomyosin cytoskeleton responding to high viscoelasticity (Fig. 5q).

## Discussion

Here we show that AGEs in the liver ECM create a faster stress-relaxing viscoelastic niche leading to the activation of mechanosignalling promoting HCC. Although matrix stiffness in advanced fibrosis/cirrhosis and its effects on cancer progression have been extensively studied, our data are the first, to our knowledge, to demonstrate how changes in matrix viscoelasticity, independent of stiffness, affect HCC growth. This is clinically very relevant as increasing viscoelasticity could be a risk factor foretelling more invasive features of HCC in NASH/T2DM. Current guidelines exclude precirrhotic patients from HCC screening

paradigms; therefore, new viscoelasticity-based imaging approaches will need to be developed to identify the population at risk. T2DM is a major risk factor not only for liver cancer progression but also in breast[36], colon[37] and pancreatic cancer[38–40], and this may point to the crucial role of a more viscoelastic matrix in patients with diabetes.

Collagen cross-linking mediated by AGEs has been thought to contribute to increasing stiffness similarly to LOXl2 or TTG, modifying the helical structure of collagen[41–44]. However, our studies identified that, in certain contexts, increased collagen cross-linking leads to weakly bound shorter fibres with lower interconnectivity, together promoting a viscoelastic niche. This in turn facilitates changes in cellular shape, cytoskeletal reorganization and the formation of invadopodia-like structures[12,13,15,16]. This could be relevant to not only the liver peritumoral matrix, but also other cancers in which changes in viscoelasticity have been observed, but are not yet implicated functionally for progression[45,46].

We also identified the major role of the TNS1–integrin β1–RhoA–YAP mechanotransduction pathway responding to increasing viscoelasticity. While the role of YAP activation in various cancers is known, as is its

role in responding to viscoelasticity in vitro[13–15,17,47,48], the in vivo role of YAP as an inducer of cancer progression in response to viscoelasticity changes was previously unclear. Furthermore, another element of interest relates to the role of TNS1 as a key mechanosensory in this integrin–YAP signalling pathway. TNS1 regulating and sustaining the activity of the molecular clutch could be essential for protrusion formation and migratory activity of cells[29,30], and these can be further explored in future studies.

In conclusion, we identified the central role of increasing viscoelasticity in the liver tumour niche. Viscoelasticity-activated mechanocellular pathways promise to be a diagnostic and/or therapeutic areas in NASH/T2DM-related HCC and beyond.

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

## Methods

### Human liver tissues

All human samples were de-identified and exempted (exemption 4). Human liver samples were obtained from the Stanford Diabetes Research Center (SDRC), Donor Network West (DNW), Stanford Tissue Bank and Clinical Biospecimen Repository and Processing Core (CBRPC) of the Pittsburgh Liver Research Center (PLRC). This was approved by Stanford University Institutional Review Board (IRB, 67378) and the Pittsburgh Liver Research Center Review Board. Histology was evaluated for necroinflammation, hepatocellular ballooning, ductular reaction and fibrosis by a hepatopathologist in a blinded manner, and non-alcoholic fatty liver disease activity score (NAS) scores were provided. Detailed information about the donors, including sex, age, diagnosis and NAS score, are summarized in Supplementary Table 1.

### Animal studies

All animal experiments were conducted according to the experimental procedures approved by the Institutional Animal Care and Use Committee at Stanford University and Palo Alto VA. Mice were maintained at temperatures and humidity ranges of 20 to 26 °C and 30% to 70%, respectively. Mice were housed in standard cages under 12 h–12 h light/dark cycles with ad libitum access to water and food unless otherwise indicated. Mice were placed on control chow or a FFD (17.4% protein, 20% fat and 49.9% carbohydrate, AIN-76A) supplemented with high-fructose corn syrup in the drinking water at a final concentration of 42 g l$^{-1}$ for up to 14 weeks, or HiAD (prepared by cooking FFD at 120 °C for 20 min) as previously described[49] supplemented with high-fructose water. Wild-type C57BL/6J male mice (aged 8 to 10 weeks) were purchased from the Jackson Laboratory. *RAGE$^{fl/fl}$* mice on the C57B6 background were gifted by B. Arnold. *RAGE$^{HepKO}$* mice were generated by crossing *RAGE$^{fl/fl}$* mice with *Albumin-cre* mice (Jackson Laboratory) for several generations. Male *RAGE$^{fl/fl}$* mice and *RAGE$^{HepKO}$* mice aged 8 to 10 weeks were used. To generate hepatocyte transgenic AGER1 (*Ddost*) mice, wild-type mice were injected with adeno-associated virus 8 (AAV8)-control green fluorescent protein (AAV8-control) or AAV8-thyroxine-binding globulin-AGER1 recombinase (AAV8-AGER1) ($5 \times 10^{11}$ genome copies, Vector BioLabs) at the end of week 5 of feeding (the rationale of the specific targeting in hepatocyte has been previously reported)[50]. To study the inhibition of AGE formation or AGE–protein cross-linking, wild-type mice fed with HiAD were injected intraperitoneally daily with pyridoxamine hydrochloride (Sigma-Aldrich) (60 mg per kg, daily)[51], alagebrium (ALT-711) (Sigma-Aldrich, 1 mg per kg)[52,53] or vehicle (Tris-HCl), from the fifth week of feeding. At the time of euthanasia, photographs of the livers were taken, and the number and the size of lesions on the surface were recorded.

### Hydrodynamic tail vein injection and plasmid preparation

Oncogenic plasmids were delivered to the mouse liver by hydrodynamic tail vein injection as previously described[20]. In brief, pT3-EF1α-h-hMET, pT3-EF1αh-WT-β-catenin-MYC-tag, pT3-EF1α-h-S45Y-mutant-β-catenin-MYC-tag and pCMV/SB transposase were from S. P. S. Monga's laboratory. Plasmids used for in vivo experiments were purified using the Invitrogen Endotoxin-free Maxi prep kit (Sigma-Aldrich). A combination of 20 µg of pT3-EF5-hMET and 20 µg of pT3-EF5α-WT-β-catenin-MYC or pT3-EF5α-S45Y-β-catenin-MYC, along with 1.6 µg of SB (25:1) was diluted in 1 ml of endotoxin-free saline (AdipoGen), filtered through 0.22 mm filter (Millipore), and injected into the tail vein of mice in 5 to 7 s at the at the end of week 7 of chow, FFD or HiAD feeding. To inhibit YAP activity, the mice were injected with dn-TEAD2 (pT3-EF1α-h-dnTEAD2, 60 µg)[54] or negative control vector (empty pT3-EF1α, 60 µg), (from S. P. S. Monga's laboratory) by hydrodynamic tail vein injection at the end of week 7 of chow or HiAD feeding with hMET, S45Y-mutant-β-catenin and SB. To knockdown

*TNS1*, mice were injected with a CRISPR–Cas9-based vector linking two sgRNAs targeting *Tns1* exon 1 and exon 7 (pX333-TNS1, 50 µg) or negative control vector (empty pX333, 50 µg) by hydrodynamic tail vein injection at the end of seventh week of chow or HiAD feeding at the same time as with hMET, S45Y-mutant-β-catenin and SB. The sequences of the sgRNAs used in this study are provided in Supplementary Table 2.

Mice were monitored 3 days per week for any sign of pain or distress, and the body weight and body condition score were evaluated after hydrodynamic tail vein injection. The mice were immediately euthanized if they lost more than 10% of their body weight or if they had a body condition score of 2 or less during that time. The tumour size (macroscopic) never reached more than 3 mm.

### Histology, immunohistochemistry and early focus quantification

Paraffin-embedded tissue samples were cut into 5 µm sections, deparaffinized and rehydrated. For antigen retrieval, slides were boiled in citrate buffer (0.01 M, pH 6.0) using a microwave oven on high power for 5 min and cooled down to room temperature. After incubation in 3% aqueous $H_2O_2$ to quench the endogenous peroxidase, the sections were washed in PBST (PBS with 0.1% Tween-20, v/v) washing buffer, blocked with 5% goat serum (EMD Millipore) diluted in PBST at room temperature for 1 h and incubated with primary antibodies diluted in 2% goat serum in PBST at 4 °C overnight.

For immunohistochemistry, slides were incubated with appropriate biotinylated secondary antibodies (Supplementary Table 3) for 1 h, and then processed according to the ABC Peroxidase Standard Staining Kit (Thermo Fisher Scientific) for 30 min. The slides were stained with 3,3'-diaminobenzidine (Abcam) for 5 s to 5 min and counterstained with haematoxylin (Thermo Fisher Scientific) for 45 s. The images were scanned using the Leica Aperio AT2 system at Stanford Human Pathology/Histology Service Center. Serial sections were incubated with GS and MYC-tag primary antibodies, and a cluster of cells (at least 20 cells) positive both for GS and MYC-tag were counted as early foci. The antibodies used in this study are shown in Supplementary Table 3.

### Measurement of AGE content

AGEs were measured using the OxiSelect Advanced Glycation End Product Competitive ELISA Kit (Cell Biolabs) in the serum and the liver homogenate according to the manufacturer's instructions. In brief, 10 mg liver samples were homogenized in PBS. After measuring the protein concentration, 300 µg protein was added to a 96-well ELISA plate and incubated for 1 h at room temperature. After incubation with the secondary HRP-conjugated anti-AGE antibodies, the reaction was halted with a stop solution, and the plates were read at 450 nm.

### RNA extraction, reverse transcription and RT–qPCR

According to the manufacturer's recommendations, total RNA was extracted from snap-frozen liver tissues and cells using the RNeasy mini kit (Qiagen). Complementary cDNA was created from an identical amount of RNA using the iScript cDNA synthesis kit (Bio-Rad). The PowerUp SYBR Green PCR Master Mix (Applied Biosystems) was used for quantitative PCR with reverse transcription (RT–qPCR) on the 7900HT machine (Applied Biosystems), and the data were evaluated using the $2^{-Ct}$ technique. As an endogenous control, *Arbp* (also known as *Rplp0*) and *GAPDH* were used to standardize the data in mouse and human, respectively. A list of the primer sequences used in this study is provided in Supplementary Table 4.

### AFM

Measurements on frozen liver tissues were performed on livers embedded in OCT compound (Sakura), snap frozen by direct immersion into liquid nitrogen and cut into 100 µm sections on the Leica CM1900-13 cryostat. The samples were kept in a protease inhibitor cocktail during the AFM analyses (Roche Diagnostic). For the hydrogel mapping

collagen or AGE-modified collagen was neutralized with 5× DMEM and 1× NaOH and the hydrogels were kept overnight on 37 °C. The Bruker Resolve BioAFM system was used to take measurements (Bruker). For the indentation, Novascan Tech modified silicon nitride cantilevers ($k = 0.01$ N m$^{-1}$, $k = 0.06$ N m$^{-1}$) with a borosilicate glass spherical tip (diameter 10 and 5 μm, respectively) were used.

For each session, cantilevers were calibrated using the thermal oscillation method. AFM force maps were performed on 94.7 μm × 94.7 μm fields. Each experimental group included at least four different human or mouse samples, with two sections from each, and three different areas generated per section. For hydrogel mapping, each group included at least four different gels, and five different areas generated per gel. Data analyses were performed using the Hertz model in NanoScope Analysis v.1.9. and Mountains SPIP v.9.

## Rheometry analysis of the human and mouse livers

Measurements were optimized to assess storage modulus, loss modulus, loss tangent and stress relaxation based on methods previously described[55]. In brief, liver samples were prepared using an 8 mm diameter punch (Integra Miltex). The height of the slices ranged from 3 to 5 mm in the uncompressed state. The samples were kept hydrated during all of the experiments with DMEM. Parallel plate shear rheometry was performed on the ARES-G2 rheometer (TA instruments) at room temperature using TA TRIOS software v.5.1.1 (TA instruments). For all of the measurements, the upper plate was initially lowered to touch the sample, and 0.01 N of nominal initial force (~300 Pa) was applied to ensure adhesive contact of the sample with the plates. Measurements were taken first with a dynamic time sweep test (2% constant strain, oscillation frequency 1 radian s$^{-1}$, measurements taken for 600 s), then stress relaxation (10% initial strain, measurements taken for 600 s).

## Measurement of total and insoluble collagen

The hydroxyproline assay for total liver collagen was performed as previously described[56]. In brief, liver samples were homogenized and denatured in 6 N HCl. Hydrolysed samples were then dried and washed three times with deionized water, followed by incubation in 50 mM chloramine T oxidation buffer for 20 min at room temperature. The samples were incubated with 3.15 M perchloric acid (Sigma-Aldrich) for 5 min, then with $p$-dimethylaminobenzaldehyde (Sigma-Aldrich). The absorbance of each sample was measured.

For insoluble collagen, the liver or collagen gel was first homogenized in 0.5 M acetic acid at a 1:4 ratio (for example, 800 μl for 200 mg liver) to make 20% liver homogenate. Next, 500 μl of 20% liver homogenate was added onto 1 ml of 0.5 M acetic acid and the tubes rotated at 4 °C overnight. The samples were centrifuged at 20,000$g$ for 30 min to collect the pellet, resuspended in pepsin (2 mg ml$^{-1}$ in 0.5 M acetic acid) and incubated at 4 °C overnight. The next day, the samples were centrifuged and the pellets were collected. The samples were then analysed using the hydroxyproline assay, described previously[56].

## Preparation of AGE-BSA

AGE-BSA was prepared as previously described[57,58]. In brief, glycolaldehyde (Sigma-Aldrich) was dissolved in 10 mg ml$^{-1}$ BSA/PBS to a final concentration of 33 mM. The solutions were incubated at 37 °C for 72 h, followed by dialysis against PBS. The dialysed solutions were sterilized with 2 μM filters, and aliquots were stored at −80 °C.

## Preparation of AGE-modified collagen and gels

Collagen type I (telopeptide intact, Corning, 354236, Corning collagen I, rat tail, used in all experiments) was incubated with 2.5 mg ml$^{-1}$ in AGE-BSA 0.1% acetic acid (Merck) to obtain 3 mg ml$^{-1}$ collagen solution at 4 °C for 4 weeks. BSA was mixed with collagen as the control. Alagebrium chloride (ALT-711, 20 mg ml$^{-1}$) was added as the AGE cross-linking inhibitor. Collagen gels were polymerized by mixing 3 mg ml$^{-1}$ AGE-modified or non-modified collagen solution with 10×

PBS and neutralized with 1× NaOH (Merck) and incubated at 37 °C for 90 min leading to the formation of gels. For most gels, 1.6 mg ml$^{-1}$ collagen was used.

## Alginate preparation

According to the manufacturer, low-molecular-mass, ultrapure sodium alginate (Provona UP VLVG, NovaMatrix) with a molecular mass of <75 kDa was used for fast-relaxing substrates. For slow-relaxing substrates, sodium alginate (high molecular mass) was used (FMC Biopolymer, Protanal LF 20/40, high molecular mass, 280 kDa). Alginate was treated with activated charcoal, dialysed against deionized water for 3–4 days (molecular mass cut-off, 3,500 Da), sterile-filtered, lyophilized and then reconstituted to 3.5 wt% in serum-free DMEM (Gibco). The use of low/high-molecular-mass alginate resulted in high/low viscoelasticity IPNs.

## Imaging of collagen fibrils

Mouse livers were decellularized in situ by detergent (0.5% (w/v), sodium deoxycholate, 250 ml per mouse) and water (50 ml per mouse) perfusion at a pump speed of 0.2 ml min$^{-1}$. After the final perfusion, the livers were removed and washed overnight in PBS. For AGE-modified collagen gels, the samples were prepared as previously described. Gels were imaged 1 day after formation.

For SHG imaging, all of the samples were imaged using the Leica TCS SP5 multiphoton confocal microscope or the Leica Stellaris 8 DIVE upright confocal microscope. The excitation wavelength was tuned to 840 nm, and a 420 ± 5 nm narrow band-pass emission controlled by a slit was used for detecting the SHG signal of collagen. The images were recorded using an inverted confocal laser-scanning microscope (Leica TCS SP8) equipped with a ×20 water-immersion objective for confocal reflection imaging. An Ar$^+$ laser at 488 nm was used to illuminate the sample, and the reflected light was detected with photomultiplier tube (PMT) detectors. Scans were at 1,024 × 1,024 pixels, and all of the images were taken 80–100 μm into the samples. Collagen measurements were performed using CT-Fire software (v.2.0 beta) (https://loci.wisc.edu/software/ctfire) and ImageJ v.1.53t (https://imagej-nih-gov.stanford.idm.oclc.org/ij/).

## IPN 3D hydrogel formation

Alginate was transferred to a 1.5 ml Eppendorf tube (a polymer tube) and kept on ice for each viscoelastic gel. For rBM-IPNs, alginate was mixed 30 times before adding rBM (Corning) at 4 °C. Collagen-IPNs were created by diluting and neutralizing AGE-modified or AGE-unmodified collagen gels with 10× DMEM and 1 M NaOH (Merck) at 4 °C. All substrates had a final concentration of 10 mg ml$^{-1}$ alginate, 4.4 mg ml$^{-1}$ rBM or 1.6 mg ml$^{-1}$ collagen after additional DMEM was added. This was pipette-mixed, and the resulting mixture was kept on ice. Calcium sulfate was added to 1 ml Luer lock syringes (Cole-Parmer) and stored on ice to maintain the constant Young's moduli of the substrates with high and low viscoelasticity. The polymer mixes were divided into individual 1 ml Luer lock syringes (polymer syringes) and placed onto ice as well. The polymer syringe was linked to the calcium sulfate syringe to create gels. The two solutions were quickly combined using 30 pumps on the syringe handles, and the resulting mixture was placed into a well of an eight-well Lab-Tek plate (Thermo Fisher Scientific) that had been precoated with rBM. After moving the Lab-Tek dish to a 37 °C incubator, the gel was allowed to form for 1 h before a full medium was added.

## Mechanical characterization of IPNs

Rheology experiments were performed using a stress controlled AR2000EX rheometer (TA Instruments). IPNs were directly deposited onto the lower Peltier plate for rheology testing. The gel was then slowly contacted by a 25 mm flat plate, creating a 25 mm disk gel. To stop dehydration, mineral oil (Sigma-Aldrich) was applied to the gel's

edges. The storage and loss moduli had equilibrated by the time the time sweep was done, which was at 1 rad s$^{-1}$, 37 °C and 1% strain.

For the stress relaxation experiments, after the time sweep, a constant strain of 5% was applied to the gel at 37 °C, and the resulting stress was recorded over the course of 4 h.

## Cell culture, transfection and CRISPR–Cas9-mediated *TNS1* and integrin β1 knockdown

Human HCC cell lines Huh7 (gift from P. Sarnow) and Hep3B (purchased from ATCC) were cultured in high-glucose DMEM (Gibco) with 10% fetal bovine serum (FBS, Gibco) with 1% penicillin–streptomycin (Life Technologies). All cells were cultured at 37 °C in 5% $CO_2$.

*TNS1* was knocked down in Huh7 and Hep3B cells using *TNS1* sgRNAs or integrin β1 sgRNAs. A control sgRNA sequence was used as the negative control. sgRNAs were cloned into pMCB306 (Addgen plasmid 89360, sgRNA expression vector with GFP, puromycin resistance), then co-transfected into cells with lentiCas9-Blast (Addgen plasmid 52962, expresses human codon-optimized *Streptococcus pyogenes* Cas9 protein and blasticidin resistance from the EFS promoter). Transfected cells were selected by puromycin and tested for TNS1 or integrin β1 expression 2 days after transfection. A list of the sequences of the sgRNAs used in this study is provided in Supplementary Table 2.

For PTB-domain-deleted TNS1 (dd-TNS1) assays, dd-TNS1 or a full-length TNS1 was cloned into a tdTomato-N1 vector (gift from S.-H. Lo). Huh7 cells were transfected with these plasmids and cultured for 24 h before encapsulation in IPNs hydrogels.

Huh7 or Hep3B cells were transfected with plasmids containing the human *TKS5* (also known as *SH3PXD2A*) with either mNeonGreen or mScarlet tag (gift from L. Hodgson) for all invadopodia analyses.

## 3D cell encapsulation in IPNs

For analyses of YAP activation, invadopodia formation and proliferation, Huh7 cells were serum-starved overnight and encapsulated in IPNs. In brief, cells were washed with PBS, trypsinized using 0.05% trypsin/EDTA, washed once, centrifuged and resuspended in serum-free medium. The concentration of cells was determined using the Vi-Cell Coulter counter (Beckman Coulter). After Matrigel was mixed with the alginate, cells were added to this polymer mixture and deposited into a cooled syringe. The solution was then vigorously mixed with a solution containing $CaSO_4$ and deposited into wells of a chambered cover glass (LabTek). The final concentration of Matrigel and alginate was 4.4 mg ml$^{-1}$ and 10 mg ml$^{-1}$, respectively. 5 mM, 15 mM and 50 mM $CaSO_4$ were used to generate varied stiffness of IPNs with 0.8 kPa, 2 kPa and 5 kPa, respectively. The final concentration of cells was $3 \times 10^6$ cells per ml of IPN. The cell-laden hydrogels were gelled in an incubator at 37 °C and 5% $CO_2$ for 60 min and then were cultured in a DMEM medium containing 10% FBS. After 1 day, cells were collected for RT–qPCR, western blotting and immunostaining analysis.

## Inhibitors

Inhibitors in the 3D cell culture were used at the following concentrations: 10 μM Y-27632 to inhibit ROCK (Abcam); 50 μM blebbistatin (Abcam) to inhibit myosin II; and 1 μg ml$^{-1}$ monoclonal integrin-β1-blocking antibody (Abcam, P5D2). Vehicle-alone controls for these inhibitors were as follows: DMSO for blebbistatin, and latrunculin-a; deionized water for Y-27632; and IgG nonspecific antibody (Sigma-Aldrich, I5381) for integrin-β1-blocking antibody. Y-27632 and blebbistatin were added to the culture medium directly. Integrin β1-blocking antibody was incubated with Huh7 cells on ice for 1 h before encapsulation in IPNs and added to the culture medium directly.

## RNA-seq, bioinformatics and KEGG analyses

*RAGE$^{fl/fl}$* (wild-type) mice were fed chow, FFD or HiAD for 14 weeks. A group of HiAD-fed mice was injected intraperitoneally daily with PM. A cohort of *RAGE$^{HepKO}$* mice was placed on a HiAD for 14 weeks as described previously[10]. RNA was prepared from 2–3 mice per group, and RNA-seq was performed at Novogene with paired-end 150 bp reads (NovaSeq 6000 Sequencing System, Illumina). Gencode gene annotations version M18 and the mouse reference genome major release GRCm38 was derived from https://www.gencodegenes.org/. Dropseq tools v.1.1249 were used for mapping the raw sequencing data to the reference genome. The resulting UMI-filtered count matrix was imported into R v.3.4.4. Before differential expression analysis using Limma v.3.40.650, sample-specific weights were estimated and used as coefficients alongside the experimental groups as a covariate during model fitting with Voom. *t*-tests were used for determining differentially ($P < 0.05$) regulated genes between all possible experimental groups. GSEA was conducted using the preranked GSEA method within the KEGG databases with the online tool g:Profiler (https://biit.cs.ut.ee/gprofiler/gost). RNA-seq heat maps and unsupervised hierarchical clustering was performed with g:Profiler. RNA-seq data are available under Gene Expression Omnibus accession number GSE245016.

## Protein extraction and western blotting

Cells were washed with PBS and lysed with RIPA buffer. The homogenate was centrifuged, and the supernatant was collected. Protein concentrations were determined using the Pierce BCA Protein Assay Kit (Thermo Fisher Scientific). Protease inhibitor (Roche) and phosphatase inhibitor (Roche) were added to all the lysis procedures mentioned above, and 10–50 μg of the protein samples were loaded onto the SDS–polyacrylamide gel. The proteins were transferred to a polyvinylidene difluoride membrane or nitrocellulose membrane, which was blocked with 5% BSA in TBST and then incubated with primary antibodies at 4 °C overnight. The blots were washed with TBST and further incubated with horseradish-peroxidase-conjugated secondary antibodies. The signal was detected by adding Western-Bright enhanced chemiluminescence substrate (Advansta) or SuperSignal West Pico PLUS Chemiluminescent Substrate (Thermo Fisher Scientific) and imaged with film or the iBright CL1500 imaging system (Thermo Fisher Scientific). The images were processed and analysed using NIH ImageJ (v.1.53t) and iBright Analysis software v.5.2.1 (Thermo Fisher Scientific). The antibodies used in this study were shown in Supplementary Table 3.

## Fluorescence immunostaining and microscopy

Frozen sections of livers or gel-embedded cells were washed twice with PBS and fixed with 4% paraformaldehyde at 4 °C overnight. Sections were permeabilized in PBS with 0.4% (v/v) Triton X-100 for 10 min. After blocking with 5% goat serum in PBST at room temperature for 1 h, cells were incubated with primary antibodies (Supplementary Table 3) diluted in 2% goat serum in PBST at 4 °C overnight. The slides were washed and then incubated with secondary antibodies at room temperature for 1 h. Coverslips were washed with PBST between incubations and mounted with an anti-fade mounting medium with DAPI. Fluorescence images were taken using the Leica TCS SP8, multi-photon Leica Stellaris 8 DIVE upright Confocal, and Zeiss Airyscan2 LSM980 systems. Images were processed using NIH ImageJ (v.1.53t). To quantify cell circularity and cell area, the confocal images of cells were analysed in ImageJ v.1.53t (https://imagej-nih-gov.stanford.idm.oclc.org/ij/). Circularity, mathematically calculated as $4\pi \times$ area $\times$ (perimeter)$^{-2}$, ranges from 0 to 1, with a value of 1 being a perfect circle.

## PLA

The Duolink proximity ligation assay kit (Sigma-Aldrich) was used to determine the interaction between TNS1 and integrin β1 in Huh7 cells. Reagents were used according to the manufacturer's instructions, and the steps were optimized. In brief, anti-TSN1 and anti-integrin β1 were used as the primary antibodies. The primary antibodies bound to a pair of oligonucleotide-labelled secondary antibodies (PLA probes), the hybridizing connector oligos joined the PLA probes if they were close

by and the ligase created the DNA template needed for rolling-circle amplification (RCA). Labelled oligos hybridized to the complementary sequences in the amplicon and produced discrete red fluorescent signals that could be seen by confocal microscopy (Leica Microsystems). NIH Image J (v.1.53t) software was used to count the signal, and the average counts were used for the plot.

## Simulation modelling

In this study, we used an agent-based model for simulating the discrete structure of collagen matrices. Details of the model and all of the parameter values used in the model are provided in the Supplementary Methods and Supplementary Table 5. The computational domain is rectangular with $20 \times 20 \times 5\ \mu m$ in the $x, y$ and $z$ directions. A periodic boundary condition exists only in the $x$ and $z$ directions. In simulations, the motions of the cylindrical elements are updated at each time step through the Langevin equation and the Euler integration scheme. Three types of matrices were used in this study: fibrillar matrix; long, tight-bundle matrix; and short, loose-bundle matrix. Fibril assembly is initiated by the nucleation of seed fibrils through the appearance of one cylindrical segment in random positions, followed by elongation up to either $3\ \mu m$ or $5\ \mu m$ through the addition of segments without consideration of depolymerization. Effective collagen concentration calculated using the specific volume of collagen ($0.73\ ml\ g^{-1}$) is $3.65\ mg\ ml^{-1}$. The fibrillar matrix is constructed with cross-linkers in the absence of bundlers (Extended Data Fig. 4d,h). During the assembly of the fibrillar matrix, individual fibrils are interconnected by cross-linkers of which two binding sites bind to any part of two fibrils. There is no preference for a cross-linking angle in this process. The long, tight bundle matrix is assembled with cross-linkers in the presence of bundlers that connect individual fibrils in a parallel, staggering manner into tight bundles (Extended Data Fig. 4g,i). Cross-linkers connect fibrils within each bundle or fibrils belonging to different bundles. The short, loose-bundle matrix is created by bundlers that connect fibrils at their ends with a specific angle (Extended Data Fig. 4e,f,j). While the bundlers are permanently bound to fibrils, the cross-linkers can unbind from fibrils at a rate that exponentially increases with an increasing force, following Bell's law. Fibrils are permanently bound to two boundaries normal to the $y$ direction (that is, $+y$ boundary located at $y = 20\ \mu m$ and $-y$ boundary located at $y = 0\ \mu m$). After completion of matrix assembly, 20% strain is applied to the $+y$ boundary in the $x$ direction, whereas the $-y$ boundary is fixed. After reaching the 20% strain, the strain is held at a constant level to measure stress relaxation.

## Statistical analyses

At least three biological replicates were performed for all in vivo experiments, and in vitro experiments were repeated at least three times. Data are presented as mean ± s.e.m. Statistical analyses were performed using GraphPad Prism v.10.0.3 (GraphPad Software). Normality distribution was assessed using the Kolmogorov–Smirnov test. Two-tailed unpaired $t$-tests and one-way ANOVA with Tukey tests were used to analyse data with a normal distribution. Data with a non-normal distribution were analysed using Wilcoxon rank-sum tests and Kruskal–Wallis tests with Dunn's test. $P < 0.05$ was considered to be statistically significant.

## Reporting summary

Further information on research design is available in the Nature Portfolio Reporting Summary linked to this article.

## Data availability

Data have been deposited in public databases. RNA-seq data are available under the accession number GSE245016. The mouse reference genome major release GRCm38 is available online (https://www.gencodegenes.org/). Source data are provided with this paper.

## Code availability

Codes used in the study are available at GitHub (https://github.com/ktyman2/liverCancer.git).

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

**Acknowledgements** Patient samples were obtained using the services of the Diabetes Clinical and Translational Core facility of the Stanford Diabetes Research Center, which is supported by the National Institute of Diabetes and Digestive and Kidney Diseases of the National Institutes of Health under Award Number P30DK116074. We thank the members of the Donor Network West for their cooperation and all of the organ and tissue donors and their families for giving the gift of life and the gift of knowledge through their donation. Patient samples were obtained using the services of the Clinical Biospecimen Repository and Processing Core of the Pittsburgh Liver Research Center (PLRC), which is supported by PLRC grant number P30 DK120531. We thank S. Pneh, J. Zhang, R. Reguram, H. Park and P. Buiga for their technical support; M. Walkiewicz for the AFM; and J. Mulholland, K. Lee and G. X. Wang for assistance with imaging. Part of this work was performed at the Stanford Nano Shared Facilities (SNSF), supported by the National Science Foundation under award ECCS-2026822. Part of this work was performed at Stanford Cell Sciences Imaging Facility (CSIF, RRID:SCR_017787), supported by award numbers 1S10OD010580-01A1, S10RR02557401 and 1S10OD032300-01 from the National Center for Research Resources (NCRR). Its contents are solely the responsibility of the authors and do not necessarily represent the official views of the NCRR or the National Institutes of Health. This research was supported by funding NIH grants R01DK083283, RO1CA277710 and 1RO1AG060726 (to N.J.T.), SPARK Award (Stanford University to N.J.T.), VA I01 BX002418 (to N.J.T.) and NCI R37 CA214136 (to O.C.). The study was also funded in part by NIH grants 1R01CA251155 and 1R01CA204586 to S.P.M. R35GM136226 to L.H. and by 1R01GM126256 to O.C. and T.K.

**Author contributions** W.F. and N.J.T. conceived and designed the study. W.F. performed most of the experiments. W.F. and K.A. established and measured collagen-AGEs-IPN hydrogel and IPN 3D cell culture system. W.F. and L.V. performed mechanical measurements of human, mouse and hydrogel samples. W.F., L.V., Yuan Li., K.K., D.C., G.M. and Y.W. performed the mouse experiments. M.F.R. and T.K. invented and performed the computer simulations. D.K.-C.C., J.T. and E.G.E. helped to establish the CRISPR–Cas9 plasmid and HDI delivery system. W.F., N.A., R.D., A.S., D.G., G.W.C., S.P.M. and N.J.T. performed the human liver sample collection and pathological diagnosis. S.H.L. provided the TNS1 plasmids and critical comments. L.H. provided TKS5 plasmids and critical comments. V.C. helped with the statistical analyses. W.F., L.V. and Yuan Li created the figures. W.F. and N.J.T. wrote the manuscript with input from T.K., O.C., R.G.W. and R.L.B. Funding was acquired by S.P.M., T.K., O.C. and N.J.T. All of the authors provided advice, and read and approved the final manuscript.

**Competing interests** W.F., N.J.T. and O.C. are listed as inventors on a patent (WO/2023/102546) related to this this research. The other authors declare no competing interests.

**Additional information**
**Correspondence and requests for materials** should be addressed to Natalie J. Török.

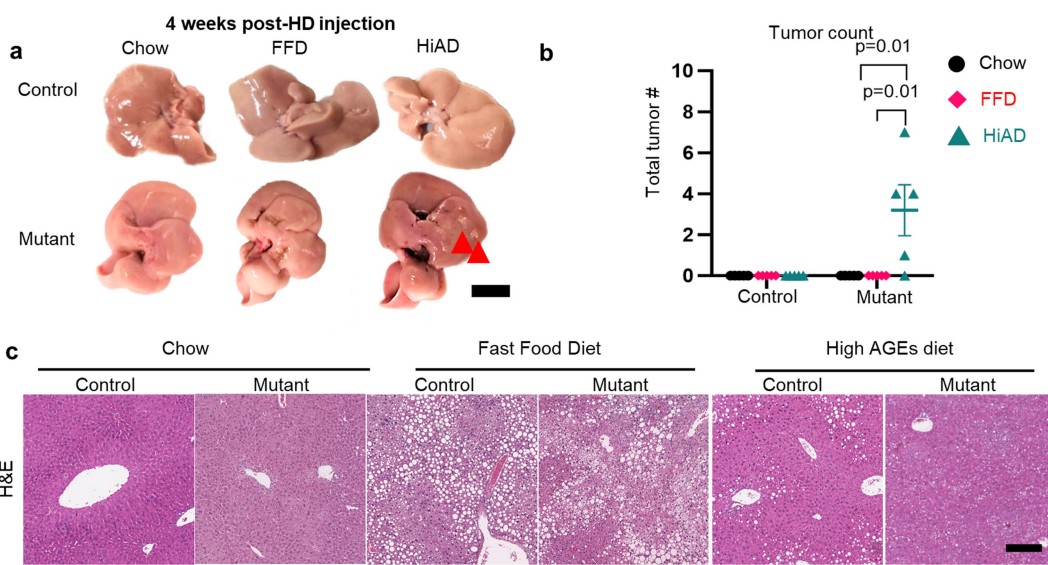

**4 weeks post-HD injection**

**a**

| | Chow | FFD | HiAD |
|---|---|---|---|
| Control | | | |
| Mutant | | | |

**b** Tumor count

**c**

| Chow | Fast Food Diet | High AGEs diet |
|---|---|---|
| Control / Mutant | Control / Mutant | Control / Mutant |

H&E

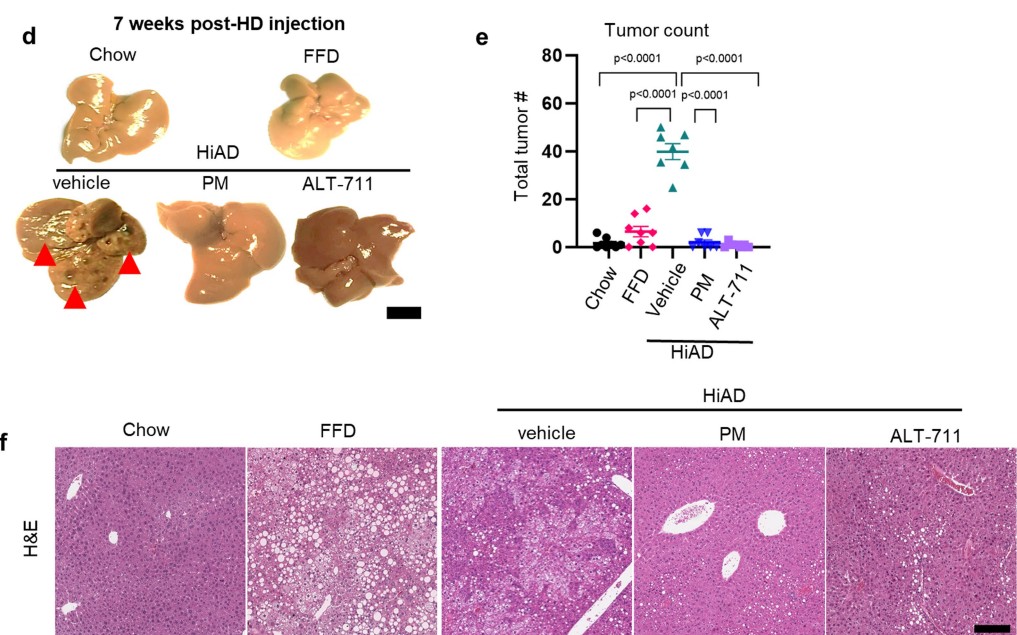

**d** **7 weeks post-HD injection**

Chow    FFD

HiAD
vehicle    PM    ALT-711

**e** Tumor count

**f**

| Chow | FFD | HiAD vehicle | PM | ALT-711 |
|---|---|---|---|---|

H&E

**Extended Data Fig. 1 | Mice on HiAD develop earlier and more numerous transformed foci following hydrodynamic injection, and exhibit AGEs-dependent higher viscoelasticity (related to the main Fig. 2). a-c.** Additional data of the early time point (4w. post-HDI) NASH/HCC model. Mice were fed for 7 weeks either chow, FFD or HiAD. Hydrodynamic injection (HDI) was performed using vectors expressing human MET gene (pT3-EF5a-hMet) and the sleeping beauty (SB) transposase combined with a vector expressing either wild-type human β-catenin (pT3-EF5a-β-catenin-myc, control group) or mutant pT3-EF5a-S45Y-β-catenin-myc, mutant group). Four weeks after HDI, mice were sacrificed. **a.** Livers from chow, FFD and HiAD-fed mice after HDI. Arrowheads, small lesions. Scale bar, 1 cm. **b.** Quantification of visible liver lesions 4 weeks after HDI (n = 5). **c.** Hematoxylin and eosin (H&E) on sequential slides corresponding to the main Fig. 2b. Scale bar, 300 μm. **d-f.** Additional data on the model 7 weeks post-HD injection. Representative images (**d**) and quantification (**e**) of visible liver lesions (n = 8, 8, 7, 7, 7 respectively). Arrowheads, lesions. Scale bar, 1 cm. **f.** H&E images, corresponding to the GS/myc images in the main Fig. 2e. Scale bar, 300 μm. Error bars represent mean ± s.e.m. n numbers refer to individual mice. One-way ANOVA was used followed by Tukey's multiple comparison test.

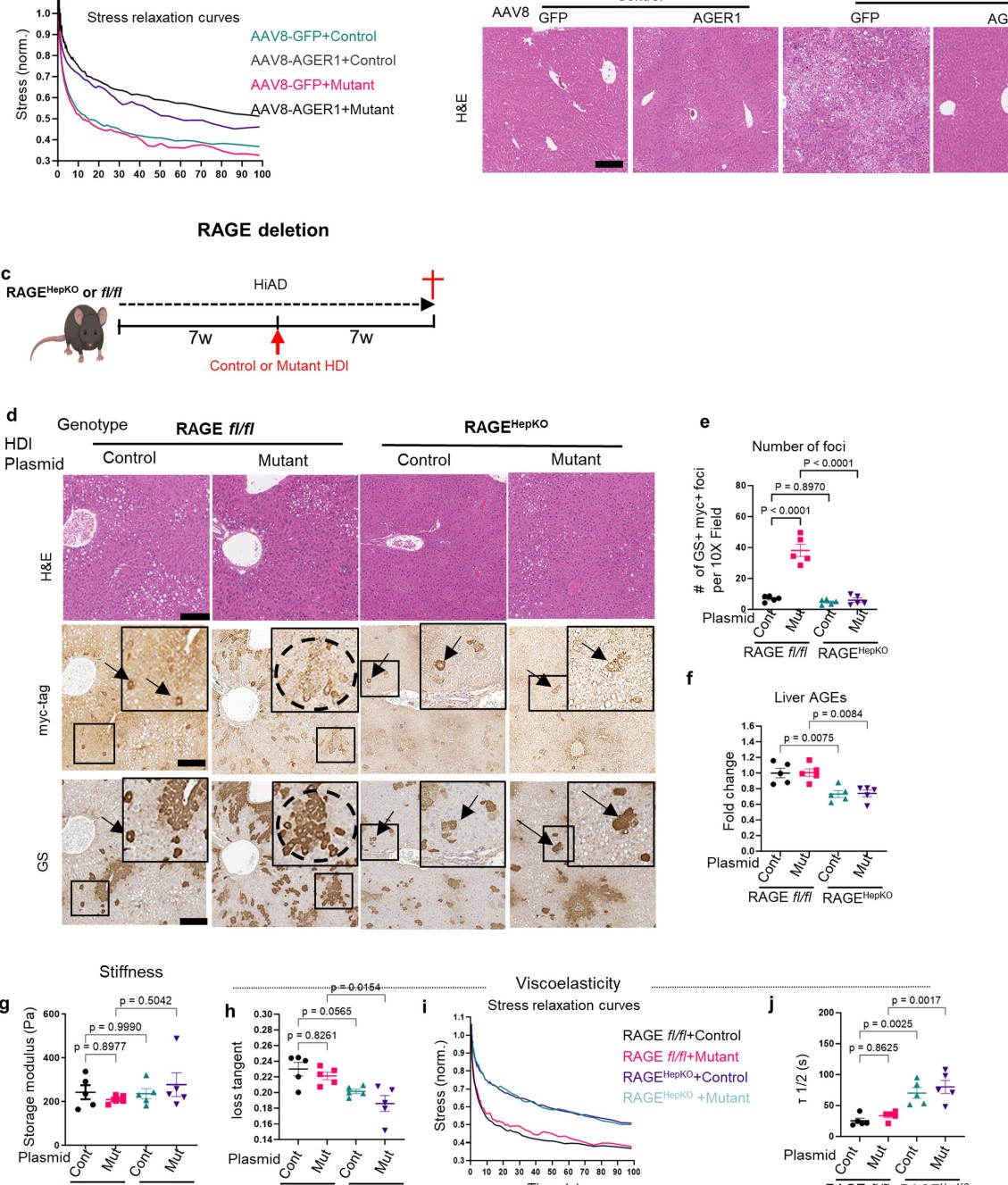

**AGER1 induction:** Extended data related to main Fig.2h-o

**a** Stress relaxation curves

- AAV8-GFP+Control
- AAV8-AGER1+Control
- AAV8-GFP+Mutant
- AAV8-AGER1+Mutant

**b** Plasmid: Control / Mutant; AAV8: GFP / AGER1; H&E

**RAGE deletion**

**c** RAGE$^{HepKO}$ or *fl/fl*, HiAD, 7w, Control or Mutant HDI, 7w

**d** Genotype: RAGE *fl/fl* (Control, Mutant), RAGE$^{HepKO}$ (Control, Mutant); HDI Plasmid; H&E, myc-tag, GS

**e** Number of foci; # of GS+ myc+ foci per 10X Field; P = 0.8970; P < 0.0001; P < 0.0001; Plasmid Cont/Mut RAGE *fl/fl* RAGE$^{HepKO}$

**f** Liver AGEs; Fold change; p = 0.0075; p = 0.0084; Plasmid Cont/Mut RAGE *fl/fl* RAGE$^{HepKO}$

**g** Stiffness; Storage modulus (Pa); p = 0.8977; p = 0.9990; p = 0.5042; Plasmid Cont/Mut RAGE *fl/fl* RAGE$^{HepKO}$

**h** loss tangent; p = 0.0154; p = 0.8261; p = 0.0565; Plasmid Cont/Mut RAGE *fl/fl* RAGE$^{HepKO}$

**i** Viscoelasticity; Stress relaxation curves; Stress (norm.); Time (s)
- RAGE *fl/fl*+Control
- RAGE *fl/fl*+Mutant
- RAGE$^{HepKO}$+Control
- RAGE$^{HepKO}$ +Mutant

**j** τ 1/2 (s); p = 0.8625; p = 0.0025; p = 0.0017; Plasmid Cont/Mut RAGE *fl/fl* RAGE$^{HepKO}$

**Extended Data Fig. 2 | AGER1 induction or RAGE deletion in hepatocytes reverses fast stress relaxation, and the appearance of transformed foci. a-b**. Additional data of NASH-related HCC model combined with AAV8-mediated AGER1 induction. Representative stress relaxation curves (**a**) with or without AGER1 induction. H&E images (**b**), corresponding to the GS/myc-tag images in the main Fig. 2o. Scale bar, 300 μm. **c-j**. Additional data of the NASH-related HCC model in the hepatocyte-specific RAGE deleted (RAGE$^{HepKO}$) mice. Schematics of the HDI using hMet/SB transposase with wild-type (control) or mutant β-catenin in RAGE$^{HepKO}$ or *fl/fl* mice (**c**). **d**. H&E and GS/myc immunohistochemistry were performed on sequential slides, circles represent foci, arrows indicate transduced cells. Scale bar, 300 μm. **e**. Quantification of foci (n = 5 each, 20 areas/per mouse). **f**. Liver AGEs were lower in RAGE$^{HepKO}$ in mice (n = 5 each). Rheometry studies, **g-j**. Liver stiffness (**g**) and viscoelasticity (**h-j**) were assessed (τ1/2 represents timescales at which the stress is relaxed to half its original value; increase in τ1/2 denotes lower viscoelasticity,**j**). There was no significant difference in stiffness, but improved viscoelasticity in RAGE$^{HepKO}$ mice (n = 5 each). Error bars represent mean ± s.e.m. n refers to individual mice. One-way ANOVA was used followed by Tukey's multiple comparison test.

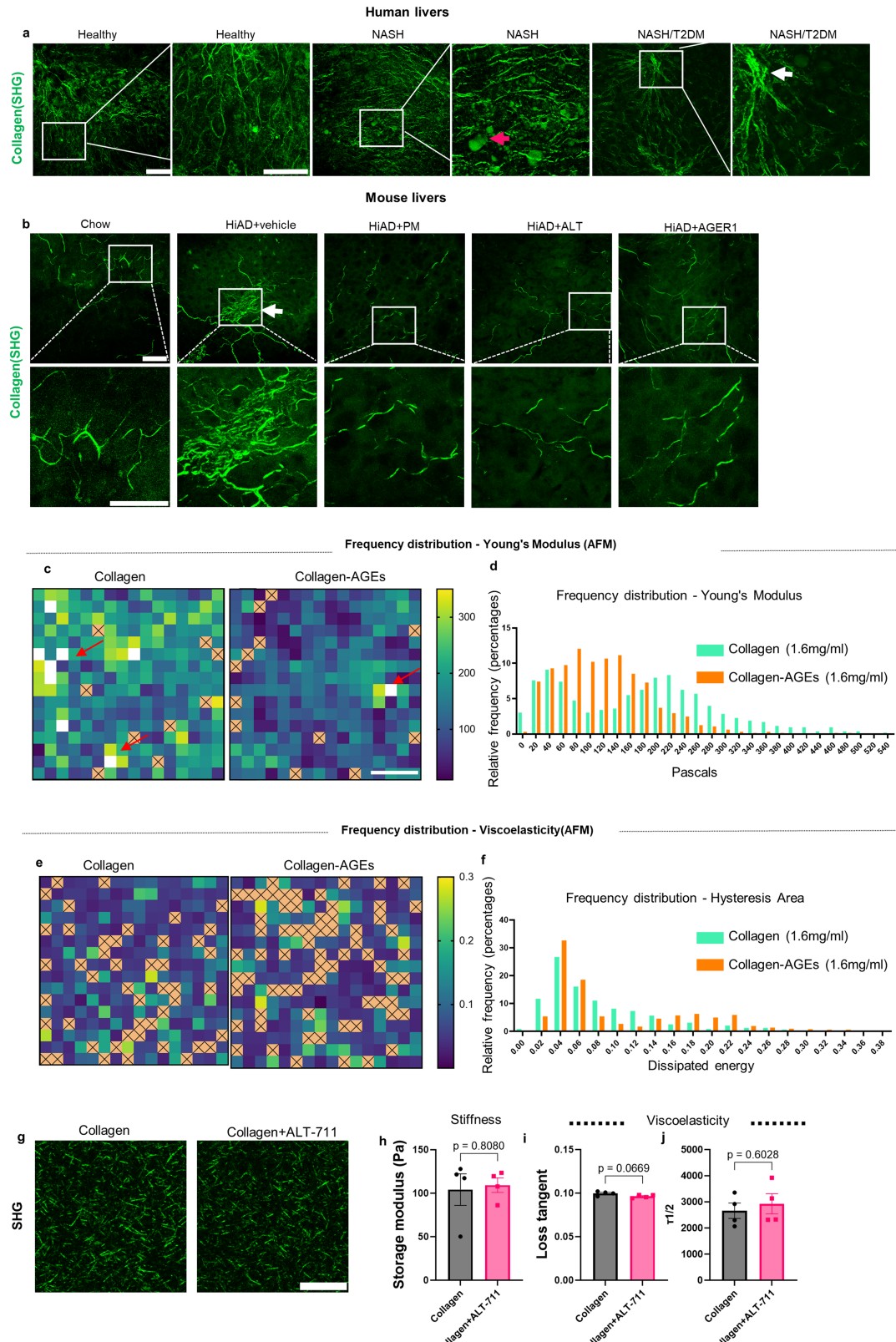

**Extended Data Fig. 3** | See next page for caption.

**Extended Data Fig. 3 | Collagen networks in patients, mouse models and 3D hydrogels. a**. Collagen fibres in human liver samples (healthy, NASH, NASH+ T2DM) were analysed by second harmonic generation (SHG) microscopy. Areas with bundling is depicted in the NASH + T2DM group (white arrow). Red arrow points to lipid droplets. Maximum intensity projection Z-stack for section thickness of 30 μm (Representative images from three individual subjects. Scale bar, 200 μm). **b**. In mouse livers images depict a more bundled appearance of collagen network (white arrow) in the HiAD+vehicle group whereas more organized fibres following PM, ALT treatment or after AGER1 reconstitution. Maximum intensity projection Z-stack for section thickness of 30 μm (Representative images from three individual mice. Scale bar, 200 μm). **c**. Local patches with higher Young's moduli could be observed in collagen hydrogels (red arrows). **d**. The ranges of elastic moduli is depicted in collagen or collagen+AGEs hydrogels. **e**. Mapping and distribution of hysteresis areas (viscoelasticity). **f**. Higher frequencies of increased viscoelasticity in collagen+AGEs gels. n = 5 gels/each group, 3 representative areas/each gel. For all maps, x's indicate regions where AFM indentation curves could not be reliably analysed. Scale bar, 20 μm. **g-j**. To analyse the potential effect of ALT on the matrix, collagen gels +/− ALT-711 were studied by SHG (**g**, Scale bar, 100 μm). Gels from **g** were loaded for rheometry to assess stiffness (**h**, storage modulus) and viscoelasticity (**i**, loss tangent; **j**, τ1/2), (n = 4 each). Error bars represent mean ± s.e.m. n numbers refer to independent experiments. Two-tailed, unpaired t-tests for statistical analysis. ns, not significant.

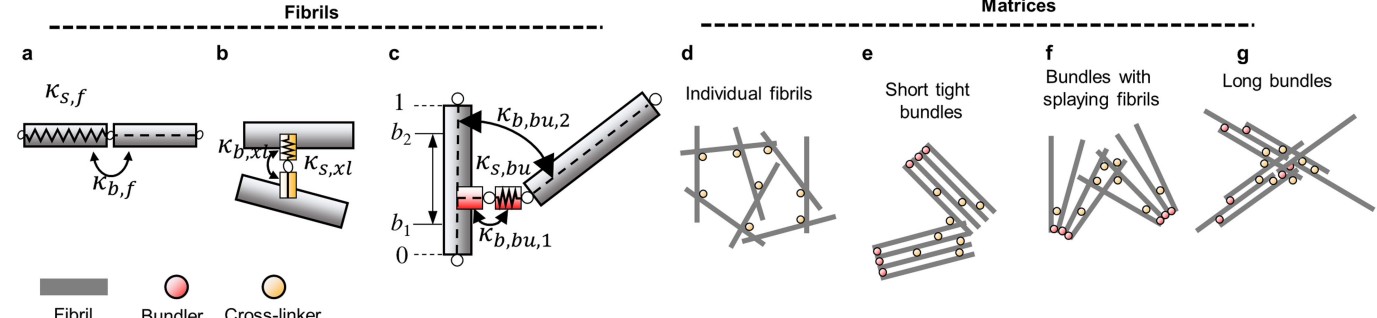

**Fibrils**

**a** $\kappa_{s,f}$ $\kappa_{b,f}$

**b** $\kappa_{b,xl}$ $\kappa_{s,xl}$

**c** $\kappa_{b,bu,2}$ $\kappa_{s,bu}$ $\kappa_{b,bu,1}$

Fibril Bundler Cross-linker

**Matrices**

**d** Individual fibrils

**e** Short tight bundles

**f** Bundles with splaying fibrils

**g** Long bundles

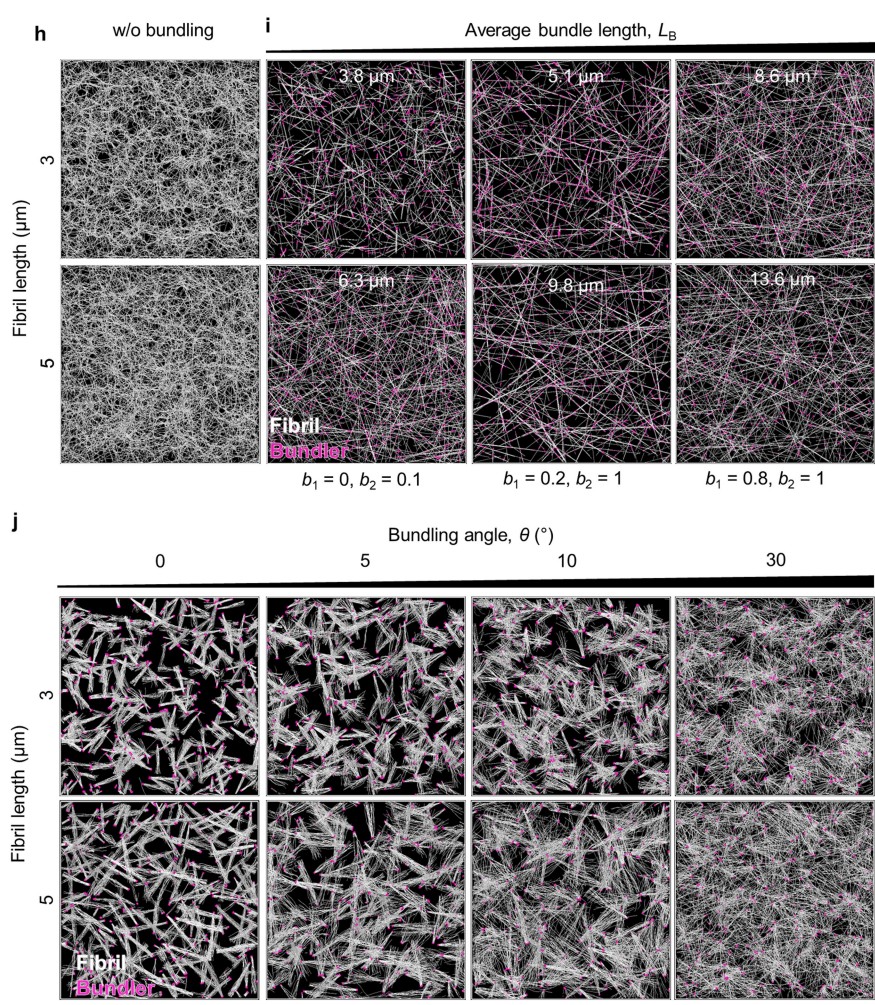

**h** w/o bundling

**i** Average bundle length, $L_B$

Fibril length (µm)

3.8 µm 5.1 µm 8.6 µm

6.3 µm 9.8 µm 13.6 µm

Fibril
Bundler

$b_1 = 0, b_2 = 0.1$ $b_1 = 0.2, b_2 = 1$ $b_1 = 0.8, b_2 = 1$

**j** Bundling angle, $\theta$ (°)

0 5 10 30

Fibril length (µm)

Fibril
Bundler

**Extended Data Fig. 4** | See next page for caption.

**Extended Data Fig. 4 | Agent-based computational model for a fibrillar matrix. a-c**. Fibrils (grey, "f"), cross-linkers (yellow, "xl"), and bundlers (red, "bu") are simplified by cylindrical segments in the model. Cross-linkers connect pairs of fibrils without preference of a cross-linking angle by binding-to-binding sites in any part of two fibrils. By contrast, bundlers connect pairs of fibrils with a specific angle and then maintain the angle. The first binding site of bundlers is always located at the end of fibrils, and the second binding site is located at specified part of fibrils. The specific part available for binding is defined by two boundaries, $b_1$ and $b_2$, between 0 and 1. Various bending ($\kappa_b$) and extensional ($\kappa_s$) stiffnesses maintain angles and lengths near their equilibrium values, respectively. Stiffnesses, equilibrium lengths, and equilibrium angles are listed in the Supplementary Table 5. **d-g**. Different types of matrices. Without bundlers, a matrix is comprised of individual fibrils cross-linked to each other, resulting in small mesh size (**d**). With bundlers which bind only to the ends of fibrils, a matrix consists of short bundles. Depending on the angle between fibrils connected by bundlers, the shape of short bundles varies (**e, f**). With bundlers that bind to the end of one fibril and the mid of another fibril, a matrix consists of longer bundles (**g**). Cross-linkers can connect fibrils within each bundle or fibrils that belong to different bundles. **h-j:** Snapshots of matrices employed for rheological measurements. The length of fibrils used for creating matrices is either 3 μm (top row) or 5 μm (bottom row). (Images displayed are representatives of 4 independent simulations). **h**. Matrix structures with a homogeneous, fine mesh, which is created without bundlers as shown in **d. i**. Matrix structures consisting of long, tight bundles with different lengths. Fibrils are connected in parallel by bundlers as shown in **g**. The length of bundles can be changed by varying $b_1$ and $b_2$. If the second binding site can bind only to part near one end of fibrils (e.g., $b_1 = 0.8$, $b_2 = 1$), the average length of bundles ($L_B$) becomes large. By contrast, if the binding can take place only near the other end of fibrils (e.g., $b_1 = 0$, $b_2 = 0.1$), $L_B$ is slightly longer than the length of individual fibrils. **j**. Matrix structures consisting of short, loose bundles with different bundling angles, $\theta$. In these cases, both binding sites of bundlers bind to the end of fibrils (i.e., $b_1 = b_2 = 0$) as shown in **e** and **f**. The shape of the bundle is varied by changing $\theta$.

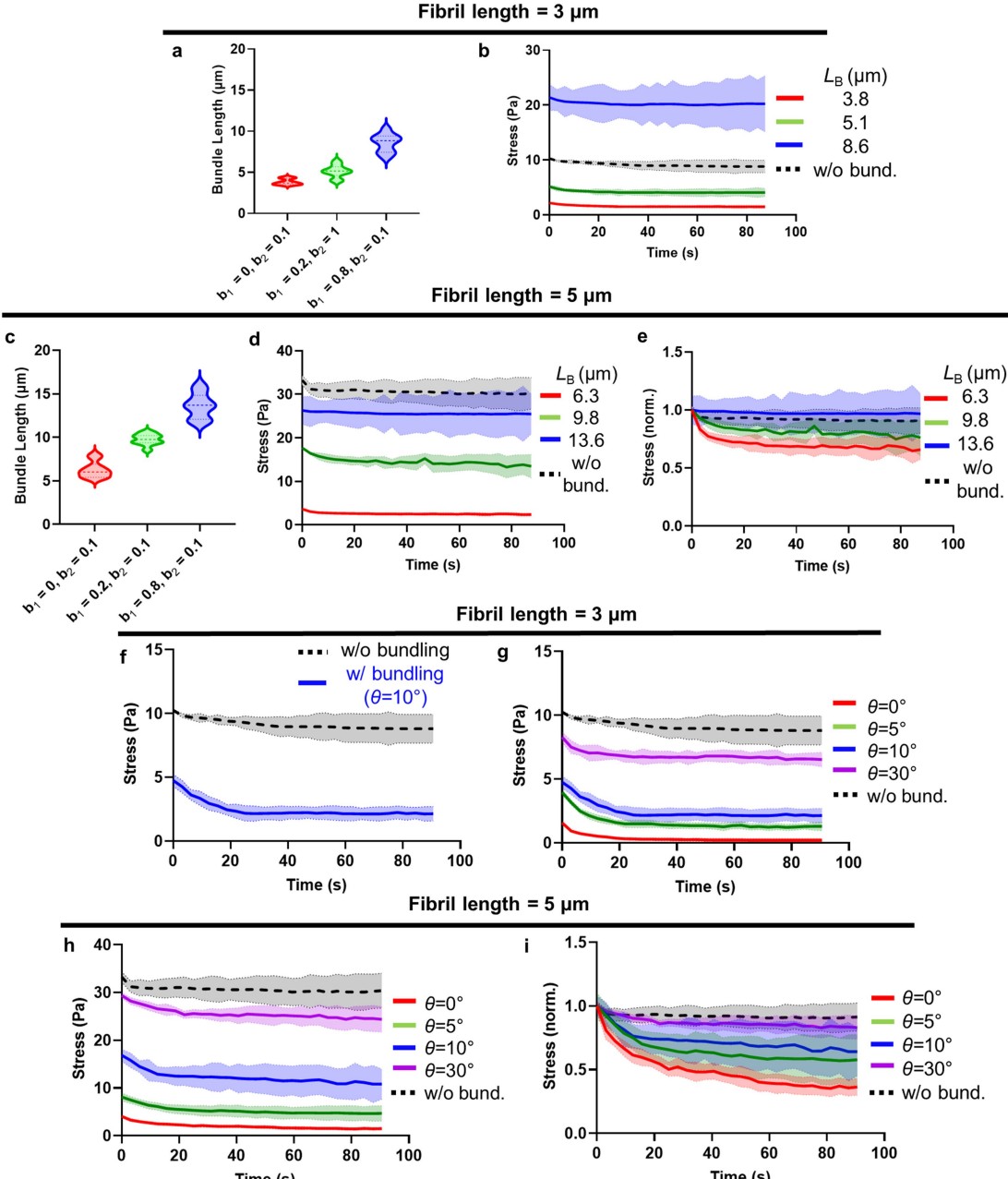

**Extended Data Fig. 5 | Stress relaxation in matrices with different bundle lengths, $L_B$, and angles, $\theta$. a, b.** Bundle length distribution with three different values of $b_1$ and $b_2$ for fibril length of 3 µm **(a)**. Stress relaxation without normalization for cases shown in Fig. 3s and a case without bundles **(b)**. **c-e.** Bundle length distribution with different $b_1$ and $b_2$ **(c)** and stress relaxation without **(d)** or with normalization **(e)** for fibril length of 5 µm. Faster stress relaxation is observed in matrices consisting of smaller bundles. **f, g.** Stress relaxation without normalization for the cases shown in main Fig. 3r **(f)** and Fig. 3t **(g)**. In **f** and **g**, the length of fibrils is 3 µm. **h, i.** Stress relaxation in cases with different angles $\theta$ for fibrils length of 5 µm. The bundle length was changed by a variation in the two boundaries defining the second binding sites for bundlers, $b_1$ and $b_2$. When these fibrils are connected in parallel ($\theta = 0°$) by bundlers, they create long densely packed bundles. The length of these bundles ($L_B$) is determined by the specific point of attachment between the bundler and the fibrils. In cases where bundlers exclusively attach to the fibril ends, the resulting matrix consists of short bundles. These loosely packed bundles have a length equal to that of the fibrils (3 µm). The shape of these short, loosely arranged bundles varies depending on the angle ($\theta$) between the connected fibrils. Error bands represent mean ± s.d. Data displayed are representatives of 4 independent simulations.

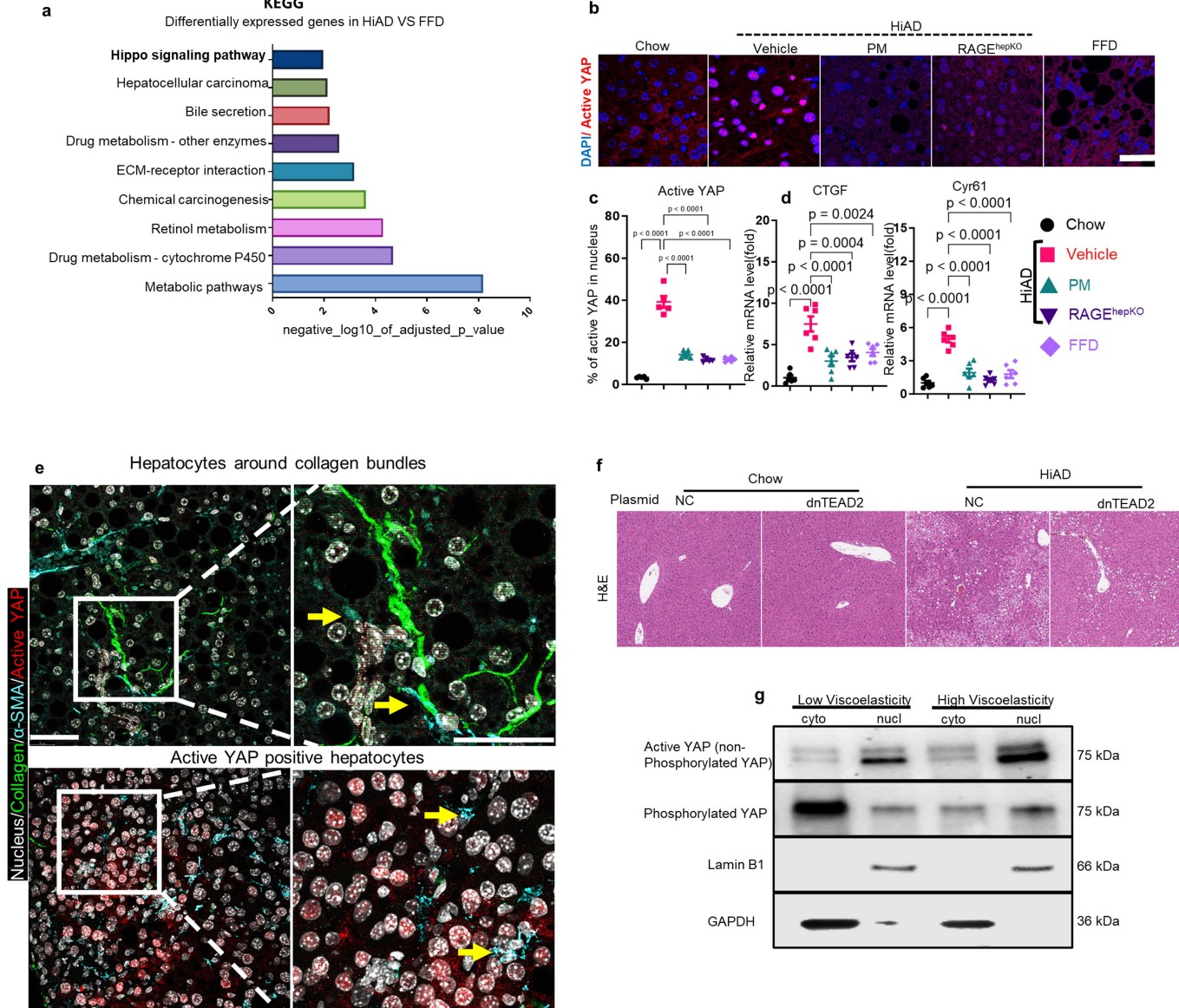

**Extended Data Fig. 6 | YAP pathways are induced by matrix viscoelasticity.**
**a.** Analyses of bulk RNA-seq data from mice fed chow, FFD, and HiAD diets. HiAD-fed mice were treated by PM or vehicle, and a group of mice with RAGE hepatocyte depletion (RAGE[HepKO]) were studied. Genes between FFD and HiAD groups with Log2 fold change (Log2FC) and p-value less than 0.05 (Fisher's exact test) were considered differentially expressed. KEGG analyses from differentially expressed genes show enrichment in Hippo signalling pathway. No HD injection was done in these experiments. **b, c.** Representative images (**b**) and quantification (**c**) of active nuclear YAP signal in mouse livers, using an antibody against the active, non-phosphorylated YAP. (n = 5 mice/group, 4 random ×20 fields/sample; data are presented as the percentage of active YAP/area/×20 field). **d.** YAP targets CTGF and Cyr61 were induced in mice on HiAD,

but not after PM or in RAGE[HepKO] on HiAD. RT-qPCR, n = 6 each. **e.** Collagen was imaged using SHG (green), active YAP (red) and α-SMA positive stellate cells (blue, arrows) by immunofluorescence. There were no hepatocytes with active YAP observed in the close proximity of collagen bundles (using an antibody against the non-phosphorylated active YAP). Scale bar 50 μm. **f.** HE images corresponding to sequential slides with myc-tag/GS immunohistochemistry in the main Fig. 4c. **g.** Nuclear and cytoplasmic YAP was assessed using antibodies against active, non-phosphorylated and inactive phosphorylated YAP in western blots (cytoplasmic and nuclear fractions), in low and high viscoelasticity hydrogels (representative of 3 different experiments). Error bars represent mean ± s.e.m. n numbers refer to individual mice. One-way ANOVA was used followed by Tukey's multiple comparison test.

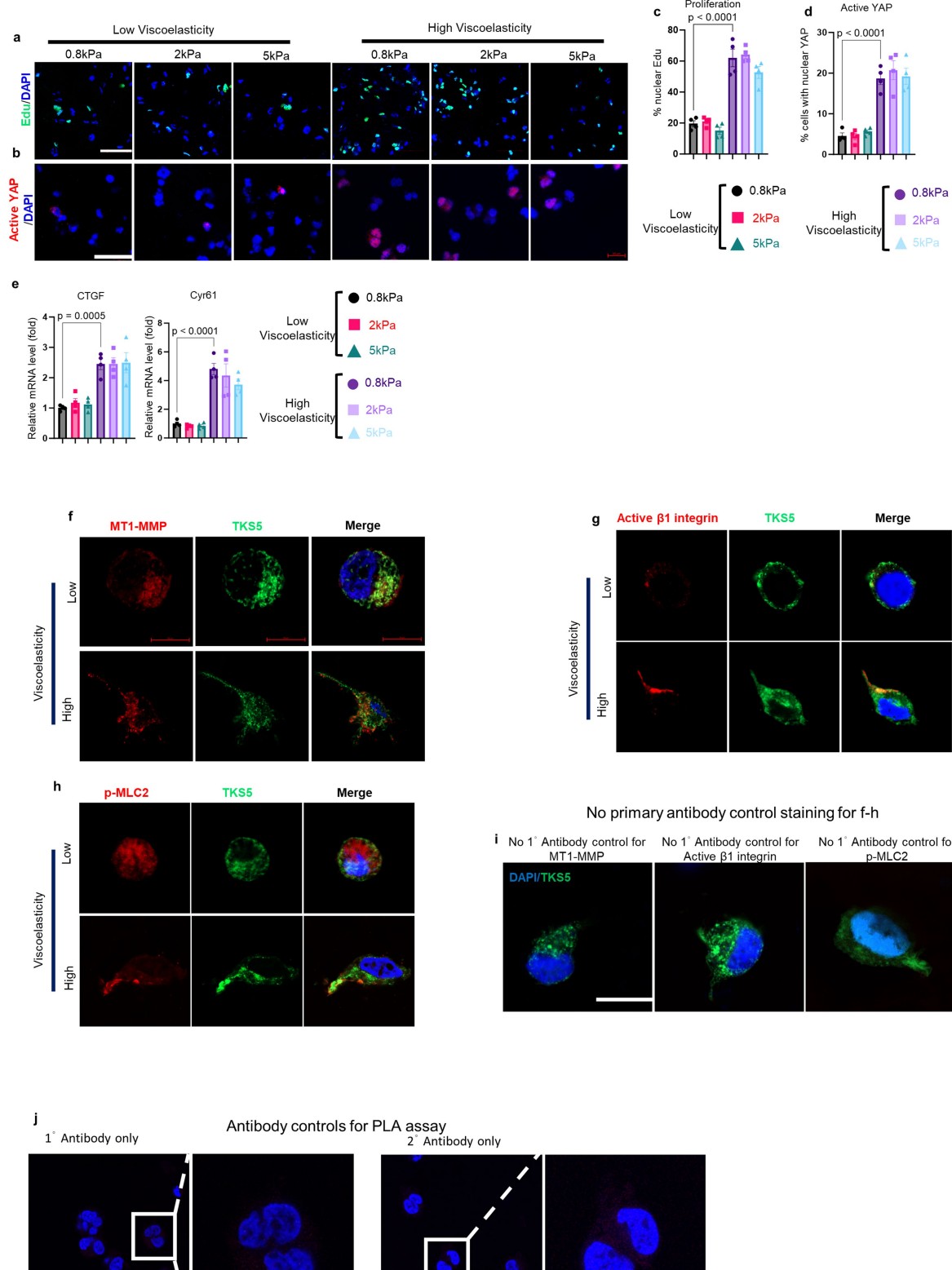

**Extended Data Fig. 7 |** See next page for caption.

**Extended Data Fig. 7 | Proliferation and YAP activation are not affected by increasing stiffness in low or high viscoelasticity hydrogels (Additional data to the main Fig. 5). a-e.** Huh7 cells were encapsulated in low or high viscoelasticity IPN hydrogels with varying stiffness. (0.8-5 kPa). Cell proliferation and YAP activity were evaluated by Edu nuclear signal (**a**, scale bar, 100 μm) and active YAP immunofluorescence (**b**, scale bar, 50 μm), and quantification (**c** and **d**). **e**. mRNA expression of YAP target genes CTGF and Cyr6. (n = 4, for **c**-**e**, Error bars represent mean ± s.e.m. n numbers refer to independent experiments. One-way ANOVA with Tukey tests was used for correction of multiple comparisons. **f-h.** Huh7 cells were transfected with Tks5-mNeonGreen, and immunofluorescence microscopy was performed using an antibody against MT1-MMP (**f**), active integrin β1 (**g**), and p-MLC2 (**h**). **i.** Controls with no primary antibody for **f-h**. Cells were imaged by high resolution Airyscan microscopy (LSM980, Zeiss). Representative images, scale bar, 10 μm. **j.** Controls for the PLA assay in the main Fig. 5l (1° antibody only, 2° antibody only). Scale bar, 10 μm. Error bars represent mean ± s.e.m. n numbers refer to independent experiments. One-way ANOVA was used followed by Tukey's multiple comparison test.

# Huh7 cells

**Extended Data Fig. 8 | TNS1 and integrin β1 knockdowns reduce proliferation, active nuclear YAP and formation of invadopodia-like structures (Additional data to the main Fig. 5). a-d,** Huh7-Cas9 cells were transfected with plasmids containing CRISPR guide RNA of TNS1 (sg-TNS1), or integrin β1 (sg-Itg β1) or control sgRNA (NC), and cells after 24 h were embedded in low or high viscoelasticity hydrogels. After 48 h, cell proliferation was evaluated by Edu analyses (**a**, Scale bar, 50 µm). YAP activity was analysed by using an antibody against active YAP (**b**, Scale bar, 20 µm). Cell circularity was analysed by F-actin, and ImageJ analyses (**c**, Scale bar, 20 µm.), and the formation of invadopodia-like structures was assessed by the TKS5 signal (**d**, Scale bar, 10 µm). **e.** mRNA expression of YAP-regulated target genes, CTGF and Cyr61 (n = 5). Error bars represent mean ± s.e.m. n numbers refer to independent experiments. One-way ANOVA test was used followed by Tukey's multiple comparison.

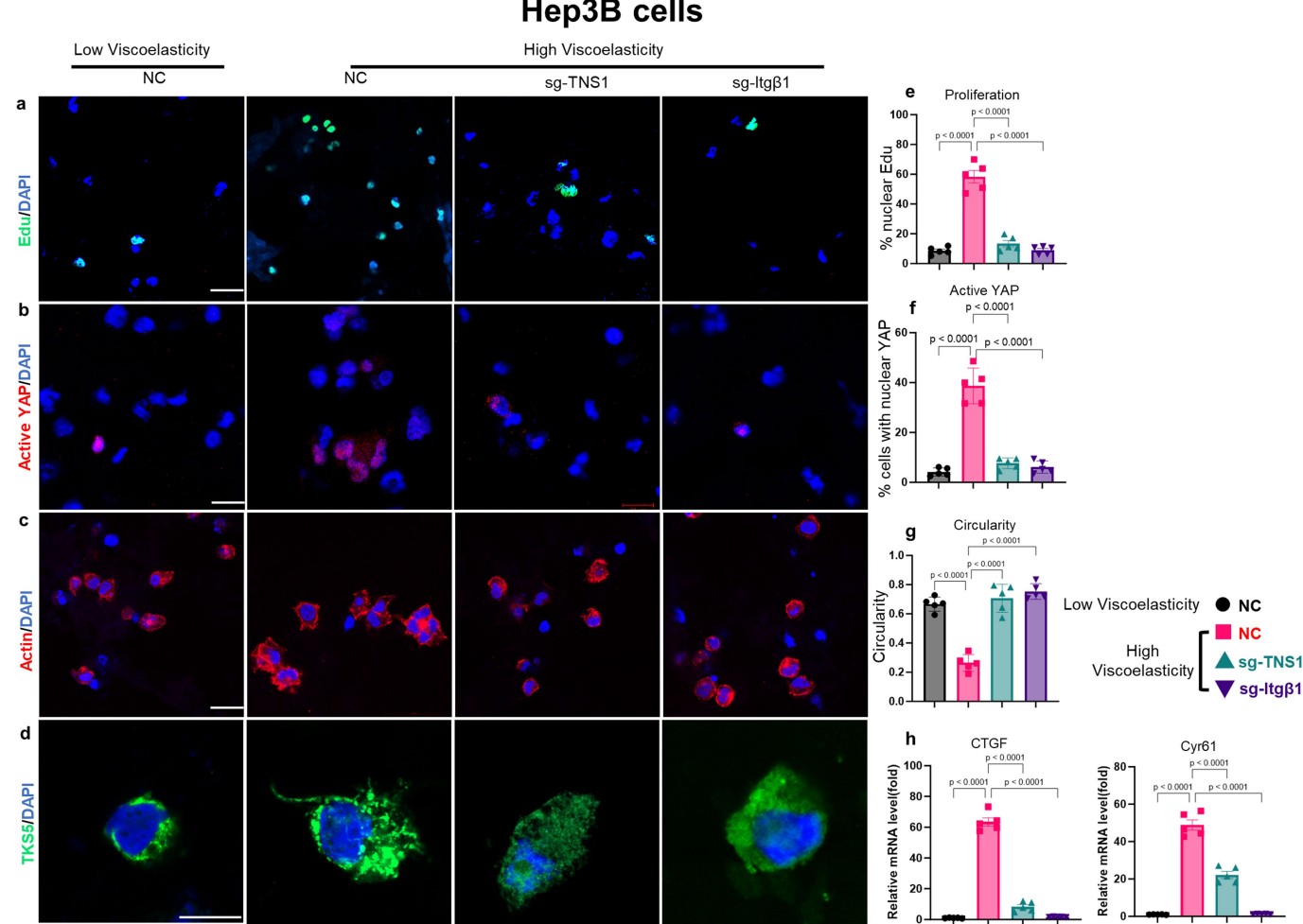

**Extended Data Fig. 9 | TNS1 and Integrin β1 knockdowns reduce proliferation, activation of YAP and formation invadopodia-like structures in Hep3B cells.** Hep3B cells were transfected with CRISPR/Cas9 plasmids to knockdown TNS1 (sg-TNS1) or β1 integrin (sg-Itgβ1). A control, not targeted sgRNA was used as control (NC). After 48 h in 3D culture in low or high viscoelasticity hydrogels, cell proliferation was evaluated by Edu (**a**, Scale bar, 50 μm) and quantification of the signal (**e**). YAP activity was analysed by using active YAP antibody (**b**, Scale bar, 20 μm), quantification (**f**), and mRNA expression of YAP-regulated targets (**h**). Cell circularity was evaluated by the F-actin signal (**c**, Scale bar, 20 μm) and ImageJ analyses (**g**). Formation of invadopodia-like structures was analysed by the TKS5 signal (**d**, Scale bar, 10 μm). n = 5 each for **e-h**. Error bars represent mean ± s.e.m. n numbers refer to independent experiments. One-way ANOVA test was used followed by Tukey's multiple comparison.

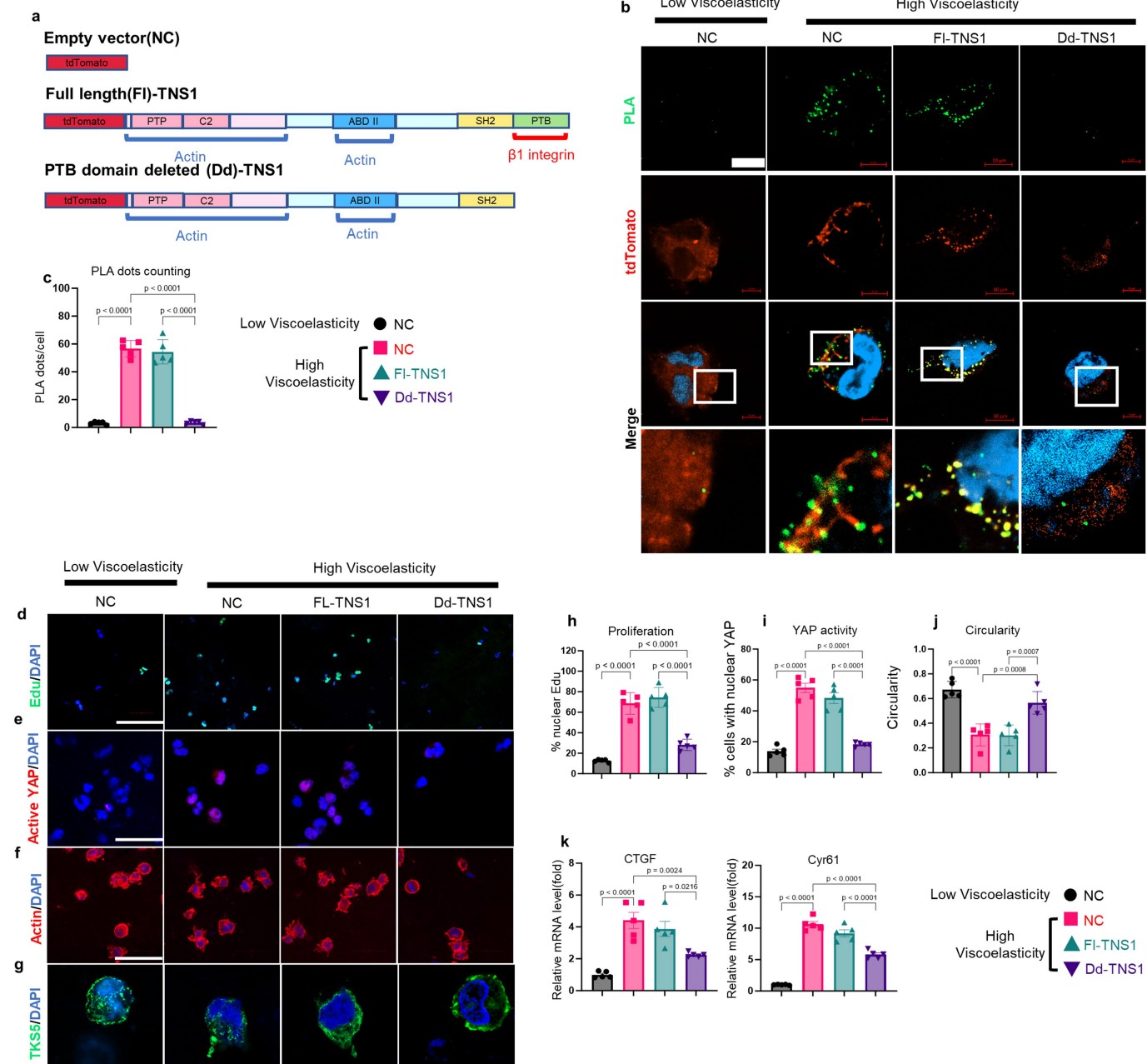

**Extended Data Fig. 10 | PTB domain deleted (dd) TNS-1 did not colocalize with integrin, and cells exhibited lower proliferation, YAP activity, and reduced formation of invadopodia-like structures in high viscoelasticity hydrogels. a**. Schematics of the PTB domain deleted (dd) TNS1 construct. Deleting this domain prevents binding to integrins however the actin-binding domain remains intact. **b, c**. Proximity ligation assays **(b)** to assess integrin β1 and TNS1 binding (green signal) in empty vector (NC), full length TNS1, and dd-TNS1 transfected cells (red, tdTomato). In full length TNS1 tomato, PLA signals colocalized whereas no colocalization was seen in dd-TNS1 transfected cells or in low viscoelasticity matrix. PLA positive dots were quantified from 30 cells in 5 gels, each group (**c**, n = 5 each). **d-k**. Cell proliferation was evaluated by Edu (**d**, Scale bar, 100 μm) and quantification (**h**). YAP activity was analysed by using active YAP immunofluorescence (**e**, Scale bar, 50 μm.), quantification (**i**), and YAP-regulated target gene mRNA expression (**k**). Cell circularity was evaluated by F-actin signal (**f**, Scale bar, 50 μm, and) and quantification (**j**, ImageJ). Formation of invadopodia-like structures was analysed by the TKS5 signal (**g**, Scale bar, 10 μm, n = 5 each. Error bars represent mean ± s.e.m. n numbers refer to independent experiments. One-way ANOVA test was used followed by Tukey's multiple comparison.

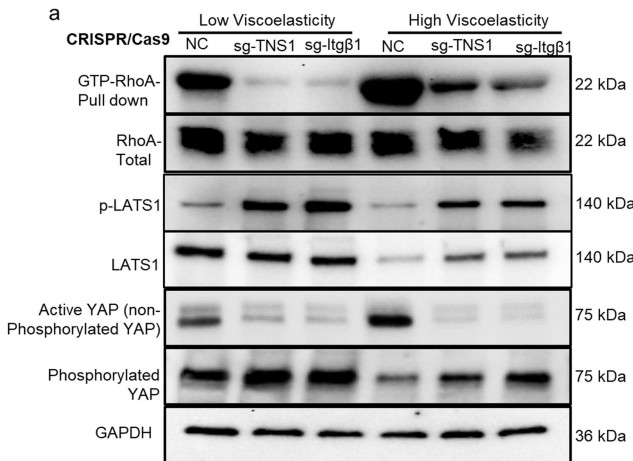

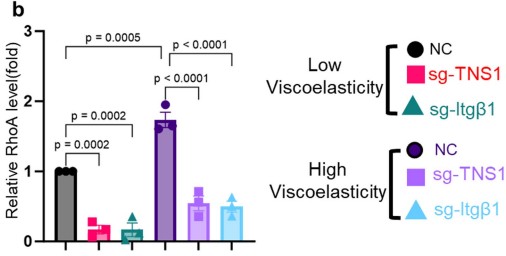

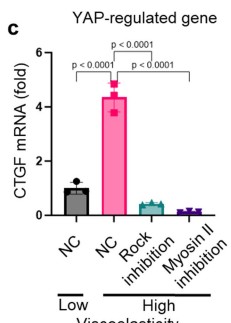

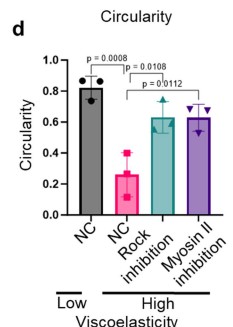

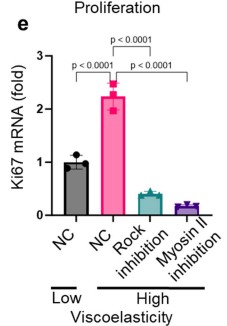

**Extended Data Fig. 11 | Integrin β1 and Tensin 1 mediate viscoelasticity-specific mechano-cellular pathways involving YAP activation (Additional data to main Fig. 5). a.** Huh7-Cas9 cells were transfected with plasmids containing CRISPR guide RNA for TNS1 (sg-TNS1), or Integrin β1 (sg-Itg β1) or control sgRNA (NC), and cells after 24 h were embedded in low or high viscoelasticity hydrogels. RhoA GTPase activity in low/high viscoelasticity conditions and after TNS-1 or Integrin β1 KDs was tested by pull-down assays. Antibodies to active (non-phosphorylated), inactive YAP (phosphorylated), as well as to phosphorylated and total LATS1, were used, and analysed by immunoblotting. Representative images of 3 independent experiments.

**b**. Quantification of GTP-RhoA/GAPDH protein levels from **a** (n = 3 each). **c-e**. Huh7 cells encapsulated in low or high viscoelasticity IPN hydrogels were incubated with ROCK (Y-27632, Abcam, 10 μM) or Myosin II inhibitors (Blebbistatin, Abcam, 50 μM). YAP activity was analysed by testing YAP-regulated target gene CTGF mRNA expression (**c**). Cell circularity was analysed by Image J (**d**), and cell proliferation was evaluated by Ki67 mRNA expression (**e**) (n = 3 each). Error bars represent mean ± s.e.m. n numbers refer to independent experiments. One-way ANOVA test was used followed by Tukey's multiple comparison.

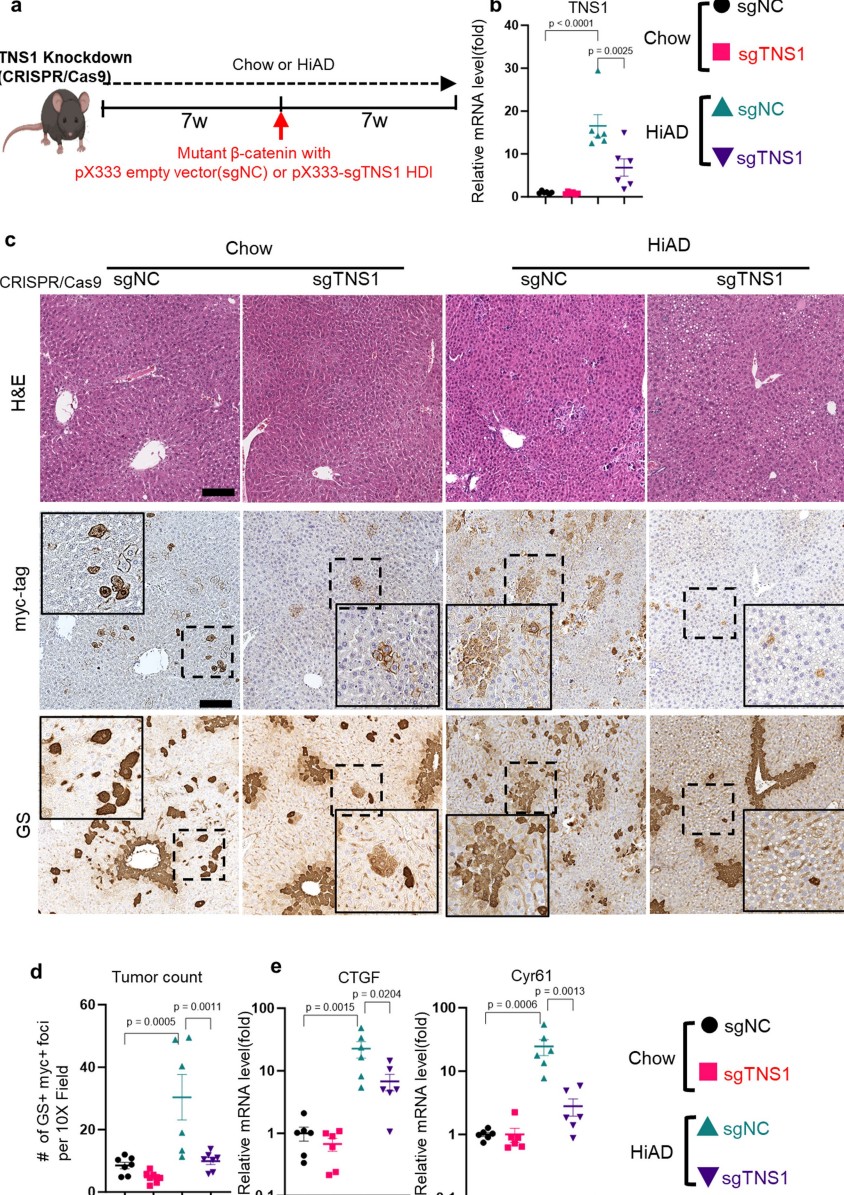

**Extended Data Fig. 12 | TNS-1 knockdown decreases the formation of transformed foci in mice on HiAD, after HDI. a.** Schematic presentation of the in vivo targeting of TNS1 by CRISPR/Cas9 in conjunction with hydrodynamic injection (HDI). Mice were fed chow or HiAD for 7w, then hydrodynamically injected with pT3-EF5a-hMet and the pT3-354 EF5a-S45Y-β-catenin-myc (mutant β-catenin), with the sleeping beauty (SB) transposase, as well as the CRISPR-Cas9-based vector either with sgRNAs targeting mouse TNS1 (pX333-sgTNS1) or empty vector (sgNC). Mice were sacrificed 7 weeks following injection. **b.** TNS1 expression in the liver was analysed by RT-qPCR (n = 6 each). **c, d.** H&E images, GS/myc immunohistochemistry on consecutive slides depict colocalization (**c**). Quantification (**d**) of GS/myc positive foci. Scale bar, 300 μm (n = 6 each). **e.** The expression of YAP targets CTGF and Cyr 61 was analysed by RT-qPCR (n = 6 each). Error bars represent mean ± s.e.m. n numbers refer to individual mice. One-way ANOVA test was used followed by Tukey's multiple comparison.

# Reporting Summary

## Statistics

For all statistical analyses, confirm that the following items are present in the figure legend, table legend, main text, or Methods section.

| n/a | Confirmed | |
|---|---|---|
| ☐ | ☒ | The exact sample size (*n*) for each experimental group/condition, given as a discrete number and unit of measurement |
| ☐ | ☒ | A statement on whether measurements were taken from distinct samples or whether the same sample was measured repeatedly |
| ☐ | ☒ | The statistical test(s) used AND whether they are one- or two-sided<br>*Only common tests should be described solely by name; describe more complex techniques in the Methods section.* |
| ☐ | ☒ | A description of all covariates tested |
| ☐ | ☒ | A description of any assumptions or corrections, such as tests of normality and adjustment for multiple comparisons |
| ☐ | ☒ | A full description of the statistical parameters including central tendency (e.g. means) or other basic estimates (e.g. regression coefficient) AND variation (e.g. standard deviation) or associated estimates of uncertainty (e.g. confidence intervals) |
| ☐ | ☒ | For null hypothesis testing, the test statistic (e.g. *F*, *t*, *r*) with confidence intervals, effect sizes, degrees of freedom and *P* value noted<br>*Give P values as exact values whenever suitable.* |
| ☒ | ☐ | For Bayesian analysis, information on the choice of priors and Markov chain Monte Carlo settings |
| ☒ | ☐ | For hierarchical and complex designs, identification of the appropriate level for tests and full reporting of outcomes |
| ☒ | ☐ | Estimates of effect sizes (e.g. Cohen's *d*, Pearson's *r*), indicating how they were calculated |

*Our web collection on statistics for biologists contains articles on many of the points above.*

## Software and code

Policy information about availability of computer code

| | |
|---|---|
| Data collection | Histology and Immunohistochemistry Images were scanned with Leica Aperio AT2 System.<br>High resolution IFC images were obtained by the ZEISS Airyscan2 LSM980 microscope.<br>AFM Data analyses were performed using the Hertz model in NanoScope Analysis V1.9, and Mountains SPIP v.9.<br>Rheometry was carried out on an ARES-G2 rheometer (TA instruments) by using TA TRIOS software V5.1.1 (TA instruments).<br>Second Harmonic Generation images and fluorescent images were obtained using Leica TCS SP8, multi-photon Leica Stellaris 8 DIVE upright Confocal, and ZEISS Airyscan2 LSM980.<br>Immunoblotting images were collected using the Invitrogen iBright Imaging Systems  (Thermo Fisher Scientific - US).<br>RNA-seq data were collected on an Illumina NovaSeq 6000 Sequencing with paired-end 150bp reads. |
| Data analysis | The AFM data and Second Harmonic Generation images were analyzed using Mountains SPIP v.9, CT Fire software (V2.0 beta), and NIH Image J software (Version 1.53t).<br>Immunoblotting images were analyzed using iBright Analysis Software (Version 5.2.1).<br>Statistical analyses were performed using Graphpad Prism (GraphPad Software, version 10).<br>For RNA-seq analysis,Gencode gene annotations version M18, GRCm38 (https://www.gencodegenes.org/), Dropseq tools v1.1249, R v3.4.4., Limma v3.40.650  were used.<br>GSEA was conducted with the pre-ranked GSEA method within the KEGG databases with the online tool g:Profiler (https://biit.cs.ut.ee/gprofiler/gost). RNA-seq Heatmaps and unsupervised hierarchical clustering was performed with g:Profiler (https://biit.cs.ut.ee/gprofiler/gost). |

For manuscripts utilizing custom algorithms or software that are central to the research but not yet described in published literature, software must be made available to editors and reviewers. We strongly encourage code deposition in a community repository (e.g. GitHub). See the Nature Portfolio guidelines for submitting code & software for further information.

# Data

Policy information about availability of data

All manuscripts must include a data availability statement. This statement should provide the following information, where applicable:

- Accession codes, unique identifiers, or web links for publicly available datasets
- A description of any restrictions on data availability
- For clinical datasets or third party data, please ensure that the statement adheres to our policy

We do not have any restrictions on data availability. All data generated during this study, are included in the article and the supplementary information files.  The mouse reference genome major release GRCm38 are available from https://www.gencodegenes.org/.
RNA-seq data are available under the accession number GSE245016. Codes used in the study are available at https://github.com/ktyman2/liverCancer.git

# Human research participants

Policy information about studies involving human research participants and Sex and Gender in Research.

| | |
|---|---|
| Reporting on sex and gender | We have obtained de-identified liver samples and studied them from 9 female and 11 male adult patients. Acquisition was according to the BRISQ Guidelines, and liver resection specimens that were stored at -80C or available fresh (for rheometry analyses) were studied. |
| Population characteristics | MASH patients (age between 30 to 80) with no HBV/HCV/HIV infection and no history of heavy alcohol drinking. Histology was evaluated for necroinflammation, hepatocellular ballooning, ductular reaction, and fibrosis by a hepato-pathologist in a blinded fashion, and NAS scores were provided. |
| Recruitment | Human liver samples were obtained from Stanford Diabetes Research Center (SDRC), Donor Network West (DNW), Stanford Tissue Bank, and the Clinical Biospecimen Repository and Processing Core (CBRPC) of the Pittsburgh Liver Research Center (PLRC). No identifying information was available. |
| Ethics oversight | All human samples were de-identified and exempted (Exemption 4). This was approved by Stanford University Institutional Review Board (IRB, #67378) and Pittsburgh Liver Research Center Review Board. |

Note that full information on the approval of the study protocol must also be provided in the manuscript.

# Field-specific reporting

Please select the one below that is the best fit for your research. If you are not sure, read the appropriate sections before making your selection.

☒ Life sciences ☐ Behavioural & social sciences ☐ Ecological, evolutionary & environmental sciences

For a reference copy of the document with all sections, see nature.com/documents/nr-reporting-summary-flat.pdf

# Life sciences study design

All studies must disclose on these points even when the disclosure is negative.

| | |
|---|---|
| Sample size | Sample size was determined by pilot experiments and resource availability. |
| Data exclusions | We have not performed any data exclusion. |
| Replication | All findings have been replicated with more than 3 biological repeats, and methods. The key findings were verified independently by other individuals. |
| Randomization | Mice in  each independent experiment, as well as the experimental groups  were age-matched.  Mice were randomly distributed into each group ensuring similar age and body weight in each  group. For in vitro experiments, cells and hydrogels were grouped randomly for treatment or test. |
| Blinding | Blinding was widely used in the study. The images were scored by at least two individuals who were blinded for the group information. Measurements of transformed foci were confirmed by a second person who was also blinded for the information on the experimental group. Investigators were blinded to allocation during experiments and outcome assessments, and data was collected and analyzed in a blinded fashion. |

# Reporting for specific materials, systems and methods

We require information from authors about some types of materials, experimental systems and methods used in many studies. Here, indicate whether each material, system or method listed is relevant to your study. If you are not sure if a list item applies to your research, read the appropriate section before selecting a response.

## Materials & experimental systems

| n/a | Involved in the study |
|-----|----------------------|
| ☐ | ☒ Antibodies |
| ☐ | ☒ Eukaryotic cell lines |
| ☒ | ☐ Palaeontology and archaeology |
| ☐ | ☒ Animals and other organisms |
| ☒ | ☐ Clinical data |
| ☒ | ☐ Dual use research of concern |

## Methods

| n/a | Involved in the study |
|-----|----------------------|
| ☒ | ☐ ChIP-seq |
| ☒ | ☐ Flow cytometry |
| ☒ | ☐ MRI-based neuroimaging |

## Antibodies

**Antibodies used**

Myc  Santa Cruz , #sc-40 IHC, 1:200
Glutamine Synthetase (GS) Santa Cruz, #sc-74430 IHC, 1:200
Tensin 1 (TNS1) Sigma-Aldrich, #SAB4200283 IF, 1:200; PLA, 1:100
Integrin β1 blocking (Itgb1) Abcam, #ab24693 Cell culture, 1:100
Active Integrin β1 (12G10) Abcam, # ab30394 IF, 1:200; PLA, 1:100
Active (non-phosphorylated) Yap Abcam, # ab 205270 IF, 1:200; WB, 1:1000
MT1-MMP (MMP14) Abcam, # ab 51074 IF, 1:200
Phospho-Myosin Light Chain 2 (Ser19) Cell Signaling Technology, # 95777 IF, 1:200
phosphorylated Yap (Ser127) Cell Signaling Technology, #4911 WB, 1:1000
GAPDH Santa Cruz, # sc-365062 WB, 1:10,00
LATS1 Cell Signaling Technology, # 3477 WB, 1:10,00
Phospho-LATS1 (Thr1079) Cell Signaling Technology, # 8654 WB, 1:10,00

Secondary Antibodies
HRP Goat α-Rabbit Abcam, #ab6721 WB, 1:5000
Alexa Fluor 488 Chicken α-Rabbit Invitrogen, #A21441 IF, 1:500
Alexa Fluor 555 Goat α-Mouse Invitrogen, #A21422 IF, 1:500
Alexa Fluor 555 Donkey α-Rabbit Invitrogen, #A31572 IF, 1:500
Biotinylated Goat α-Mouse  Vector Lab,#BA-9200 IHC, 1:500

**Validation**

Myc  https://www.scbt.com/p/c-myc-antibody-9e10
Glutamine Synthetase (GS) https://www.scbt.com/p/gl-syn-antibody-e-4
Tensin 1 (TNS1) https://www.sigmaaldrich.com/US/en/product/sigma/sab4200283
Integrin β1 blocking (Itgb1) https://www.abcam.com/integrin-beta-1-antibody-p5d2-ab24693.html
Active Integrin β1 (12G10) https://www.abcam.com/products/primary-antibodies/integrin-beta-1-antibody-12g10-ab30394.html
Active (non-phosphorylated) Yap https://www.abcam.com/products/primary-antibodies/active-yap1-antibody-epr19812-ab205270.html
MT1-MMP (MMP14) https://www.abcam.com/products/primary-antibodies/mmp14-antibody-ep1264y-ab51074.html
Phospho-Myosin Light Chain 2 (Ser19) https://www.cellsignal.com/products/primary-antibodies/phospho-myosin-light-chain-2-thr18-ser19-e2j8f-rabbit-mab/95777
phosphorylated Yap (Ser127) https://www.cellsignal.com/products/primary-antibodies/phospho-yap-ser127-antibody/4911?site-search-type=Products&N=4294956287&Ntt=%234911&fromPage=plp&_requestid=538663
GAPDH https://www.scbt.com/p/gapdh-antibody-g-9?gclid=Cj0KCQjwnf-kBhCnARIsAFlg490L7314d4A25tXUFjzFpS2TOuYoCeZvaUwYZ_Qixt_PgNhvTkFXH_AaAhL5EALw_wcB
LATS1 https://www.cellsignal.com/products/primary-antibodies/lats1-c66b5-rabbit-mab/3477?site-search-type=Products&N=4294956287&Ntt=%23+3477+&fromPage=plp&_requestid=538731
Phospho-LATS1 (Thr1079) https://www.cellsignal.com/products/primary-antibodies/phospho-lats1-thr1079-d57d3-rabbit-mab/8654?site-search-type=Products&N=4294956287&Ntt=8654+&fromPage=plp&_requestid=538760
Secondary Antibodies:
HRP Goat α-Rabbit Abcam, #ab6721 https://www.abcam.com/products/secondary-antibodies/goat-rabbit-igg-hl-hrp-ab6721.html
Alexa Fluor 488 Chicken α-Rabbit Invitrogen, #A21441https://www.thermofisher.com/antibody/product/Chicken-anti-Rabbit-IgG-H-L-Cross-Adsorbed-Secondary-Antibody-Polyclonal/A-21441
Alexa Fluor 555 Goat α-Mouse Invitrogen, #A21422 https://www.thermofisher.com/antibody/product/Goat-anti-Mouse-IgG-H-L-Cross-Adsorbed-Secondary-Antibody-Polyclonal/A-21422
Alexa Fluor 555 Donkey α-Rabbit Invitrogen, #A31572 https://www.thermofisher.com/antibody/product/Donkey-anti-Rabbit-IgG-H-L-Highly-Cross-Adsorbed-Secondary-Antibody-Polyclonal/A-31572
Biotinylated Goat α-Mouse  Vector Lab,#BA-9200 https://vectorlabs.com/products/biotinylated-goat-anti-mouse-igg/

# Eukaryotic cell lines

Policy information about cell lines and Sex and Gender in Research

| Cell line source(s) | Huh7 cell line was from the Sarnow lab (Stanford), and Hep3B cells were purchased from ATCC. |
| --- | --- |
| Authentication | Validation was performed by STR profiling using ATCC cell authentication service (ASN-0002-2022). |
| Mycoplasma contamination | Routine testing for mycoplasma was conducted by MycoAlertTM Mycoplasma Detection Kits (Lonza, 75870-454). All cells tested negative for mycoplasma. |
| Commonly misidentified lines (See ICLAC register) | No commonly misidentified cell lines were used. |

# Animals and other research organisms

Policy information about studies involving animals; ARRIVE guidelines recommended for reporting animal research, and Sex and Gender in Research

| Laboratory animals | All studies were approved by the Stanford APLAC or Palo Alto VA, and were strictly following the ARRIVE Guidelines.
Wild type C57BL/6J (WT) 8- to 10-week-old male mice were purchased from the Jackson Laboratory. Ragefl/fl mice on a C57B6 background were gifted by Dr. B. Arnold from German Cancer Research Center, Heidelberg, Germany. RageHepKO mice were generated by crossing Ragefl/fl mice with Albumin-cre mice (the Jackson Laboratory) for several generations. Eight- to 10-week-old male Ragefl/fl mice and RageHepKO mice were used. To generate hepatocyte transgenic AGER1 mice, WT mice were injected with adeno-associated virus 8 (AAV8)-control green fluorescent protein (AAV8-control) or AAV8-thyroxine-binding globulin-AGER1 recombinase (AAV8-AGER1) (5x10e11 genome copies, Vector BioLabs) at week 6th of feeding.
To knockdown TNS1, mice were injected with CRISPR/Cas9-based vector linking two sgRNAs targeting TNS1 exon 1 and exon 7 (pX333-TNS1, 50 μg) or negative control vector (empty pX333, 50 μg), by hydrodynamic tail vein injection at the beginning of the 8th week of chow or HiAD feeding at the same time as with hMet, s45y-mutant- β-catenin, and SB
All mice used were on the same background and kept in the same facility. Mice were maintained at macroenvironmental temperature and humidity ranges of 17.8 to 26.1 °C and 30% to 70%. Mice were housed in standard cages with 12:12 hour light/dark cycles and ad libitum access to water and food unless otherwise indicated. |
| --- | --- |
| Wild animals | This study did not involve wild animals. |
| Reporting on sex | Only male mice were used. |
| Field-collected samples | This study did not involve samples collected in the field. |
| Ethics oversight | All animal experiments were conducted according to the experimental procedures approved by the Institutional Animal Care and Use Committee at Stanford University and Palo Alto VA. (APLAC #33374) |

Note that full information on the approval of the study protocol must also be provided in the manuscript.

