## [Peer Review File · Nature]

Manuscript Title: Matrix Viscoelasticity Promotes Liver Cancer Progression in the Pre-Cirrhotic Liver

Reviewer Comments & Author Rebuttals

Reviewer Reports on the Initial Version:

Referees' comments:

Referee #1 (Remarks to the Author):

In this manuscript, Fan et al. investigate the impact of nonalcoholic steatohepatitis (NASH) and type 2 diabetes mellitus (T2DM) on hepatocellular carcinoma (HCC) disease progression. The authors demonstrate that livers of patients with NASH/T2DM compared to those without T2DM exhibit increased viscoelasticity due to the accumulation of advanced glycation end products (AGEs). AGEs alter the crosslinking of collagen fibers within the ECM leading to enhanced ECM viscoelasticity and HCC progression. Inhibiting AGEs production with PM or breaking the formation of AGEs-collagen crosslinks with ALT or reconstituting the clearance receptor AGER1 reverses the changes in viscoelasticity, diminishes HCC growth and improves survival. The authors also suggest that HCC cells respond to altered ECM viscoelasticity via integrin β 1, tensin-1 and YAP. Overall, this is an interesting manuscript, which integrates different animal models with sophisticated in vitro experiments and simulations. The authors neatly demonstrate that viscoelasticity enhances liver cancer progression. However, additional work is required to reinforce their proposed mechanistic findings centering on the involvement of integrin β 1, tensin-1 and YAP.

General comments:

1. In Figs. 2h-n, the authors show that AAV8-mediated AGER1 delivery lowers AGEs and stress relaxation, and reduces the growth of tumor foci, presumably due to AGER1 reconstitution. The authors should also provide the expression levels of AGER1 in mice injected with AAV8-TBG-AGER1 and AAV8-TBG-GFP (control).
2. In Fig. 3, the authors show that the collagen network in mice on HiAD is altered compared to mice on Chow. How do these collagen networks compare to mice on HiAD+PM, HiAD+ALT or HiAD+AGER1 (mutant)? Presumably, ALT, PM and AGER1 revert the network back to a more normal (Chow-like) state. The authors should also show how the collagen network in patients with NASH/T2DM compares to NASH and healthy patients (Fig. 1).
3. In Fig. 5c, the cortactin stain appears to be localized to the entire cell rather than invadopodia. Since cortactin also participates in focal adhesions and cell-cell junctions, the authors should stain for a more specific marker of invadopodia, such as Tks5, which they used in a previous publication (PMID: 30297715).
4. Additional experiments are required to highlight the importance of tensin-1 in HCC progression (identified in Fig. 4c) and strengthen the model presented in Fig. 5s:
 - a. How does tensin-1 knockout affect HCC foci formation in vivo?
 - b. Does tensin-1 knockout affect YAP localization, CTGF mRNA levels, cell circularity and proliferation (Fig. 5b-j)?
 - c. It is unclear how tensin-1 modulates RhoA/ROCK activity in the model presented in Fig. 5s. HCC cells are known to have elevated RhoA activity because of downregulated DLC-1 (deleted in liver cancer) expression (PMID: 18519636). Therefore, RhoA activity in HCC cells exposed to low and high viscoelasticity should be quantified. The effect of integrin β 1 knockdown and tensin-1 knockout on RhoA activity should also be quantified.

d. The proximity ligation assay (PLA) used in Fig. 5o,p is an excellent way to demonstrate that integrin β 1 and tensin-1 interact within cells; however, this technique can generate non-specific signals due to the amplification method used to detect interactions. The authors should include control stains (e.g., 1^o antibody only, 2^o antibody only) in Extended Data Figures to strengthen the claim that integrin β 1 and tensin-1 interact.

e. Tensin-1 has a low binding affinity for integrin (PMID: 17473008) and its localization to focal adhesions is largely influenced by other factors such as Src (PMID: 11493667). Therefore, the authors should demonstrate that tensin-1 interaction with integrin β 1 is critical to the mechano-signaling induced by viscoelasticity (e.g., using a tensin-1 mutant that interacts with actin but cannot localize to focal adhesions (PMID: 12495434)).

f. Because the major pathway is integrin β 1, tensin, actin, Rho, and YAP, were Rho pathway-related genes differentially expressed in the RNA seq analysis in Fig. 4?

g. The authors should investigate whether a second HCC cell line or even better (if possible) a human PDX specimen also relies on integrin β 1, tensin-1 and RhoA mechano-signaling.

h. The authors should verify their data obtained with an integrin β 1 blocking antibody with integrin β 1 knockdown.

Together, these experiments would reinforce the authors' claim that viscoelasticity initiates an intracellular signaling cascade involving integrin β 1, tensin-1 and Rho/ROCK/myosin-II to modulate Hippo pathway signaling.

5. In light of the authors' findings showing that HiAD alone increases liver viscoelasticity (e.g., Fig. 1o,p) but has little effect on development of tumor foci (e.g., Fig. 2c), the authors should soften their title by replacing "drives" with "promotes or accelerates" liver cancer progression.

6. The statistics and error bars for each panel should be checked. For example, the data for "vehicle" in Fig. 1m do not appear to have a normal distribution. Yet, the authors state that statistical comparison was performed using ANOVA. Do the authors mean ANOVA after log transformation? Also, data should be represented by SD instead of SEM.

Minor/specific points:

1. Individual data points for all graphs should be shown (instead of bar graphs).
2. In Fig. 2f, how did the mice on FFD diet die if not due to HCC foci?
3. Extended Data Fig. 2h is not described in the figure legend.
4. The red 'Bundler' signal in Extended Data Fig. 4 is not visible to color-blind readers. The authors may want to change the color to magenta or green.
5. It is unclear what the authors mean by "active" YAP since an antibody for total YAP is used. The authors are suggested to present the relevant data as nuclear:cytoplasmic YAP ratio.
6. Was tensin-1 knocked down or knocked out? The text and relevant figure legends state "TNS1 knockdown", whereas the materials and methods describe CRISPR/Cas9 knockout with a guide RNA.
7. Since tensin-1 has been shown to affect focal adhesion dynamics (PMID: 19826001), vinculin may not be the most appropriate loading control in Fig. 5q.
8. Line 133: the authors may want to replace "improved viscoelasticity" with "restored viscoelasticity"
9. The authors should correct a typo in line 174, where they wrote "...we used integrin β 1-specific blocking antibodies, and this reduced YAP activation (Fig. 5h), cell circularity (Fig. 5i and Extended Data Fig. 7a), and proliferation (Fig. 5j)". However, the graphs show that cell circularity is

increased.

10. Why was collagen pre-incubated with BSA or AGEs at 4°C for 4 weeks in Fig. 3c? Would it be better to be consistent with respect to time duration and temperature for the mice experiments in Fig. 2d?

Referee #2 (Remarks to the Author):

In this work, the authors document that liver tissue becomes more viscoelastic in certain pathological conditions, notably NASH and type II diabetes. This is attributed to the accumulation of advanced glycation endproducts (AGEs). Moreover, this change in material properties has functional consequences for disease biology. In particular, extracellular matrix (ECM)/substrates with viscoelastic properties permit stronger activation of YAP1 function and subsequent cancer growth. The work spans a nice range of human tissue analysis, mouse models, in vitro cell biology, and computational modeling. Thus, it is a well-constructed study that makes clear point about the importance of considering the viscous properties of tissue and goes beyond the current simplistic view that increasing elastic modulus correlates with aggressive cancer biology. However, the authors don't sufficiently analyze how the inter-relationship between elastic and loss moduli determines the formation of mechano-responsive integrin containing complexes. Judging the suitability of this study for Nature is not straightforward. The role of viscoelastic properties, as opposed to purely elastic, is increasingly appreciated (including in previous work by the authors – Adebawale et al 2021), but this study makes the point clearly and links to disease pathology in a powerful way. It is also well-conducted and combines multiple approaches elegantly. However, the mechanistic angle of why viscoelastic deformation boosts 'mechano-responses' such as YAP is under-developed, nor is the intersection between varying elastic and viscoelastic properties of the substrate/matrix fully explored.

Specific points

1. Throughout the manuscript the quality of the myc-tag staining is sub-optimal. In some panels, the signal that the authors claim is visible, but in others it is not. Technical improvement is required.
2. Figure 3b&c should be quantified using CT-FIRE or similar tools.
3. A more detailed mapping of the relative location of the thicker collagen bundles in HiAD mice and other liver cells types would be very interesting. In particular, are the bundles associated with hepatocytes (potentially with nuclear YAP1) or activated hepatic stellate cells.
4. Figure 3d – the reconstituted collagen matrix is pretty soft. The authors should exclude the effect of ALTs on collagen that has a higher starting elastic modulus.
5. Can the authors clarify if the collagen used in in vitro assays is telopeptide intact or pepsin digested? I could not find this information, only the supplier name. Hopefully, they have used the telopeptide intact form as it is more physiologically relevant.
6. Although the authors report that AGE does not affect the elastic modulus of collagen when using bulk measurement methods, they should also explore if it leads to increased heterogeneity in local elastic modulus by AFM mapping. The collagen images in Figure 3c suggest that the mechanical heterogeneity of the matrix is likely to be changed. It may be that viscoelastic properties of the collagen network allow higher elastic moduli to develop in focal patches. Related to this, are the filopodia documented later in the paper preferentially adhering to the larger collagen bundles?
7. LOX enzymes and crosslinking have been reported to confer elastic properties on collagen gels (e.g Petrie et al 2012). Does LOX-mediated crosslinking counteract the effect of ALT? More generally, some consideration of how other classes of collagen modifying enzymes, including MMPs, change during NASH and type II diabetes would be welcome.
8. The authors should use their nice inter-penetrating compound matrix system to vary both elastic and loss moduli in a systematic manner. The expectation is that if the loss modulus is kept low and unchanging, then increasing the elastic modulus should lead to increased YAP activity.

9. The authors refer to active YAP immune-fluorescence – what do they mean? Is it total YAP staining, followed by evaluation of its nuclear vs cytoplasmic localization? Also, the staining in Figure 5b is not convincing – the brightness of the nuclear signal does not seem consistent with a translocation of the cytoplasmic signal into the nucleus.

10. The authors should measure proliferation using EdU staining or measuring changes in cell number, not Ki67, in the in vitro assays in Figure 5.

11. The invadopodia staining is not convincing. There is no evidence of matrix proteolysis, which is central to the definition of invadopodia - cortactin puncta are not equivalent to invadopodia. It would be more informative to show higher magnification images of integrin-mediated adhesions. Staining using conformation specific active integrin β 1 antibodies would be informative, as would staining for focal and fibrillar adhesion components and phosphorylated myosin light chain.

12. The tensin analysis in Extended Data Figure 7 is not convincing. The result is marginal and it appears that the authors have selected their cut-off to maximize the difference. They should choose more objective criteria for their cut-off (such as top and bottom quartiles or above and below the median) and present analysis for more than one cohort.

Referee #3 (Remarks to the Author):

This study is showing an additional evidence of ECM regulation of HCC development, particularly in the condition of T2DM. The authors first determined that hepatic AGE levels are increased in NASH patients with T2DM, which is associated with increased viscoelasticity in the liver. Then, the study showed high AGE diet increased hepatic AGEs along with viscoelasticity without increased stiffness. The importance of AGE and AGE-mediated signaling in NASH-HCC development was also validated in the HCC animal model. Subsequently, the authors determined that the AGE-collagen crosslink regulates viscoelasticity. Additionally, the study determined that integrin β 1-TNS1-YAP signaling is crucial for AGE-mediated HCC development. Overall, the experimental approaches the study used are highly mechanistic. The data provided supported their hypothesis. Because in addition to cirrhotic background, NAFLD-HCC is also known to develop in non-cirrhotic livers, this study that determined the critical pathways that are activated in non-cirrhotic liver, especially T2DM, which promotes HCC development is highly significance. However, there are several gaps, which need to be determined.

Specific comments:

1. Figure 1b. Liver AGE levels was increased in NASH/T2DM, but not in NASH only. What is hepatic AGE level in T2DM without NAFLD (NAFL + NASH)? This is also an important condition. If this is altered, hepatic AGE and/or viscoelasticity (Fig.1e) is altered prior to or without NASH development.

2. Not only liver AGE levels, the AGE levels of tumor lesions with and without T2DM should also be evaluated.

3. Is viscoelasticity associated with AGEs, independent of collagen deposition? In NASH-HCC with cirrhosis, does AGE-mediated viscoelasticity is also a crucial factor for promoting HCC development? Is this the same mechanism between NASH-HCC with and without cirrhosis? Or in cirrhosis case, does stiffness play more important role in activating tumor-promoting cellular signaling? Or is still AGE and viscoelasticity important? Because ALT treatment has a protective effect, collagen should play some roles. Also, ALT treatment reduced hepatic AGE level (Fig.1m). It is unclear how the AGE-collagen crosslink regulates hepatic AGE level.

4. Figure 2 analyzed the role of AGE in NASH-HCC development. The study measured AGE levels in liver tissues (which could include both tumor lesions and non-tumor livers). This reviewer is curious about both AGE and collagen levels in tumor lesions. Are AGE and collagen in HCC lesion increased compared with those in non-tumor livers? This reviewer wondered whether AGE and AGE-collagen crosslink play a role only in HCC lesions or are also important in non-tumor livers that affect to HCC development and growth by unknown secreting factors from hepatocytes. Does

High AGE diet increase collagen content in HCC lesion? Is collagen produced from activated HSCs or pre-activated HSCs?

5. Fig.4f-g, the effect of dnTEAD2 in YAP activity in HCC lesion should be shown. Also, the assessment of YAP downstream targets is required.

6. Fig.4-5. The study showed CTGF as a YAP regulated gene. More comprehensive YAP signaling assessment is required - more markers and more intermediate signaling molecular states; whether integrin and TNS1 inhibition reduces YAP activity more clearly. Fig.5q showed "active YAP". However, this is unclear. Is this either phosphorylated YAP or nuclear total YAP?

7. The study nicely showed the effect of Itgb1 and TNS1, but in vivo evidence of Itgb1, TNS1, and YAP needs to be validated. Also, it is unclear how integrin b1 is activated. Is integrin b1 activated by collagen, AGE, or crosslinked AGE-collagen?

Author Rebuttals to Initial Comments:

Referee expertise:

Referee #1: mechanobiology in cancer (response starts from page 2)

Referee #2: ECM, cell migration, cancer (response starts from page 14)

Referee #3: NASH, liver cancer (response starts from page 25)

Response summary:

Thank you very much for the reviewers comments. All were important and well-taken, and we believe performing these studies significantly enhanced the manuscript. First, we would like to summarize the main studies performed. The detailed point-by-point responses are to follow.

- More mechanistic studies were done on invadopodia formation/imaging, in relation to integrin $\beta 1$, and TNS1 (Fig. 5, Extended Data Figs. 9, 10, and 11).
- We performed experiments on TNS1 interaction with integrin $\beta 1$ showing its central role in mechano-signaling induced by viscoelasticity (Ext. Data Fig. 12).
- We generated an in vivo experimental model of TNS1 KD to study tumor formation (Fig. 6).
- More detailed studies on mechano-signaling involving integrin $\beta 1$, and TNS1-dependent RhoA activation were performed (Fig. 5r).
- Imaging and mapping studies were conducted on collagen bundles (Ext. Data Fig. 3) in relation to liver cells (Ext. Data Fig. 7).
- Studies on the mechanical heterogeneity of the matrix by AFM (Ext. Fig. 4), and invadopodia in relation to collagen bundles in the viscoelastic matrix were performed (Fig. R3).
- We conducted experiments using the inter-penetrating compound matrix system and varying both the elastic and loss moduli (Ext. Data Fig. 8).
- Data regarding AGE content, stiffness and viscoelasticity in T2DM patients were added (Fig. 1).
- We have included data regarding the DN-TEAD2 animal model, including active YAP, and downstream targets.
- Experiments were performed on tumor vs. non-tumor areas to assess AGE and collagen content.
- We improved the myc and GS immunohistochemistry/images, including verification of the active/total YAP by western blots and all the controls for the experiments, using antibodies.

Referee #1 (Remarks to the Author):

In this manuscript, Fan et al. investigate the impact of nonalcoholic steatohepatitis (NASH) and type 2 diabetes mellitus (T2DM) on hepatocellular carcinoma (HCC) disease progression. The authors demonstrate that livers of patients with NASH/T2DM compared to those without T2DM exhibit increased viscoelasticity due to the accumulation of advanced glycation end products (AGEs). AGEs alter the crosslinking of collagen fibers within the ECM leading to enhanced ECM viscoelasticity and HCC progression. Inhibiting AGEs production with PM or breaking the formation of AGEs-collagen crosslinks with ALT or reconstituting the clearance receptor AGER1 reverses the changes in viscoelasticity, diminishes HCC growth and improves survival. The authors also suggest that HCC cells respond to altered ECM viscoelasticity via integrin β 1, tensin-1 and YAP. Overall, this is an interesting manuscript, which integrates different animal models with sophisticated in vitro experiments and simulations. The authors neatly demonstrate that viscoelasticity enhances liver cancer progression. However, additional work is required to reinforce their proposed mechanistic findings centering on the involvement of integrin β 1, tensin-1 and YAP.

Thank you very much for the positive comments. We added further experimental data to address the reviewer's questions, and enhance the quality.

General comments:

1. In Figs. 2h-n, the authors show that AAV8-mediated AGER1 delivery lowers AGEs and stress relaxation, and reduces the growth of tumor foci, presumably due to AGER1 reconstitution. The authors should also provide the expression levels of AGER1 in mice injected with AAV8-TBG-AGER1 and AAV8-TBG-GFP (control).

Thanks for the comment. We have included the RT-qPCR data, showing the expression of AGER1 following AAV8-TBG-AGER1 or AAV8-TBG-GFP injection (included in Fig. 2i).

Figure 2i: AGER1 was reconstituted by AAV8-TBG-AGER1 delivery, and liver AGER1 has increased (n=5, mean±SEM, **P<0.01, ANOVA).

2. In Fig. 3, the authors show that the collagen network in mice on HiAD is altered compared to mice on Chow. How do these collagen networks compare to mice on HiAD+PM, HiAD+ALT or HiAD+AGER1 (mutant)? Presumably, ALT, PM and AGER1 revert the network back to a more normal (Chow-like) state. The authors should also show how the collagen network in patients with NASH/T2DM compares to NASH and healthy patients (Fig. 1).

We have performed experiments visualizing collagen networks (SHG microscopy images) on livers from healthy controls, patients with NASH, and NASH+T2DM. These show longer fiber networks in control and NASH patients with no T2DM, whereas in patients with NASH/T2DM, the fibers exhibit bundle formation (white arrow, Extended Data Fig. 3a).

SHG images in mice on HiAD, HiAD+PM, and HiAD+ALT, as well as in the HiAD+AGER1 groups depict collagen networks with more collagen bundling seen in the HiAD+vehicle group (Ext. Data Fig. 3b).

Extended Data Figure 3. Collagen networks exhibit a more bundled appearance in patients with NASH/T2DM, or in mice on HiAD

a. Collagen fibers in human liver samples were analyzed by second harmonic generation (SHG) microscopy. Bundling is visible in the NASH+T2DM group (white arrow). Red arrow point to a lipid droplet.
b. In mice on HiAD+vehicle, there is a more bundled appearance of the collagen network (white arrow) Maximum intensity projection Z-stack for section thickness of 30 μm (Scale bar, 200 μm).

3. In Fig. 5c, the cortactin stain appears to be localized to the entire cell rather than invadopodia. Since cortactin also participates in focal adhesions and cell-cell junctions, the authors should stain for a more specific marker of invadopodia, such as Tks5, which they used in a previous publication (PMID: 30297715).

We agree with this important comment, and we have obtained the Tks5-mNeonGreen and Tks5- mScarlet plasmids (from Dr. L Hodgson, Albert Einstein College of Med), and transfected them (vs. control) into HCC cells, for all invadopodia analyses. Transfected cells were embedded in hydrogels with low or high viscoelasticity. Analyses of signals for Tks5 and other functional markers were performed, including MT1-MMP that signifies proteolytic activity (Fig. 5d). Additionally, we show active integrin β1 (Mouse monoclonal 12G10, Abcam), and p-MLC2(Ser19, Cell Signaling Technology#3671) immunofluorescence studies in low and high viscoelasticity conditions (Extended Data Fig. 9). High-resolution imaging using the ZEISS LSM 980 with Airyscan 2 depicted invadopodia that were positive for active integrin β1, and p-MLC2, image below.

Fig. 5d. Invadopodia formation in high viscoelasticity hydrogels.

Invadopodia were analyzed after transfecting cells with the Clontech-N1 plasmid containing human Tks5-mNeonGreen (green). Immunofluorescence analysis for MT1-MMP (red) and Airyscan microscopy depict the signals (d, scale bar, 10 μm).

Extended Data Fig. 9: Huh 7 cells cultured in high viscoelasticity hydrogels form invadopodia.

a. Schematic depiction of invadopodia. **b-d.** Huh 7 cells were transfected with Tks5-mNeonGreen, and immunofluorescence microscopy was performed using an antibody against MT1-MMP (**b**, showing all channels of 5d), active integrin $\beta 1$ (**c**), and p-MLC2 (**d**). Cells were imaged by high resolution Airyscan (LSM980, Zeiss) microscopy. (Scale bar, 10 μm).

4. Additional experiments are required to highlight the importance of tensin-1 in HCC progression (identified in Fig. 4c) and strengthen the model presented in Fig. 5s:

a. How does tensin-1 knockout affect HCC foci formation in vivo?-

We thank the reviewer for the excellent comment and questions. To strengthen the data regarding the role of tensin 1, we used a CRISPR/Cas9-mediated in vivo approach whereby TNS1 is targeted at the time of hydrodynamic injection (PMID: 36455783 DOI: 10.1016/j.jhep.2022.10.037). Mice on Chow or HiAD were hydrodynamically injected as described in Fig. 2., as well as with the sgTNS1 or control guide RNA plasmids. Mice were sacrificed 7 weeks following the injections (**Fig. 6a**). Data show that in sgTNS1-injected mice the expression of TNS1 (**Fig. 6b**), and the number of transformed foci (myc and GS+) have significantly decreased (**Fig. 6c, d**). The expression of YAP targets CTGF and Cyr61 has decreased significantly, as well (**Fig. 6g**).

Figure 6: TNS-1 knockdown decreases formation of transformed foci after HDI

a. Schematic presentation of the in vivo targeting of TNS1 by CRISPR/Cas in conjunction with hydrodynamic injection. Mice were fed chow or HiAD for 7w, then hydrodynamically injected with pT3-EF5a-hMet/ pT3-EF5a-S45Y- β -catenin-myc (mutant β -catenin), with the sleeping beauty (SB) transposase, as well as the CRISPR-Cas9-based vector linking two sgRNAs targeting mouse TNS1(pX333-sgTNS1) or empty vector (sgNC). Mice were sacrificed 7 weeks following injection. **b.** TNS1 expression was analyzed by RT-qPCR. (n=6, mean \pm SEM; **p < 0.01, ****p < 0.0001, ns not significant, ANOVA, post-hoc Tukey test) **c, d.** GS/myc immunohistochemistry depict the signals (**e**) and quantification (**d**) of GS/myc positive foci. There was an increase in foci in control sgRNA-injected mice, and less foci were seen after sgTNS1. Scale bar, 300 μ m. (n=6, mean \pm SEM; *p < 0.05, **p < 0.01, ANOVA, post-hoc Tukey test) **e.** The expression of YAP targets gene CTGF and Cyr 61 was analyzed by RT-qPCR. (n=6, mean \pm SEM; *p < 0.05, **p < 0.01, ***p < 0.001 ANOVA, post-hoc Tukey test). **f.** Schematic presentation of TNS1 that serves as a key component of the ECM mechano-sensor complex by binding to integrin $\beta 1$ in high viscoelasticity ECM conditions. The illustration was created using BioRender.com.

b. Does tensin-1 knockout affect YAP localization, CTGF mRNA levels, cell circularity and proliferation (Fig. 5b-j)?

We present new data showing that tensin-1 KD decreases nuclear, active YAP and proliferation. In addition, YAP target genes CTGF and Cyr61 expression were significantly reduced in tensin 1 KD cells cultured in high viscoelasticity hydrogels (**Extended Data Fig. 10, and Fig. 5n-q**).

For comment **4e**, we have used a PTB domain deleted TNS1 (dd-TNS1) transfection approach, and observed similar results with a decrease in proliferation and expression of YAP target genes in the dd-TNS1-transfected cells (**Ext. Fig. 12, see later**).

Extended Data Fig. 10. Tensin-1 or integrin β 1 knockdown decrease proliferation, active nuclear YAP, invadopodia formation and improve cell circularity (Additional Data to main Fig. 5).

Upper Panel: Huh7-Cas9 cells were transfected with plasmids containing CRISPR guide RNA for TNS1 (sg-TNS1), or integrin β 1 (sg-Itg β 1) or control sgRNA (NC), and cells after 24 hours were embedded in low or high viscoelasticity hydrogels. After 48 hours in 3D cultures, cell proliferation was evaluated by Edu nuclear signal (**b**, Scale bar, 50 μ m). YAP activity was analyzed by immunofluorescence using active YAP antibody (**c**, Scale bar, 20 μ m), and cell circularity was evaluated by actin (**d**, scale bar, 20 μ m). Invadopodia were studied by TKS5 signal after transfection (**e**, scale bar, 10 μ m).

Lower panel: main Fig. 5 n-q: Proliferation (**n**), YAP activation (% cells with active nuclear YAP, **o**), and its targets CTGF and Cyr61 (**p**) significantly decreased after TNS1 and integrin β 1 KDs, in high viscoelasticity hydrogels while cell circularity (**q**) has improved (n=5, mean \pm SEM; *p<0.05, **p<0.01, ***p<0.001, ****p<0.0001, ANOVA, post-hoc Tukey test).

c. It is unclear how tensin-1 modulates RhoA/ROCK activity in the model presented in Fig. 5s. HCC cells are known to have elevated RhoA activity because of downregulated DLC-1 (deleted in liver cancer) expression (PMID: 18519636). Therefore, RhoA activity in HCC cells exposed to low and high viscoelasticity should be quantified. The effect of integrin $\beta 1$ knockdown and tensin-1 knockout on RhoA activity should also be quantified.-

Thanks for this comment. To address RhoA activity in low/high viscoelasticity conditions, we performed RhoA pull-down assays to detect the GTP-bound active form (Fig. 5r). High viscoelasticity ECM increased RhoA-GTPase activity in cells, in a TNS1- and integrin $\beta 1$ -dependent manner. P-LATS1 (an important negative regulator of YAP) was reduced, while active non-phosphorylated YAP increased in high viscoelasticity hydrogels in a TNS1 and integrin $\beta 1$ -dependent way.

Fig. 5r Integrin $\beta 1$ -Tensin 1-YAP axis mediates viscoelasticity-specific mechano-cellular pathways

RhoA GTPase activity was studied in low/high viscoelasticity ECM in control sgRNA (NC), or sg-TNS1 or sg-Integrin $\beta 1$ transfected cells. GTP-RhoA increased in high viscoelasticity hydrogels, but lower activity was seen after TNS1 or integrin $\beta 1$ KDs. Lower phospho-LATS1, and higher active (non-Phosphorylated YAP) signals depict induction of YAP, in a TNS1 or integrin $\beta 1$ -mediated manner, in high viscoelasticity ECM (representative immunoblot).

d. The proximity ligation assay (PLA) used in Fig. 5o,p is an excellent way to demonstrate that integrin $\beta 1$ and tensin-1 interact within cells; however, this technique can generate non-specific signals due to the amplification method used to detect interactions. The authors should include control stains (e.g., 1 $^\circ$ antibody only, 2 $^\circ$ antibody only) in Extended Data Figures to strengthen the claim that integrin $\beta 1$ and tensin-1 interact.

Thanks for this comment. We included the antibody controls in the **Extended Data Figs 10a**. These do not show visible non-specific signal.

Extended Fig. 10a. Antibody validation

a. control (1 $^\circ$ antibody only, 2 $^\circ$ antibody only) for PLA assays. Scale bar, 10 μ m., blue: DAPI

e. Tensin-1 has a low binding affinity for integrin (PMID: 17473008) and its localization to focal adhesions is largely influenced by other factors such as Src (PMID: 11493667). Therefore, the authors should demonstrate that tensin-1 interaction with integrin $\beta 1$ is critical to the mechano-signaling induced by viscoelasticity (e.g., using a tensin-1 mutant that interacts with actin but cannot localize to focal adhesions (PMID: 12495434)). To address this question, we obtained the PTB domain-deleted TNS1 (dd-TNS1) from Dr. Su-Hao Lo (UC Davis). Deleting the PTB domain prevents tensin-1 binding to integrin $\beta 1$ NPXY motif, thus the activation/function of the molecular clutch (PMID: 33597154; 17473008), however the actin binding site remains intact. To analyze whether dd-Tensin1 can affect invadopodia formation and proliferation, we transfected cells with a vector containing the mutant or full-length TNS1 (Ext. Data Fig. 12a) and embedded them in slow or fast-relaxing hydrogels. We then studied whether dd-TNS1 can bind to integrin $\beta 1$, by PLA. In dd-TNS1 transfected cells, the PLA signal was very low, whereas in the full-TNS1 (tomato) positive cells, the signals co-localized (Ext. Data Fig. 12b, c: counting). Dd-TNS1-transfected cells in fast relaxing hydrogels had lower proliferation (Ext. Data Fig. 12d, h), YAP activation and its target genes (Ext. Data Fig. 12e, j, and k), were unable to form invadopodia (Ext. Data Fig. 12g), and their circularity improved (Ext. Data Fig. 12j).

Extended Data Fig. 12. Huh-7 cell transfected with the domain deleted (dd) TNS-1 construct did not show tensin-1 colocalization with integrin, exhibited lower proliferation, YAP activity and invadopodia formation in high viscoelasticity hydrogels.

a. Schematics of the PTB domain deleted (dd)-TNS1 construct. This prevents binding to integrins however the actin-binding domain remains intact.

b, c. Proximity ligation assay (PLA, **b**) to assess integrin $\beta 1$ and TNS1 binding (green) in empty vector (NC), full length TNS1, and dd-TNS1 transfected cells (red, tdTomato). In full length TNS1 tomato and PLA signals colocalized whereas no colocalization was seen in dd-TNS1 transfected cells, or in low viscoelasticity matrix. PLA positive dots (**c**) were quantified from 30 cells in 5 gels, each group(m), (n=5, mean \pm SEM, ****p<0.0001, ANOVA, post-hoc Tukey test).

d-k. Cell proliferation was evaluated by Edu (**d**, Scale bar, 100 μ m.) and quantification (**h**). YAP activity was analyzed by active YAP immunofluorescence (**e**, Scale bar, 50 μ m.), quantification (**i**), and YAP-regulated target genes mRNA expression (**k**). Cell circularity was analyzed by actin (**f**, Scale bar, 50 μ m), and Image J analysis (**j**). Invadopodia were studied by TKS5 signal in transfected cells (**g**, Scale bar, 10 μ m). (n=5, mean \pm SEM; **P < 0.01, ***p < 0.001 ****p < 0.0001, ANOVA, post-hoc Tukey test).

f. Because the major pathway is integrin $\beta 1$, tensin, actin, Rho, and YAP, were Rho pathway-related genes differentially expressed in the RNA seq analysis in Fig. 4?-

Interestingly, we found that other RhoA-associated targets such as SOX9 and c-Jun were induced in HiAD, but not after inhibition of AGE production (PM) or crosslinking (ALT). (**Reply to reviewer Fig. R1**). These results are intriguing, as Sox9 plays a role in cell plasticity, and fate determination in HCC (Liu et al, Hepatology. 2016 Jul;64(1):117-29). C-Jun on the other hand, plays a role in inflammation-driven HCC (such as in NASH), by the activation of c-Jun N-terminal kinase (JNK) (Yang et al Semin Liver Dis. 2019 Feb;39(1):26-42), and can promote tumor stemness (Kuo, Stem Cells. 2016 Nov;34(11):2613-2624.). Thus, RhoA activation could be linked to other targets involved in hepatocarcinogenesis, which could be a focus of future studies.

Reply to reviewer Fig. R1: RhoA-associated target genes

Analyses of RhoA-associated target genes in bulk RNA-seq data from mice fed chow, FFD, and HiAD diets. Mice on HiAD were treated with PM injection and a group of mice with RAGE hepatocyte depletion (RAGE^{HepKO}) was studied. (n=3, mean \pm SEM; *p < 0.05 **p < 0.01, ***p < 0.001, ANOVA, post-hoc Tukey test).

g. The authors should investigate whether a second HCC cell line or even better (if possible) a human PDX specimen also relies on integrin $\beta 1$, tensin-1 and RhoA mechano-signaling.

We repeated the key experiments using the Hep3B cell line, and data regarding cell proliferation, YAP activation, and invadopodia formation were consistent with those obtained from the Huh7 cell line (Ext Data Fig. 11).

h. The authors should verify their data obtained with an integrin $\beta 1$ blocking antibody with integrin $\beta 1$ knockdown. –

We have generated integrin $\beta 1$ knockdown by CRISPR/Cas9, transfected Huh7 (Extended Data Fig. 10, and Fig. 5n-q; see above), and Hep3B cells (Extended Data Fig. 11; see above), and cultured them in low and high viscoelasticity conditions. Invadopodia formation was much decreased in sg-Integrin $\beta 1$ transfected cells, and cell circularity increased. Integrin $\beta 1$ KD reduced nuclear active YAP, downstream signals (CTGF and Cyr61), as well as cell proliferation as assessed by Edu.

Together, these experiments would reinforce the authors' claim that viscoelasticity initiates an intracellular signaling cascade involving integrin $\beta 1$, tensin-1 and Rho/ROCK/myosin-II to modulate Hippo pathway signaling.

5. In light of the authors' findings showing that HiAD alone increases liver viscoelasticity (e.g., Fig. 1o,p) but has little effect on development of tumor foci (e.g., Fig. 2c), the authors should soften their title by replacing "drives" with "promotes or accelerates" liver cancer progression.

Thanks for this comment, we changed the wording. In Fig. 2c the control groups had the control non-mutated vector injected, thus developed significantly less tumor foci in contrast to mutant vector-injected mice (transformed foci >20+ cells, encircled).

Fig. 2. Mice on HiAD develop more transformed foci following hydrodynamic injection, and exhibit AGEs-dependent higher viscoelasticity.

a. Schematics of the NASH-related hepatocellular carcinoma (HCC) models. Mice were fed for 7 weeks either chow, FFD or HiAD. Hydrodynamic injection (HDI) was performed using vectors expressing human MET gene (pT3-EF5a-hMet-V5) and the sleeping beauty (SB) transposase combined with a vector expressing either wild-type human β -catenin (pT3-EF5a- β -catenin-myc, control group) or mutant pT3-EF5a-S45Y- β -catenin-myc, mutant group). Four weeks after HDI, mice were sacrificed for analysis.

b. Immunohistochemistry, co-localization of glutamine synthetase (GS) and myc-tag positive foci (more than 20 cells are considered forming a focus (circles)). Scattered cells denote transduced cells (arrows). GS at baseline marks pericentral cells. Scale bar, 300 μ m.

c. Quantification of GS and myc positive foci. (n=5, mean \pm SEM; ***p<0.001, ns not significant, ANOVA, Tukey's post hoc).

6. The statistics and error bars for each panel should be checked. For example, the data for "vehicle" in Fig. 1m do not appear to have a normal distribution. Yet, the authors state that statistical comparison was performed using ANOVA. Do the authors mean ANOVA after log transformation? Also, data should be represented by SD instead of SEM.

Thanks for this comment. We added the error bars for each panel. Human or mouse data can have a not normal distribution, as in Fig 1m. We implemented a non-parametric statistical test (Wilcoxon's rank sum test) to

compare the distributions of the fold-change in AGEs across the different diet/treatment groups. This test requires no stringent distributional assumptions. The interpretation and statistical significance of the comparisons presented in Fig 1m remain the same (**Reply to reviewer Fig. R2**). We have also repeated both tests on the data sets in Fig. 1, and the overall interpretation and statistical significance of the comparisons were the same with the non-parametric statistical test.

We did not use log transformation because our samples are not sampled from a lognormal distribution. In our analyses, we largely focus on differences in the mean values across groups (e.g. hence the use of ANOVA), and thus we felt that providing the SEM rather than the SD (which quantifies the dispersion in the overall set of observations), was more appropriate.

Reply to reviewer Fig. R2: Parametric or non-parametric statistical tests performed on Fig. 1m datasets

Fig 1m was analyzed by parametric statistical test (ANOVA plus Tukey's post hoc, left) and a non-parametric statistical test (Wilcoxon's rank sum test, right).

Minor/specific points:

1. Individual data points for all graphs should be shown (instead of bar graphs).

Thanks for the comment, we changed these.

2. In Fig. 2f, how did the mice on FFD diet die if not due to HCC foci? One mouse out of 8 died in the FFD group. Based on the heart size at necropsy, it most likely had a cardiac complication.

3. Extended Data Fig. 2h is not described in the figure legend.

Thanks for noting, we corrected this.

4. The red 'Bundler' signal in Extended Data Fig. 4 is not visible to color-blind readers. The authors may want to change the color to magenta or green.

Thanks, we changed the color to magenta.

5. It is unclear what the authors mean by "active" YAP since an antibody for total YAP is used. The authors are suggested to present the relevant Data as nuclear:cytoplasmic YAP ratio.

We used an antibody against active YAP that recognizes the non-phosphorylated, active form (Recombinant Anti-active YAP1 antibody [EPR19812], Abcam, #ab205270). To confirm our data, we also present studies with western blots and nuclear/cytoplasmic YAP (**Suppl. Data Fig. 2**).

Suppl. Data Fig. 2. Antibody validation

a. Nuclear and cytoplasmic YAP was assessed using antibodies against active, non-phosphorylated and inactive phosphorylated YAP in western blots, in low and high viscoelasticity hydrogels (cytoplasmic and nuclear fractions).

6. Was tensin-1 knocked down or knocked out? The text and relevant figure legends state "TNS1 knockdown", whereas the materials and methods describe CRISPR/Cas9 knockout with a guide RNA.-

In our protocol, TNS1 guide RNAs were transfected into Huh7 or Hep3B cells with continuous Cas9 protein expression (generated by lentivirus). The transfected cells were embedded in low or high-viscoelasticity hydrogels 24 hours after transfection. Since the TNS1 knockdown efficiency may vary, the term "CRISPR/Cas9 mediated knockdown" is more appropriate in this context. We clarified this in the Methods section.

7. Since tensin-1 has been shown to affect focal adhesion dynamics (PMID: 19826001), vinculin may not be the most appropriate loading control in Fig. 5q.

Thanks for the comment. We have now included GAPDH as control.

8. Line 133: the authors may want to replace "improved viscoelasticity" with "restored viscoelasticity"-

This was corrected.

9. The authors should correct a typo in line 174, where they wrote "...we used integrin β 1-specific blocking antibodies, and this reduced YAP activation (Fig. 5h), cell circularity (Fig. 5i and Extended Data Fig. 7a), and proliferation (Fig. 5j)". However, the graphs show that cell circularity is increased.

Thanks, we corrected this.

10. Why was collagen pre-incubated with BSA or AGEs at 4°C for 4 weeks in Fig. 3c? Would it be better to be consistent with respect to time duration and temperature for the mice experiments in Fig. 2d?-

The collagen (telopeptide intact form collagen, Product Number354236, Corning® Collagen I, Rat Tail) and AGEs (Glycolaldehyde-AGE-BSA) we used, are not stable at 37°C, in vitro. Based on the literature and our pre-test, we incubated AGEs with non-neutralized collagen at 4°C for 4 weeks to generate AGEs-crosslinked-collagen for hydrogel assays, and to reproduce the collagen network's dynamic re-organization.

Referee #2 (Remarks to the Author):

In this work, the authors document that liver tissue becomes more viscoelastic in certain pathological conditions, notably NASH and type II diabetes. This is attributed to the accumulation of advanced glycation endproducts (AGEs). Moreover, this change in material properties has functional consequences for disease biology. In particular, extracellular matrix(ECM)/substrates with viscoelastic properties permit stronger activation of YAP1 function and subsequent cancer growth. The work spans a nice range of human tissue analysis, mouse models, in vitro cell biology, and computational modeling. Thus, it is a well-constructed study that makes clear point about the importance of considering the viscous properties of tissue and goes beyond the current simplistic view that increasing elastic modulus correlates with aggressive cancer biology. However, the authors don't sufficiently analyze how the inter-relationship between elastic and loss moduli determines the formation of mechano-responsive integrin containing complexes. Judging the suitability of this study for Nature is not straightforward. The role of viscoelastic properties, as opposed to purely elastic, is increasingly appreciated (including in previous work by the authors – Adebowale et al 2021), but this study makes the point clearly and links to disease pathology in a powerful way. It is also well-conducted and combines multiple approaches elegantly. However, the mechanistic angle of why viscoelastic deformation boosts 'mechano-responses' such as YAP is under-developed, nor is the intersection between varying elastic and viscoelastic properties of the substrate/matrix fully explored.

We thank the reviewer for the review and constructive critiques. We have addressed all the comments as described below.

Specific points

1. Throughout the manuscript the quality of the myc-tag staining is sub-optimal. In some panels, the signal that the authors claim is visible, but in others it is not. Technical improvement is required.-

Thanks for this comment. We realize that the background on the myc-tag stained slides was often high. We have repeated these experiments, and the new panels are now included in **Figs. 2, 4** and **Ext. Data Fig. 2**.

Fig. 2. Mice on HiAD develop more transformed foci following hydrodynamic injection, and exhibit AGEs-dependent higher viscoelasticity.

e. GS/myc immunohistochemistry in mice on chow, FFD and HiAD following HDI and PM/ALT, vs. vehicle treatment. Scale bar, 300 μm .

2. Figure 3b&c should be quantified using CT-FIRE or similar tools.

As suggested, we performed quantification using CT fire, and the data are included in **Fig. 3**. These show that fiber lengths, and fiber-fiber angles decreased in decellularized samples of HiAD mice (**Fig. 3b**), as well as in collagen hydrogels exposed to AGEs (**Fig. 3c**).

Fig. 3. AGEs modulate collagen architecture and network connectivity leading to enhanced viscoelasticity in mouse livers and 3D hydrogels.

b. Collagen fibers in decellularized liver ECM from chow and HiAD-fed mice were analyzed by second harmonic generation (SHG) microscopy. Red arrows indicate altered collagen architecture with bundle formation. Scale bar, 100 μm. Fiber length, studied by CT-fire and angle (ImageJ) decreased in decellularized samples of HiAD mice.

c. Collagen fibers in collagen hydrogels were analyzed by SHG. Collagen was pre- incubated with BSA or AGEs at 4 °C for 4 weeks. AGEs promoted bundling of fibers, scale bar, 100 μm. Fiber length, studied by CT-fire and angle (ImageJ) decreased in collagen+AGEs hydrogels. 5 pictures were assessed in each sample, in each group (n=3, mean ± SEM; ****P < 0.0001, unpaired t test).

3. A more detailed mapping of the relative location of the thicker collagen bundles in HiAD mice and other liver cells types would be very interesting. In particular, are the bundles associated with hepatocytes (potentially with nuclear YAP1) or activated hepatic stellate cells.-

We imaged collagen bundles (SHG), and performed immunofluorescence for nuclear active (non-phosphorylated) YAP, and anti-smooth muscle α actin (α-SMA) to visualize active stellate cells. Hepatocytes were YAP negative in the proximity of the collagen bundles (**a**). No visible collagen deposition occurred yet around hepatocytes with nuclear active YAP at this early stages fibrosis but αSMA-expressing active HSC were seen (**b**, yellow arrows).

Extended Data Fig. 7. Representative images of collagen bundles and YAP immunofluorescence in HiAD liver tissues

a, Collagen was imaged using SHG (green). Active YAP (red) and α -SMA positive stellate cells (blue, yellow arrows) were identified by immunofluorescence. There were no hepatocytes with active nuclear YAP (using an antibody against the non-phosphorylated active YAP) observed in the close proximity of the collagen bundles. **b** There were α -SMA positive stellate cells seen near the YAP positive hepatocytes (Scale bar 50 μ m).

4. Figure 3d – the reconstituted collagen matrix is pretty soft. The authors should exclude the effect of ALTs on collagen that has a higher starting elastic modulus.-

Thanks for this comment. We performed these experiments at a higher starting elastic modulus by generating pure collagen gels with a concentration of 3.6 mg/ml. Collagen architecture (Suppl. Data Fig. 1a), the storage modulus (Suppl. Data Fig. 1b), loss tangent (Suppl. Data Fig. 1c) and $\tau_{1/2}$ (Suppl. Data Fig. 1d) did not change significantly after adding ALT to collagen hydrogels with a higher starting elastic modulus.

Suppl. Data Fig. 1. ALT treatment alone did not affect collagen network (a) or matrix stiffness (b) or viscoelasticity (c, d).

Telopeptide intact form of collagen (Product Number 354236, Corning® Collagen I, Rat Tail) was pre-incubated with or without Alagebrium chloride (ALT, 20 mg/ml) at 4°C for 4 weeks. Collagen was neutralized with 1× NaOH (Merck) to form hydrogels at 3.6 mg/ml final concentration. Collagen fibers in the hydrogels were analyzed by SHG (a). Scale bar, 100 μ m. Collagen gels were loaded on the rheometer for stiffness (b, storage modulus) and viscoelasticity (c, loss tangent; d, $\tau_{1/2}$). (n=4-5, mean \pm SEM, ns not significant, unpaired t-test).

5. Can the authors clarify if the collagen used in in vitro assays is telopeptide intact or pepsin digested? I could not find this information, only the supplier name. Hopefully, they have used the telopeptide intact form as it is more physiologically relevant.-

We apologize for not including this information, we indeed used the telopeptide intact form of collagen (Product Number 354236, Corning® Collagen I, Rat Tail) in all experiments. We added this information to the Materials and Methods.

6. Although the authors report that AGE does not affect the elastic modulus of collagen when using bulk measurement methods, they should also explore if it leads to increased heterogeneity in local elastic modulus by AFM mapping. The collagen images in Figure 3c suggest that the mechanical heterogeneity of the matrix is likely to be changed. It may be that viscoelastic properties of the collagen network allow higher elastic moduli to develop in focal patches.

Thanks for the comment. We performed AFM mapping of AGEs-treated and untreated collagen hydrogels (Ext. Data Fig. 4.). We found that while there was heterogeneity in local elastic moduli in both cases, the range of elastic moduli was slightly higher in the collagen only case (Ext. Data Fig. 4a, b). The frequency distribution of hysteresis areas (viscoelasticity), demonstrates higher frequencies in Collagen+AGEs gels (Ext. Data Fig. 4c, d).

Extended Data Figure 4. Representative areas of collagen and collagen-AGEs hydrogels mapped by AFM for Young's moduli (a, b), and for hysteresis areas (c, d).

- a.** Local patches with higher Young's moduli could be observed in both hydrogels (red arrows).
- b.** The ranges of elastic moduli in both hydrogels.
- c.** Mapping and distribution of hysteresis areas (viscoelasticity),
- d.** Higher frequencies of increased viscoelasticity in collagen+AGEs gels. n=4-5 gels/each group, 3 representative areas/each gel. For all maps, x's indicate regions where AFM indentation curves could not be reliably analyzed. Scale bar, 20µm.

Related to this, are the filopodia documented later in the paper preferentially adhering to the larger collagen bundles.

To visualize invadopodia, we transfected Huh7 cells with a plasmid containing Tks5-mScarlet, and embedded them in collagen hydrogels with or without AGEs. In low viscoelasticity hydrogels (collagen, only) cells did not form invadopodia. In collagen+AGEs hydrogels (high viscoelasticity), invadopodia did not appear to clearly attach to collagen bundles (white arrows, scale bar: 20 μ m, **Reply to Reviewer Fig. 3**).

Reply to reviewer Figure R3.

Huh-7 cells were transfected with Tks-mScarlet and embedded in low or high viscoelasticity hydrogels. In low viscoelasticity hydrogels (collagen, only) the cells were rounded. In high viscoelasticity hydrogels (Collagen+AGEs), invadopodia formation was visible. Invadopodia did not appear to attach to collagen bundles (arrows). Representative images, multiphoton microscopy.

7. LOX enzymes and crosslinking have been reported to confer elastic properties on collagen gels (e.g Petrie et al 2012). Does LOX-mediated crosslinking counteract the effect of ALT? More generally, some consideration of how other classes of collagen modifying enzymes, including MMPs, change during NASH and type II diabetes would be welcome.

The reviewer brings up several excellent suggestions. In our studies, we focused on earlier stages of the disease. At this stage, using ALT increased the solubility of the collagen (**Fig. 3a**), restored viscoelasticity (**Fig. 1o, p, r-t**), and had an important effect on network remodeling. We find that at the pre-cirrhotic stage there is no significant increase in LOXL2 yet in NASH/T2DM patients however, it is significantly induced in cirrhotic patients {REDACTED}.

Thus, collagen elastic properties may dominate at the cirrhotic stage of NASH. We did address this by testing fibrotic and non-fibrotic areas in cirrhotic NASH patients by AFM, (also at the request of Reviewer 3). We found that while fibrotic areas had increased storage moduli, non-fibrotic areas in the liver had increased viscoelasticity in patients with NASH/T2DM, compared to those without T2DM {REDACTED}. Regarding MMPs, MMP9 was significantly induced in cirrhotic patients whereas MT1-MMP (MMP14) was induced in T2DM+NASH. Furthermore, we found MT1-MMP was also induced in mouse livers on HiAD, in an AGEs-dependent manner {REDACTED}.

Related to this, we show new data with MT1-MMP signal at the tips of invadopodia in high viscoelasticity hydrogels (**Main Fig. 5d**).

REDACTED

REDACTED

{REDACTED}

Fig. 5. Invadopodia formation in high viscoelasticity hydrogels.

Invadopodia formation was analyzed after transfecting cells with the Clontech-N1 plasmid containing human Tks5-mNeonGreen (green). Immunofluorescence analysis for MT1-MMP (red) and Airyscan microscopy depict the signals (d, scale bar, 10 μ m).

8. The authors should use their nice inter-penetrating compound matrix system to vary both elastic and loss moduli in a systematic manner. The expectation is that if the loss modulus is kept low and unchanging, then increasing the elastic modulus should lead to increased YAP activity.-

Thank you for this comment. To study if increasing elastic moduli change YAP activity, we encapsulated Huh7 cells in low or high viscoelasticity IPN hydrogels with varying stiffness (0.8-5 kPa, we chose this range to simulate the range in a pre-cirrhotic liver that is the subject of our studies). Cell proliferation was evaluated by Edu nuclear signal (**a**, Scale bar, 100 μ m) and quantification (**c**). YAP activity was analyzed by active YAP immunofluorescence (**b**, Scale bar, 50 μ m), quantification (**d**), and mRNA expression of target genes CTGF and Cyr61 (**e**). We did not observe a stiffness related increase in cell proliferation, YAP activation, or expression of its target genes within this stiffness range (**Extended Data Fig. 8**). In cirrhotic livers with very high stiffness >15 kPa, YAP activation can occur in hepatocytes, but with this material system we were not able to reach and evaluate this range.

Extended Data Fig. 8. Proliferation and Yap activation are not affected by increasing stiffness in low or high viscoelasticity hydrogels. Huh7 cells were encapsulated in low or high viscoelasticity IPN hydrogels with varying stiffness. (0.8-5 kPa).

Cell proliferation was evaluated by Edu nuclear signal (**a**, Scale bar, 100 μ m) and quantification (**c**). YAP activity was analyzed by active YAP immunofluorescence (**b**, Scale bar, 50 μ m), quantification (**d**), and mRNA expression of target genes CTGF and Cyr61 (**e**). Scale bar is 50 μ m. (n=4, mean \pm SEM; ***p<0.001 ****p<0.0001, ANOVA, post-hoc Tukey test).

9. The authors refer to active YAP immune-fluorescence – what do they mean? Is it total YAP staining, followed by evaluation of its nuclear vs cytoplasmic localization? Also, the staining in Figure 5b is not convincing – the brightness of the nuclear signal does not seem consistent with a translocation of the cytoplasmic signal into the nucleus.-

Thanks for this comment. To clarify, we used an antibody against active, non-phosphorylated YAP throughout the paper. We added the reference to the legends. To corroborate our data, we performed western blots depicting cytoplasmic/nuclear YAP signals (**Suppl. Fig. 2.**). We also improved the quality of all figures with YAP immunofluorescence (**Fig. 5c, and Extended Data Fig. 8, 10, 11, 12**).

Suppl. Data Fig. 2. Antibody validation

a, Nuclear and cytoplasmic YAP was assessed using antibodies against active, non-phosphorylated and inactive phosphorylated YAP in western blots, in low and high viscoelasticity hydrogels (cytoplasmic and nuclear fractions).

Fig. 5. Yap activation in high viscoelasticity hydrogels.

After 1 day in 3D culture in low or high viscoelasticity hydrogels, YAP activity was analyzed using an antibody against active YAP (**c**, scale bar, 20 μm).

10. The authors should measure proliferation using EdU staining or measuring changes in cell number, not Ki67, in the in vitro assays in Figure 5.-

Thank you for the suggestion. We replaced the proliferation data with Edu analyses. All the conclusions regarding proliferation remain the same.

11. The invadopodia staining is not convincing. There is no evidence of matrix proteolysis, which is central to the definition of invadopodia- cortactin puncta are not equivalent to invadopodia. It would be more informative to show higher magnification images of integrin-mediated adhesions. Staining using conformation specific active integrin β1 antibodies would be informative, as would staining for focal and fibrillar adhesion components and phosphorylated myosin light chain.

Thanks for this important comment. We have transfected cells with the Tks5-mNeonGreen plasmid, as Tks5 is a key marker of invadopodia, and placed them in low or high viscoelasticity hydrogels. Immunofluorescence analyses with Airyscan high resolution microscopy (LSM980, ZEISS) using MT1-MMP antibody on transfected cells reveal Tks5 and MT1-MMP positive protrusions (**Fig. 5d**). In addition, as recommended we used the mouse monoclonal [12G10] antibody against active β1 integrin, and found it was localized to Tks5 positive invadopodia in high, but not low viscoelasticity hydrogels (**Ext. Data Fig. 9b**). In low viscoelasticity condition, there were only very few invadopodia, and the cells remained more circular.

The p-MLC2 signal also localized to invadopodia in high viscoelasticity hydrogels (**Ext. Data Fig. 9c**).

Together, the morphological characteristics of the protrusions, and the positive signal for Tks5, MT1-MMP, active β1 integrin, and p-MLC2, confirm the protrusions to be invadopodia.

Extended Data Fig. 9: Huh 7 cells cultured in high viscoelasticity hydrogels form invadopodia.

a. Schematic depiction of invadopodia. **b-c.** Huh 7 cells were transfected with Tks5-mNeonGreen, and immunofluorescence microscopy were performed using an antibody against MT1-MMP (b), active integrin $\beta 1$ (c) and p-MLC2 (d). Cells were imaged by the high resolution Airyscan (LSM980, Zeiss) microscopy.

Fig. 5d. Invadopodia formation in high viscoelasticity hydrogels.

Invadopodia formation was analyzed after transfecting cells with the Clontech-N1 plasmid containing human Tks5-mNeonGreen (green). Immunofluorescence analysis for MT1-MMP (red) and Airyscan microscopy depict the signals (d, scale bar, 10 μ m).

12. The tensin analysis in Extended Data Figure 7 is not convincing. The result is marginal and it appears that the authors have selected their cut-off to maximize the difference. They should choose more objective criteria for their cut-off (such as top and bottom quartiles or above and below the median) and present analysis for more than one cohort.-

The data derived from the TCGA database, and there were only 99 patients with HCC after viral, alcohol and other etiologies excluded (a). While we observed a strong trend towards worse survival with higher TNS1 expression in patients with NASH/HCC (b) vs. other causes of HCC (c, median survival represented), after an extensive search we could not find other well-phenotyped cohorts for this population, especially for pre-cirrhotic HCC. NASH/HCC patients overall had higher TNS1 compared to other etiologies (d), this was also seen in an independent study (Wong et al. J Hepatol. 2022 Aug;77(2):410-423).

We have an ongoing biobanking at our institution and are developing a well characterized cohort with histological, treatment and survival data.

Reply to reviewer Fig. R7. TNS1 in NASH/HCC patients is linked to worse survival compared to HCC of other etiologies.

- a. Proportional distribution of HCC etiologies in the TCGA database.
- b, c. Survival in patients with high or low TNS1 expression in NASH-HCC (b) and non-NASH HCC (c).
- d. TNS1 expression is higher in NASH-HCC cohort compared to non-NASH HCC. ***p<0.005.

Referee #3 (Remarks to the Author):

This study is showing an additional evidence of ECM regulation of HCC development, particularly in the condition of T2DM. The authors first determined that hepatic AGE levels are increased in NASH patients with T2DM, which is associated with increased viscoelasticity in the liver. Then, the study showed high AGE diet increased hepatic AGEs along with viscoelasticity without increased stiffness. The importance of AGE and AGE-mediated signaling in NASH-HCC development was also validated in the HCC animal model. Subsequently, the authors determined that the AGE-collagen crosslink regulates viscoelasticity. Additionally, the study determined that integrin b1-TNS1-YAP signaling is crucial for AGE-mediated HCC development. Overall, the experimental approaches the study used are highly mechanistic. The data provided supported their hypothesis. Because in addition to cirrhotic background, NAFLD-HCC is also known to develop in non-cirrhotic livers, this study that determined the critical pathways that are activated in non-cirrhotic liver, especially T2DM, which promotes HCC development is highly significance. However, there are several gaps, which need to be determined.

We thank the reviewer for the positive comments, and constructive suggestions. We have addressed all the reviewer's comments as described below.

Specific comments:

1. Figure 1b. Liver AGE levels was increased in NASH/T2DM, but not in NASH only. What is hepatic AGE level in T2DM without NAFLD (NAFL + NASH)? This is also an important condition. If this is altered, hepatic AGE and/or viscoelasticity (Fig.1e) is altered prior to or without NASH development.-

Thank you for this important comment. We tested liver samples of patients with T2DM without known NASH, and found that AGEs were increased. Of note, on histology 2 of the samples had steatosis (5-10%), but no inflammation or fibrosis. This is common for patients with T2DM, different studies estimate the prevalence of steatosis in T2DM 30-80%. The stiffness (Young's modulus or storage modulus) was not significantly different whereas viscoelasticity (hysteresis, loss tangent, and stress relaxation) has increased (Fig. 1b, d-e). Thus, matrix remodeling and viscoelastic changes can precede NASH/fibrosis in T2DM.

Fig. 1: Viscoelasticity is increased in livers of patients with T2DM and NASH/T2DM.

- a. Schematic representation of AGEs increase in NASH/T2DM. The illustration was created using BioRender.com.
- b. AGEs are increased in livers of T2DM, NASH/T2DM patients compared to healthy or NASH livers (n=4-6, mean±SEM, *p < 0.05, **p < 0.01, ANOVA, Tukey's post hoc).
- c. Schematic of the Atomic Force Microscopy (AFM). Indent-retract was used to get force-distance curves. Indent-constant height was used to get stress relaxation curves.
- d. Atomic Force Microscopy (AFM) experiments were performed on snap-frozen human liver samples from healthy subjects and those with T2DM, NASH or NASH with T2DM, respectively. Liver stiffness is represented as Young's modulus. (n=4-6, mean±SEM, ns not significant, ANOVA, Tukey's post hoc). The average stiffness of a human cirrhotic liver was indicated by a dashed line.
- e, f. Representative force-distance curves show larger hysteresis areas (arrow, the area between approach and retraction force curves) in patients with T2DM (e). Trapz function in MATLAB was used to measure the hysteresis area in (e), 50 force-distance curves were measured in each sample, each group (f). (n=4-6, mean±SEM, *p < 0.05, ****p < 0.0001, ANOVA, Tukey's post hoc).
- g. Schematics depicting the rheometry tests performed on human fresh liver tissues.
- h, i. Rheometry was performed with 100~300 Pa of initial force. Dynamic time sweep test (2% constant strain, oscillation frequency 1 radian/s, measurements for 600s) was done first to collect the storage modulus (stiffness) and loss tangent. There were no significant differences between healthy samples and those from NASH w/o T2DM. Dynamic time sweep test showed a significant change of loss tangent (i) but not storage modulus (stiffness) (h) in livers from T2DM, or NASH patients with T2DM (n=4-6, mean±SEM, *p < 0.05, **p < 0.01, ns not significant, ANOVA, Tukey's post hoc).

2. Not only liver AGE levels, the AGE levels of tumor lesions with and without T2DM should also be evaluated.-

We have obtained samples of tumor lesions (tumor, T) and surrounding parenchyma (non-tumor, NT) from NASH patients with/without T2DM. The AGE levels were overall higher in patients with T2DM, however in the tumors were not significantly higher compared to the surrounding liver tissue {REDACTED}.

REDACTED

3. Is viscoelasticity associated with AGEs, independent of collagen deposition?-

Our findings indicate that ALT treatment that reduces AGE/collagen association restores normal viscoelasticity. This means that modulation of the collagen network architecture is a major factor. However, it is possible that AGEs can affect other components e.g. glycosaminoglycans in the ECM, causing structural and biophysical changes. This could be an exciting area to study in the future.

In NASH-HCC with cirrhosis, does AGE-mediated viscoelasticity is also a crucial factor for promoting HCC development? Is this the same mechanism between NASH-HCC with and without cirrhosis? Or in cirrhosis case, does stiffness play more important role in activating tumor-promoting cellular signaling? Or is still AGE and viscoelasticity important?

To answer this interesting comment, we tested samples from cirrhotic NASH patients by AFM, both in the fibrotic and non-fibrotic areas. As expected, stiffness (Young's modulus) increased in the fibrotic areas in both groups. Viscoelasticity increased in those patients with T2DM, in the non-fibrotic areas, whereas overall viscoelasticity was lower over the fibrotic bands. Therefore, we postulate that depending on where HCC originates in the liver, both viscoelasticity (in T2DM), and stiffness can play a unique role in HCC progression. However, exploring this further with in vivo studies would require an extensive work. We will plan to perform these studies in the future.

REDACTED

Because ALT treatment has a protective effect, collagen should play some roles. Also, ALT treatment reduced hepatic AGE level (Fig.1m). It is unclear how the AGE-collagen crosslink regulates hepatic AGE level.- Our findings indicate that in mice treated with ALT, AGER1 (AGE uptake receptor) levels were restored, resulting in an improved uptake/metabolism of AGEs {REDACTED}. This could potentially be linked to lower RAGE expression with improving viscoelasticity {REDACTED}. In hydrogel studies we found that in the higher viscoelastic ECM, RAGE was induced {REDACTED}, whereas AGER1 was downregulated {REDACTED}. Thus restoring viscoelasticity can improve AGE uptake. In our previous study (JCI, 2020) at a cellular level, we showed how RAGE induction results in cullin 3 neddylation and lower Nrf2 transcriptional stimulus for AGER1 in high AGE conditions {REDACTED}. It would be interesting to explore RAGE/AGER1 regulation in different ECMs in future studies.

REDACTED

4. Figure 2 analyzed the role of AGE in NASH-HCC development. The study measured AGE levels in liver tissues (which could include both tumor lesions and non-tumor livers)- This reviewer is curious about both AGE and collagen levels in tumor lesions. Are AGE and collagen in HCC lesion increased compared with those in non-tumor livers? This reviewer wondered whether AGE and AGE-collagen crosslink play a role only in HCC lesions or are also important in non-tumor livers that affect to HCC development and growth by unknown secreting factors from hepatocytes. Does High AGE diet increase collagen content in HCC lesion? Is collagen produced from activated HSCs or pre-activated HSCs.

Our current premise is that the AGE-modified liver matrix with higher viscoelasticity is a pro-tumoral niche, facilitating mechano-signals that lead to invadopodia formation and proliferation of transformed cells. In this study, we tested AGEs in non-tumoral tissues (Fig. 1).

As to what happens in the tumor tissue, is an interesting question. Our data indicate that AGEs and collagen were not significantly increased within the lesions in HiAD fed mice {REDACTED}, compared to the surrounding tissue, similar to patient data {REDACTED}. The collagen amount, tested by OH-proline assay, was not significantly different in the tumor, compared to the surrounding tissue {REDACTED}. α SMA positive cells (red) started to accumulate around transformed cells {REDACTED}. Although CAFs are not a direct subject of our study, based on prior studies a certain sub-population of these cells is responsible for peritumoral collagen production (Filliol et al, Nature. 2022 Oct; 610(7931): 356–365.).

REDACTED

5. Fig.4f-g, the effect of dnTEAD2 in YAP activity in HCC lesion should be shown. Also, the assessment of YAP downstream targets is required.-

Thanks for this comment. We show that active non-phosphorylated YAP (red) localized to the nuclei of GS positive cells (green) in control vector (NC) injected mice on HiAD. In dnTEAD2 injected mice much lower number of GS positive cells were seen, and these were negative for active, nuclear YAP. Downstream targets CTGF and Cyr61 were downregulated after DN-TEAD2 injection (**Fig. 4j**).

Fig. 4 YAP is involved in HCC growth promoted by high viscoelasticity.

h-i. GS/myc immunohistochemistry depict colocalization (**h**, top and middle row) and quantification (**i**) of GS/myc positive foci showing an increase in double positive foci in control vector-injected mice. Less foci were seen in those injected with dn-TEAD2. Scale bar, 300 μ m. (n=5, mean \pm SEM; ****p < 0.0001, ns not significant, ANOVA, post-hoc Tukey test).

Active non-phosphorylated YAP (red) localized to the nuclei of GS positive cells (lower row, green) in control vector injected mice on HiAD. In the dnTEAD2-injected mice much lower number of GS positive cells were seen, and these had less nuclear YAP. Scale bar, 100 μ m

j. YAP targets CTGF and Cyr61 were downregulated after dn TEAD2 injection. (n=5, mean \pm SEM; **p<0.01, ***p<0.001, ****p < 0.0001, ANOVA, post-hoc Tukey test)

6. Fig. 4-5. The study showed CTGF as a YAP regulated gene. More comprehensive YAP signaling assessment is required - more markers and more intermediate signaling molecular states; whether integrin and TNS1 inhibition reduces YAP activity more clearly. Fig.5q showed "active YAP". However, this is unclear. Is this either phosphorylated YAP or nuclear total YAP?

We used an antibody against active YAP that recognizes the non-phosphorylated, active form (Recombinant Anti-active YAP1 antibody [EPR19812], Abcam). To confirm our data, we also present studies with western blots and nuclear/cytoplasmic YAP immunostaining (**Suppl. Data Fig. 2**).

To further assess the effect on the YAP pathway, we targeted TNS1 or integrin β 1 by CRISPR/Cas KD, vs control sgRNA. RhoA activation (GTP-bound state), LATS1 (phospho and total) and phospho (active), and non-phosphorylated YAP were assessed in each condition by western blots (**Fig. 5r**). Our data indicate that RhoA-GTP, and YAP activation increased in high viscoelasticity hydrogels but reduced after integrin β 1 or TNS1 knockdown. We also included studies on cell proliferation, active YAP signal, Cyr61 and CTGF as downstream targets as well as data on cell circularity, in all conditions (**Fig. 5**).

Suppl. Data Fig. 2. Antibody validation

a, Nuclear and cytoplasmic YAP was assessed using antibodies against active, non-phosphorylated and inactive phosphorylated YAP in western blots, in low and high viscoelasticity hydrogels (cytoplasmic and nuclear fractions).

Fig. 5r. Integrin and β 1-Tensin are involved in mechano-signaling leading to YAP activation in high viscoelasticity hydrogels.

r. RhoA GTPase activity was studied in low/high viscoelasticity conditions and after TNS-1 or Integrin β 1 KDs. Phosphorylated and total LATS1, furthermore active (non-phosphorylated YAP), and inactive YAP (Phosphorylated YAP), were analyzed by immunoblotting. (Representative blot)

Fig. 5: Integrin β 1-Tensin 1-YAP axis mediates viscoelasticity-specific mechano-cellular pathways

n-q: Proliferation (**n**), YAP activation (% cells with active nuclear YAP, **o**), and its targets CTGF and Cyr61 (**p**) decreased after TNS1 and integrin β 1 KDs, in high viscoelasticity hydrogels while cell circularity(**q**) has improved (n=5, mean \pm SEM; *p<0.05, **p<0.01, ***p<0.001, ****p < 0.0001, ANOVA, post-hoc Tukey test).

7. The study nicely showed the effect of Itgb1 and TNS1, but in vivo evidence of Itgb1, TNS1, and YAP needs to be validated. Also, it is unclear how integrin β 1 is activated. Is integrin β 1 activated by collagen, AGE, or crosslinked AGE-collagen?

We have generated a new mouse model where we used CRISPR/Cas KD of Tensin1 in conjunction with the hydrodynamic injection (**Fig. 6**). In this model, we found decreased formation of transformed foci, and decreased YAP targets CTGF and Cyr61.

To address the *in vivo* relevance of YAP, we have presented data with the dn-TEAD2 model (Fig. 4). In this, we extended our data by YAP staining and presenting downstream targets.

We observed activated integrin $\beta 1$ signal in HiAD-fed mice, that decreased after Tensin1 KD [REDACTED]. Integrins can be activated from a closed, bent conformation to an extended, open conformation by intracellular and extracellular signaling cues ('inside-out' and 'outside-in'). Interaction between integrins and their ECM ligands (collagen /fibronectin/laminin/etc.) can activate integrins and downstream signaling through changing mechanical force. It is known that integrin $\beta 1$ engages collagen; thus, we expect viscoelasticity might impact the lifetime and affinity of this binding, as well as integrin clustering, as has been shown previously (Adebowale, Nature Materials 2021; Chaudhuri, Nature Materials 2016). Additionally, RhoA is a critical intracellular signal that drives integrin activation inside-out, and we showed increased activation in high viscoelastic hydrogels (Fig 5r, above). In future studies, it would be interesting to explore RhoA and integrins feed-forward signaling in different ECMs.

Figure 6: TNS-1 knockdown decreased formation of transformed foci after HDI

a. Schematic presentation of the *in vivo* targeting of TNS1 by CRISPR/Cas9 in conjunction with hydrodynamic injection. Mice were fed chow or HiAD for 7w, then hydrodynamically injected with pT3-EF5a-hMet and the pT3-354 EF5a-S45Y- β -catenin-myc (mutant β -catenin), with the sleeping beauty (SB) transposase, as well as the CRISPR-Cas9-based vector linking two sgRNAs targeting mouse TNS1 (pX333-sgTNS1) or empty vector (sgNC). Mice were sacrificed 7 weeks following injection. **b.** TNS1 expression was analyzed by RT-qPCR. (n=6, mean \pm SEM; **p < 0.01, ****p < 0.0001, ns not significant, ANOVA, post-hoc Tukey test) **c, d.** GS/myc immunohistochemistry depict the signals (c) and quantification (d) of GS/myc positive foci. Scale bar, 300 μ m. (n=6, mean \pm SEM; *p < 0.05, **P < 0.01, ANOVA, post-hoc Tukey test) **e.** The expression of YAP targets gene CTGF and Cyr 61 was analyzed by RT-qPCR. (n=6, mean \pm SEM; *p < 0.05, **p < 0.01, **p < 0.001 ANOVA, post-hoc Tukey test). **f.** Schematic presentation of TNS1 that serves as a key component of the ECM mechano-sensor complex by binding to integrin $\beta 1$ in high viscoelasticity ECM conditions. The illustration was created using BioRender.com.

REDACTED

Reviewer Reports on the First Revision:

Referees' comments:

Referee #1 (Remarks to the Author):

The authors have performed a series of new experiments and satisfactorily addressed the reviewer's concerns. This reviewer finds the revised manuscript worthy of publication in Nature, pending some very minor revisions:

1. Lines 86-88 "These (HiAD) mice exhibit... higher liver AGEs compared to those on regular (chow) or FFD (Fig. 1l, m)". In view of this statement, the authors should also show statistical significance between HiAD (vehicle) and FFD.
2. Lines 135-138: the authors should add "and HiAD-fed mice" as suggested here: "The collagen network exhibited bundling in patients with T2DM/NASH, and mice on HiAD, whereas the network appeared more organized with thinner fibers in healthy human livers, mice on chow diet, "and HiAD-fed mice" following PM, ALT treatment or AGER1 reconstitution".
3. Quantification of RhoA activity from Fig. 5r should be added, possibly in the Extended Data Fig. 13.

Referee #2 (Remarks to the Author):

This revised study documents the linkage between advanced glycation end-products (AGEs), which are linked to diet, and aggressive liver cancer. In particular, AGEs are shown to alter extracellular matrix (ECM) mechanics leading to increased viscosity and greater YAP1 activation. The study is comprehensive and generally performed to a high standard. The authors have addressed many of the concerns raised previously.

Remaining issues and questions

1. It remains unclear to this reviewer why AGEs lead to shorter collagen fibers. This is fairly central to the manuscript. I accept the modelling that shorter fibers with altered bundling leads to a more heterogeneous mechanical landscape with greater stress relaxation. However, the model is not well explained. Bundle length is presented almost as a variable in Ext. Data Fig. 5, but isn't the variable the amount of bundler, and the bundle length is the result. The detailed explanation in the supplementary text is helpful, but it doesn't clarify this. Also, I don't understand what the difference is between the top left panel in Ext. Data Fig. 5h and panel j in the same figure – presumably it is the presence of bundlers in panel j, but how many and what is the resulting bundle length? It doesn't seem to match any of the images in panel i – what is the bundling angle in these panels? Overall, the modelling needs improved presentation and explanation. Crucially, this needs to be better connected with the experimental observations and include a mechanism(s) for AGEs to lead to shorter fibers. In this regard, the authors present interesting new analysis showing elevated MMP2, 9, & 14 in conditions of elevated viscosity. Are they the cause of the shorter collagen fibers? Addressing these issues might require some additional experiments, but these should not be overly time consuming.
2. Parametric statistical tests are used throughout the study and it is not clear if this is appropriate. Certainly, some of the plots look like the criterium of normality would not be met. These needs to be checked thoroughly and the effect on the robustness of any conclusions determined.
3. The authors should probably refer to the invasive structures as invadopodia-like. The dimensions are larger than would be expected for invadopodia. Also, the invadopodia cartoon in Ext. Data Fig. 9a is horrible – the nucleus is tiny compared to the invadopod, the scaling of the other molecular components is weird, some integrins are not connected to actin. It should be

improved or removed.

4. The link between viscoelasticity, YAP1 activation, and aggressive cancer phenotypes was recently reported by Elosegui-Artola et al (Nature Materials). This should be cited and the authors should explain how their work goes beyond this study in terms of mechanistic insight into mechano-transduction. It is clearly distinct in terms of the in vivo analysis.

5. Figure 1d should report the statistical significance of control vs NASH/T2DM

Referee #3 (Remarks to the Author):

The authors very nicely addressed all concerns from this reviewer. There are no further concerns and comments.

One minor comment is that - regarding the Reporting Summary, the authors stated the source of HuH7 cells is ATCC. However, it is well-known that ATCC does not provide HuH7 cells. Please check it more carefully.

Author Rebuttals to First Revision:

Referee #1 (Remarks to the Author):

The authors have performed a series of new experiments and satisfactorily addressed the reviewer's concerns. This reviewer finds the revised manuscript worthy of publication in Nature, pending some very minor revisions:

We thank the Reviewer for the positive comments.

1. Lines 86-88 "These (HiAD) mice exhibit... higher liver AGEs compared to those on regular (chow) or FFD (Fig. 1l, m)". In view of this statement, the authors should also show statistical significance between HiAD (vehicle) and FFD. Thanks, this is now added.

Fig.1m. Liver AGEs increased in mice on HiAD and decreased following PM or ALT treatment. (n=5-9, mean±SEM, *p < 0.05, ***p < 0.001, ****p < 0.0001, ANOVA, Tukey's post hoc).

2. Lines 135-138: the authors should add "and HiAD-fed mice" as suggested here: "The collagen network exhibited bundling in patients with T2DM/NASH, and mice on HiAD, whereas the network appeared more organized with thinner fibers in healthy human livers, mice on chow diet, "and HiAD-fed mice" following PM, ALT treatment or AGER1 reconstitution". We added this, as suggested.

3. Quantification of RhoA activity from Fig. 5r should be added, possibly in the Extended Data Fig. 13. We performed the quantification and it is now added to the Extended Data Fig. 13a.

Extended Data Fig. 13a.

Quantification of GTP-RhoA/GAPDH protein levels from Fig. 5r (n = 3, mean ± SEM; ***p<0.001, ****<.0001, ANOVA, post-hoc Tukey test).

Referee #2 (Remarks to the Author):

This revised study documents the linkage between advanced glycation end-products (AGEs), which are linked to diet, and aggressive liver cancer. In particular, AGEs are shown to alter extracellular matrix (ECM) mechanics leading to increased viscosity and greater YAP1 activation. The study is comprehensive and generally performed to a high standard. The authors have addressed many of the concerns raised previously. Thank you for the positive comment.

Remaining issues and questions

1. It remains unclear to this reviewer why AGEs lead to shorter collagen fibers. This is fairly central to the manuscript. I accept the modelling that shorter fibers with altered bundling leads to a more heterogeneous mechanical landscape with greater stress relaxation. However, the model is not well explained. Bundle length is presented almost as a variable in Ext. Data Fig. 5, but isn't the variable the amount of bundler, and the bundle length is the result. The detailed explanation in the supplementary text is helpful, but it doesn't clarify this. Also, I don't understand what the difference is between the top left panel in Ext. Data Fig.5h and panel j in the same figure – presumably it is the presence of bundlers in panel j, but how many and what is the resulting bundle length? It doesn't seem to match any of the images in panel i – what is the bundling angle in these panels? Overall, the modelling needs improved presentation and explanation. Crucially, this needs to be better connected with the experimental observations and include a mechanism(s) for AGEs to lead to shorter fibers –

We appreciate the reviewer's excellent comment. In the revised manuscript, we improved the clarity of the model description by adding explanations in the section "Simulation Modeling", the caption of Ext. Data Fig. 5, as well as in the Suppl. Material, under "Dynamics of fibrils, bundlers, and cross-linkers".

The two matrices shown in panel h consist of individual fibrils because there are only cross-linkers without bundlers. Matrices

in panel j were created with cross-linkers as well as bundlers that connect fibrils at their ends (at various angles) to form bundles. Matrices in panel i were also created with bundlers, but these bundlers connect fibrils in a parallel, staggering manner into longer, tight bundles. As the reviewer correctly assumed, the bundle length is an output quantity as a result of stochastic binding events between bundlers and fibrils, which can be modulated. The caption and figure have been edited to make this point clearer.

Extended Data Fig. 5. Agent-based computational model for a fibrillar matrix.

a-c. Fibrils (gray, “f”), cross-linkers (yellow, “xl”), and bundlers (red, “bu”) are simplified by cylindrical segments in the model. Cross-linkers connect pairs of fibrils without preference of a cross-linking angle by binding to any part of two fibrils. By contrast, bundlers connect pairs of fibrils with a specific angle and then maintain the angle. The first binding site of bundlers is always located at the end of fibrils, and the second binding site is located at specified part of fibrils. The specific part available for binding is defined by two boundaries, b_1 and b_2 , between 0 and 1. Various bending (κ_b) and extensional (κ_s) stiffnesses maintain angles and lengths near their equilibrium values, respectively. Stiffnesses, equilibrium lengths, and equilibrium angles are listed in Table 5.

d-g. Different types of matrices. Without bundlers, a matrix is comprised of individual fibrils cross-linked to each other, resulting in small mesh size (d). With bundlers which bind only to the ends of fibrils, a matrix consists of short bundles. Depending on the angle between fibrils connected by bundlers, the shape of short bundles varies (e, f). With bundlers that bind to the end of one fibril and the mid of other fibril, a matrix consists of longer bundles (g). Cross-linkers can connect fibrils within each bundle or fibrils that belong to different bundles.

h-j: Snapshots of matrices employed for rheological measurements. The length of fibrils used for creating matrices is either 3 μm (top row) or 5 μm (bottom row).

h. Matrix structures with a homogeneous, fine mesh, which is created without bundlers as shown in d.

i. Matrix structures consisting of long, tight bundles with different lengths. Fibrils are connected in parallel by bundlers as shown in g. The length of bundles can be changed by varying b_1 and b_2 . If the second binding site can bind only to part near one end of fibrils (e.g., $b_1=0.8$, $b_2=1$), the average length of bundles (LB) becomes large. By contrast, if the binding can take place only near the other end of fibrils (e.g., $b_1=0$, $b_2=0.1$), LB is slightly longer than the length of individual fibrils.

j. Matrix structures consisting of short, loose bundles with different bundling angles, θ . In these cases, both binding sites of bundlers bind to the end of fibrils (i.e., $b_1=b_2=0$) as shown in e and f. The shape of the bundle is varied by changing θ .

Regarding the general point of the reviewer, in both in vivo tissues and in vitro conditions, AGEs act to decrease the length of the collagen fibers and increase network heterogeneity.

The observed reduction in collagen fiber length after AGEs can be explained by a lower availability of free telocollagen units, due to AGEs targeting the telopeptide lysine side chains, instigating the Amadori reaction (J Biol Chem. 2018 Oct 5; 293(40): 15620–15627. PMID: 30143533). This reduction could hinder fiber elongation and the formation of a structurally cohesive collagen network in the collagen/AGEs system compared to naturalized collagen. Telopeptides are known to catalyze fibril formation through the "toe-hold" mechanism (Biophys J. 2016 Dec 6; 111(11): 2404–2416. PMID: 27926842).

Alternatively, or complementarily, it is plausible that AGEs modification of telopeptides encourages more a lateral fiber association, promoting lateral growth over elongation. This aligns with our findings of increased width of fibers in the presence of AGEs, in vitro and in vivo (see **Figure R1**).

These would be very interesting questions to pursue through molecular dynamics simulations, or high-resolution microscopy (e.g., TEM) in follow-up studies, but beyond the scope of the current manuscript. We have now discussed this interesting point in the manuscript.

Reply to reviewer Fig. R1. AGEs promote thicker fibers formulation in vivo and in vitro.

a-b, Quantification of collagen fiber width (CT-Fire) in mice and gels from Fig.3b, c. (n=3, mean ± SEM; ****p < 0.0001, Wilcoxon's rank sum test) HiAD: High AGE diet.

c, Quantification of collagen fiber width (CT-Fire) in IPN gels from Fig. 3g. (n=3, mean ± SEM; ****p < 0.0001, Kruskal–Wallis test followed by a post hoc Dunn's test).

In this regard, the authors present interesting new analysis showing elevated MMP2, 9, & 14 in conditions of elevated viscosity. Are they the cause of the shorter collagen fibers? Addressing these issues might require some additional experiments, but these should not be overly time consuming.-

Thanks for this interesting observation. Indeed, MT1-MMP (MMP14), MMP2, 9 were regulated by elevated viscoelasticity. However, these are gelatinases targeting the basement membrane collagen IV, playing a role in invasion.

To better address this question in regard to collagenases (as we used collagen I, the major constituent in our gels), we performed RNAseq analyses in liver tissues on chow, HiAD +/- PM, RAGE^{hepko} background and after FFD, and found that MMP13 collagenase was differentially regulated in high viscoelasticity conditions {REDACTED}. This was confirmed by RT-qPCR {REDACTED}. Next, we used an in-situ digestion method in collagen hydrogels with active MMP13 (Abcam, # ab227435). We found that MMP13 collagenase indeed digested collagen (arrow points to degradation products, {REDACTED}). AFM data show that viscoelasticity following collagenase treatment showed slight increase albeit not significant in this hydrogel system {REDACTED}.

However, we acknowledge that the in vivo situation is likely to be much more complex with the coordinated action of multiple MMPs and their inhibitors (TIMPs) playing a dynamic role. To investigate this would require extensive experiments. In summary, we cannot exclude the possibility that collagen degradation products with shorter fibers play a role in generating higher stress relaxation. This interesting area could be a focus of future studies.

{REDACTED}

{REDACTED}

2. Parametric statistical tests are used throughout the study and it is not clear if this is appropriate. Certainly, some of the plots look like the criterium of normality would not be met. These needs to be checked thoroughly and the effect on the robustness of any conclusions determined.-To address this, we used the Smirnov-Kolmogorov test then performed analyses using parametric and non-parametric tests for each data sets throughout the manuscript. We present data in parallel. (Reply to Reviewer **Fig. R4-R9**). Indeed, for the original Fig. 3 b, c, i, j non-parametric tests had to be used, these were replaced. Otherwise, the overall conclusions for all data sets remain the same.

Reply to reviewer Fig. R4. Parametric and non-parametric statistical tests performed on Fig. 1 datasets
 Fig 1 was analyzed by parametric statistical test (ANOVA plus Tukey's post hoc, top) and a non-parametric statistical test (Kruskal–Wallis test followed by a post hoc Dunn's test, bottom).

Parametric test

Non-parametric test

Reply to reviewer Fig. R5. Parametric and non-parametric statistical tests performed on Fig. 2 datasets

Fig 2 was analyzed by parametric statistical test (ANOVA plus Tukey's post hoc, top) and a non-parametric statistical test (Kruskal–Wallis test followed by a post hoc Dunn's test, bottom).

Parametric test

Non-parametric test

Reply to reviewer Fig. R6. Parametric and non-parametric statistical tests performed on Fig. 3 datasets

Fig 3 was analyzed by parametric statistical test (ANOVA plus Tukey's post hoc, top) and non-parametric statistical test (Fig 3b and c were analyzed by Wilcoxon's rank sum test, the rest of Fig 3 were analyzed by Kruskal–Wallis test plus a post hoc Dunn's test, at the bottom).

Reply to reviewer Fig. R7. Parametric and non-parametric statistical tests performed on Fig. 4 datasets

Fig 4 was analyzed by parametric statistical test (ANOVA plus Tukey's post hoc, top) and a non-parametric statistical test (Kruskal–Wallis test followed by a post hoc Dunn's test, bottom).

Parametric test

Non-parametric test

Reply to reviewer Fig. R8. Parametric and non-parametric statistical tests performed on Fig. 5 datasets

Fig 5 was analyzed by parametric statistical test (ANOVA plus Tukey's post hoc, top) and a non-parametric statistical test (Kruskal–Wallis test followed by a post hoc Dunn's test, bottom).

Reply to reviewer Fig. R9. Parametric or non-parametric statistical tests performed on Fig. 6 datasets

Fig 6 was analyzed by parametric statistical test (ANOVA plus Tukey’s post hoc, top) and a non-parametric statistical test (Kruskal–Wallis test followed by a post hoc Dunn’s test, bottom).

3. The authors should probably refer to the invasive structures as invadopodia-like. The dimensions are larger than would be expected for invadopodia. Also, the invadopodia cartoon in Ext. Data Fig. 9a is horrible – the nucleus is tiny compared to the invadopod, the scaling of the other molecular components is weird, some integrins are not connected to actin. It should be improved or removed. -We removed the diagram and replaced the name as invadopodia-like structures.

4. The link between viscoelasticity, YAP1 activation, and aggressive cancer phenotypes was recently reported by Elosegui-Artola et al (Nature Materials). This should be cited and the authors should explain how their work goes beyond this study in terms of mechanistic insight into mechano-transduction. It is clearly distinct in terms of the in vivo analysis. -Thanks, we added the citation (#17).

5. Figure 1d should report the statistical significance of control vs NASH/T2DM—We added the statistical significance for Fig. 1d.

Fig.1d. Atomic Force Microscopy (AFM) experiments were performed on snap-frozen human liver samples from healthy subjects and those with T2DM, NASH or NASH with T2DM, respectively. Liver stiffness is represented as Young’s modulus. (n=4-6, mean±SEM, ns not significant, ANOVA, Tukey’s post hoc). The average stiffness of a human cirrhotic liver was indicated by a dashed line.

Referee #3 (Remarks to the Author):

The authors very nicely addressed all concerns from this reviewer. There are no further concerns and comments. Thanks for the positive comment.

One minor comment is that - regarding the Reporting Summary, the authors stated the source of HuH7 cells is ATCC. However, it is well-known that ATCC does not provide HuH7 cells. Please check it more carefully.—We apologize for this. We now included the correct source of cells. We have performed the verification by STR profiling using ATCC cell authentication service (ASN-0002-2022).

Reviewer Reports on the Second Revision:

Referees' comments:

Referee #1 (Remarks to the Author):

According to this reviewer's opinion, the authors have explained the basic and constitutive equations of their model well enough. Please see below some minor suggestions that will improve this manuscript:

1. The authors may want to add a sentence about the initial fibril assembly/seeding process during simulation and to what collagen density does it correspond experimentally.
2. There must be a typo in the Stress Relaxation plots (Stress vs Time in Extended Fig. 6b,d,f,g,h); the unit of Stress on the y-axis is mentioned as "s" instead of "Pascal" or "N/m²".
3. In Table 5, there must be a typo in the viscosity value of the medium. 8.6 Pa s is too high for a "water-based" medium. It should be 8.6×10^{-4} Pa s
4. In lines 418-419 (legend of Figs. 3s,t), the authors may want to clarify that: "Stress relaxation is measured using matrices with different bundle length (LB) for a $\theta=10^\circ$ (s) or different bundling angle (θ)(t) for $L_s=3.8 \mu\text{m}$. Authors should check the values added by the reviewer.
5. The x-axis in Extended Fig. 6 a, c could include labels showing how these fiber lengths were obtained, as explained in Extended Fig. 5 i (the range of possible b1 and b2 values).
6. In lines 559-560, the authors mention "Cross linkers connect pairs of fibrils ... by binding to any part of two fibrils". A clearer statement would be "...by binding to binding sites in any part of two fibrils". This is because the authors mention that cross-linkers can only bind at binding sites (which are spaced by 50 nm throughout the fibril) in the supplementary text.

Referee #2 (Remarks to the Author):

The authors have responded well to the comments and I recommend publication. I would advise that the supplementary text describing the model is re-read by a 'fresh pair of eyes' and perhaps made clearer. However, this is a matter for the editor and authors.

Author Rebuttals to Second Revision:

Response to Referee #1 (Remarks to the Author):

According to this reviewer's opinion, the authors have explained the basic and constitutive equations of their model well enough. Please see below some minor suggestions that will improve this manuscript:

Thank you very much for the suggestions, we have updated our main text and figure legends, accordingly.

1. The authors may want to add a sentence about the initial fibril assembly/seeding process during simulation and to what collagen density does it correspond experimentally.

- We appreciate the suggestion. As the reviewer might have noticed, we explained how fibrils are assembled in "Dynamics of fibrils, bundlers, and cross-linkers" in the supplementary material. We added one sentence in "Simulation Modeling" of the main text to describe the fibril assembly: "Fibril assembly is initiated by the nucleation of seed fibrils via appearance of one cylindrical segment in random positions, followed by elongation up to either 3 μm or 5 μm via addition of segments without consideration of depolymerization."

In addition, we added a sentence to mention effective collagen concentration: "Effective collagen concentration calculated using the specific volume of collagen (0.73 ml/g) is 3.65 mg/ml."

2. There must be a typo in the Stress Relaxation plots (Stress vs Time in Extended Fig. 6b,d,f,g,h); the unit of Stress on the y-axis is mentioned as "s" instead of "Pascal" or "N/m²".

- Thank you for pointing this out. We fixed the typo and changed the y-axis unit to Pascal (Pa).

3. In Table 5, there must be a typo in the viscosity value of the medium. 8.6 Pa s is too high for a "water-based" medium. It should be 8.6×10^{-4} Pa s

- We apologize for the typo. The viscosity of the medium used in simulations is 0.86 Pa, not 8.6 Pa. Admittedly, this is ~1,000-fold higher than that of the water-based medium. We used such high viscosity to increase the time step and thus increase the time-scale of simulations. (A time step is directly proportional to the medium viscosity in inertia-neglected systems.) This way has been widely used in our previous studies and other modeling studies with agent-based models (e.g., 0.3 Pa s used in DOI: 10.1371/journal.pcbi.1009506). With high viscosity, there can be greater stress buildup in response to an increase in strain at the beginning of simulations because of high drag forces. To avoid this artifact, in all simulations, without cross-linker unbinding, we increased the strain up to 20% and held the strain at 20% for 100 s to relax drag-induced stress. Then, we activated the cross-linker unbinding. Before running our simulations, we confirmed that initial stress relaxation was very similar between cases run with different medium viscosity, which means that most of the stress relaxation originates from cross-linker unbinding, not drag force. We did not include this comparison in the original manuscript because we felt that it is too detailed information. In the revised manuscript, we corrected the viscosity value in the table and added

more explanation about how the stress relaxation test was done in "Matrix assembly and bulk rheology" of Supplementary Text.

4. In lines 418-419 (legend of Figs. 3s,t), the authors may want to clarify that: "Stress relaxation is measured using matrices with different bundle length (L_B) for a $\theta=10^\circ$ (s) or different bundling angle (θ) (t) for $L_s=3.8 \mu\text{m}$. Authors should check the values added by the reviewer.

- Thank you for the suggestion. We revised the part: "s, t. Stress relaxation is measured using matrices with different bundle length (L_B) with $\theta = 0^\circ$ (s) or different bundling angle (θ) with $L_B = 3 \mu\text{m}$ (t)."

5. The x-axis in Extended Fig. 6 a, c could include labels showing how these fiber lengths were obtained, as explained in Extended Fig. 5 i (the range of possible b_1 and b_2 values).

- Thank you for your suggestion. We added the values of b_1 and b_2 on the x-axis labels to enhance the clarity of Extended Figure 5a and 5c. We also revised the figure caption significantly (Now Ext Data Fig. 6. is Ext Data Fig. 5.).

6. In lines 559-560, the authors mention "Cross linkers connect pairs of fibrils ... by binding to any part of two fibrils". A clearer statement would be "...by binding to binding sites in any part of two fibrils". This is because the authors mention that cross-linkers can only bind at binding sites (which are spaced by 50 nm throughout the fibril) in the supplementary text.

- We appreciate the suggestion. We modified the statement in accordance with the reviewer's recommendation (in Ext. Data Fig. 4a-c).